# Safe and just Earth system boundaries

Johan Rockström[1,2,3 ✉], Joyeeta Gupta[4,5], Dahe Qin[6,7,8], Steven J. Lade[3,9,10 ✉], Jesse F. Abrams[11], Lauren S. Andersen[1], David I. Armstrong McKay[3,11,12], Xuemei Bai[10], Govindasamy Bala[13], Stuart E. Bunn[14], Daniel Ciobanu[3], Fabrice DeClerck[15,16], Kristie Ebi[17], Lauren Gifford[18], Christopher Gordon[19], Syezlin Hasan[14], Norichika Kanie[20], Timothy M. Lenton[11], Sina Loriani[1], Diana M. Liverman[18], Awaz Mohamed[21], Nebojsa Nakicenovic[22], David Obura[23], Daniel Ospina[9], Klaudia Prodani[4], Crelis Rammelt[4], Boris Sakschewski[1], Joeri Scholtens[4], Ben Stewart-Koster[14], Thejna Tharammal[24], Detlef van Vuuren[25,26], Peter H. Verburg[27,28], Ricarda Winkelmann[1,29], Caroline Zimm[22], Elena M. Bennett[30,31], Stefan Bringezu[32], Wendy Broadgate[9], Pamela A. Green[33], Lei Huang[34], Lisa Jacobson[9], Christopher Ndehedehe[14,35], Simona Pedde[9,36], Juan Rocha[3,9], Marten Scheffer[37], Lena Schulte-Uebbing[25,38], Wim de Vries[38], Cunde Xiao[6,39], Chi Xu[40], Xinwu Xu[7,8], Noelia Zafra-Calvo[41] & Xin Zhang[42]

The stability and resilience of the Earth system and human well-being are inseparably linked[1–3], yet their interdependencies are generally under-recognized; consequently, they are often treated independently[4,5]. Here, we use modelling and literature assessment to quantify safe and just Earth system boundaries (ESBs) for climate, the biosphere, water and nutrient cycles, and aerosols at global and subglobal scales. We propose ESBs for maintaining the resilience and stability of the Earth system (safe ESBs) and minimizing exposure to significant harm to humans from Earth system change (a necessary but not sufficient condition for justice)[4]. The stricter of the safe or just boundaries sets the integrated safe and just ESB. Our findings show that justice considerations constrain the integrated ESBs more than safety considerations for climate and atmospheric aerosol loading. Seven of eight globally quantified safe and just ESBs and at least two regional safe and just ESBs in over half of global land area are already exceeded. We propose that our assessment provides a quantitative foundation for safeguarding the global commons for all people now and into the future.

Humanity is well into the Anthropocene[6], the proposed new geological epoch where human pressures have put the Earth system on a trajectory moving rapidly away from the stable Holocene state of the past 12,000 years, which is the only state of the Earth system we have evidence of being able to support the world as we know it[7,8]. These rapid changes to the Earth system undermine critical life-support systems[1,9,10], with significant societal impacts already felt[1,3], and they could lead to triggering tipping points that irreversibly destabilize the Earth system[7,11,12]. These changes are mostly driven by social and economic systems run on unsustainable resource extraction and consumption. Contributions to Earth system change and the consequences of its impacts vary greatly among social groups and countries. Given these interdependencies between inclusive human development and a stable and resilient Earth system[1–3,13], an assessment of safe and just

[1]Potsdam Institute for Climate Impact Research (PIK), Member of the Leibniz Association, Potsdam, Germany. [2]Institute of Environmental Science and Geography, University of Potsdam, Potsdam, Germany. [3]Stockholm Resilience Centre, Stockholm University, Stockholm, Sweden. [4]Amsterdam Institute for Social Science Research, University of Amsterdam, Amsterdam, The Netherlands. [5]IHE Delft Institute for Water Education, Delft, The Netherlands. [6]State Key Laboratory of Cryospheric Science, Northwest Institute of Eco-Environment and Resources, Chinese Academy of Sciences, Lanzhou, China. [7]China Meteorological Administration, Beijing, China. [8]University of Chinese Academy of Sciences, Beijing, China. [9]Future Earth Secretariat, Stockholm, Sweden. [10]Fenner School of Environment & Society, Australian National University, Canberra, Australia. [11]Global Systems Institute, University of Exeter, Exeter, UK. [12]Georesilience Analytics, Leatherhead, UK. [13]Center for Atmospheric and Oceanic Sciences, Indian Institute of Science, Bengaluru, India. [14]Australian Rivers Institute, Griffith University, Brisbane, Australia. [15]EAT, Oslo, Norway. [16]Alliance of Bioversity International and CIAT of the CGIAR, Montpellier, France. [17]Center for Health & the Global Environment, University of Washington, Seattle, WA, USA. [18]School of Geography, Development and Environment, University of Arizona, Tucson, AZ, USA. [19]Institute for Environment and Sanitation Studies, University of Ghana, Legon, Ghana. [20]Graduate School of Media and Governance, Keio University, Fujisawa, Japan. [21]Functional Forest Ecology, Universität Hamburg, Barsbüttel, Germany. [22]International Institute for Applied Systems Analysis, Laxenburg, Austria. [23]CORDIO East Africa, Mombasa, Kenya. [24]Interdisciplinary Center for Water Research, Indian Institute of Science, Bengaluru, India. [25]Copernicus Institute of Sustainable Development, Utrecht University, Utrecht, The Netherlands. [26]PBL Netherlands Environmental Assessment Agency, The Hague, The Netherlands. [27]Swiss Federal Institute for Forest, Snow and Landscape Research, Birmensdorf, Switzerland. [28]Institute for Environmental Studies, Vrije Universiteit Amsterdam, Amsterdam, The Netherlands. [29]Institute of Physics and Astronomy, University of Potsdam, Potsdam, Germany. [30]Bieler School of Environment, McGill University, Montreal, Canada. [31]Department of Natural Resource Sciences, McGill University, Montreal, Canada. [32]Center for Environmental Systems Research, Kassel University, Kassel, Germany. [33]Environmental Sciences Initiative, Advanced Science Research Center at the Graduate Center, City University of New York, New York, NY, USA. [34]National Climate Center, Beijing, China. [35]School of Environment & Science, Griffith University, Nathan, Australia. [36]Soil Geography and Landscape Group, Wageningen University & Research, Wageningen, The Netherlands. [37]Department of Environmental Sciences, Wageningen University & Research, Wageningen, The Netherlands. [38]Environmental Systems Analysis Group, Wageningen University & Research, Wageningen, The Netherlands. [39]State Key Laboratory of Earth Surface Processes and Resource Ecology, Beijing Normal University, Beijing, China. [40]School of Life Sciences, Nanjing University, Nanjing, China. [41]Basque Centre for Climate Change bc3, Scientific Campus of the University of the Basque Country, Biscay, Spain. [42]Appalachian Laboratory, University of Maryland Center for Environmental Science, Frostburg, MD, USA. ✉e-mail: johan.rockstrom@pik-potsdam.de; steven.lade@futureearth.org

boundaries is required that accounts for Earth system resilience and human well-being in an integrated framework[4,5].

We propose a set of safe and just Earth system boundaries (ESBs) for climate, the biosphere, fresh water, nutrients and air pollution at global and subglobal scales. These domains were chosen for the following reasons. They span the major components of the Earth system (atmosphere, hydrosphere, geosphere, biosphere and cryosphere) and their interlinked processes (carbon, water and nutrient cycles), the 'global commons'[14] that underpin the planet's life-support systems and, thereby, human well-being on Earth; they have impacts on policy-relevant timescales; they are threatened by human activities; and they could affect Earth system stability and future development globally. Our proposed ESBs are based on existing scholarship, expert judgement and widely shared norms, such as Agenda 2030. They are meant as a transparent proposal for further debate and refinement by scholars and wider society.

First, we identify 'safe' boundaries at subglobal and global scales for "maintain[ing] and enhanc[ing] the stability and resilience of the Earth system over time, thereby safeguarding its functions and ability to support humans and all other living organisms"[4]. To determine safe boundaries, we use assessments of tipping point risks among local and regional tipping elements, evidence on declines in Earth system functions, analyses of historical variability and expert judgement. We assess the uncertainty in and confidence of these ESBs. Tipping elements are those components or processes that regulate the functioning and state of the planet and that show evidence of having thresholds at which small additional perturbations can trigger self-reinforcing changes that undermine Earth system resilience[15,16]. We do not exclusively rely on tipping points for setting safe ESBs, however, and the ESBs should not be interpreted as representing tipping points. As a reference state for human life support on Earth, we use an interglacial Holocene-like Earth system functioning dominated by balancing feedbacks that cope with, buffer and dampen disturbances. Methods and Supplementary Information have details on how safe boundaries are determined.

Second, we use three criteria to assess whether adhering to the safe ESBs could protect people from significant harm (Box 1): 'interspecies justice and Earth system stability' (I1)[17]; 'intergenerational justice'[18] between past and present generations (I2a) and present and future generations (I2b); and 'intragenerational justice' (I3) between countries[19], communities and individuals through an intersectional lens[20]. These criteria sit within a wider Earth system justice framework that goes beyond planetary and issue-related justice to take a multi-level transformative justice approach focusing on ends (boundaries and access levels) and means[21,22]. Methods and Supplementary Information have more detailed discussions of the justice approach applied in this paper. We define harm as negative impacts on humans, communities and countries from Earth system change in addition to background rates. The most recent Intergovernmental Panel on Climate Change (IPCC) report identifies 'severe' risks and 'high' reasons for concern when tens to hundreds of millions of people are exposed to changes in climate, such as increases in temperature and extreme events[23]. In this paper, we define significant harm as widespread severe existential or irreversible negative impacts on countries, communities and individuals from Earth system change, such as loss of lives, livelihoods or incomes; displacement; loss of food, water or nutritional security; and chronic disease, injury or malnutrition (a glossary is in the Supplementary Methods).

Third, we combine these justice criteria with historical analyses, international health standards, Earth system modelling and expert judgement to quantify safe and just ESBs that minimize human exposure to significant harm (no significant harm (NSH)) from Earth system change. Minimizing significant harm is a cornerstone of national and international law and corrective justice[24,25]. We focus on assessing the levels of Earth system change leading to

# Box 1

# The '3I' justice criteria used to analyse safe ESBs

Further explanation is in Gupta et al.[22]. Discussion of the caveats related to the justice approach applied in this paper is in Methods and Supplementary Information.

*Interspecies justice and Earth system stability (I1)*

Interspecies justice aims to protect humans, other species and ecosystems, rejecting human exceptionalism. In many domains, interspecies justice could be achieved by maintaining Earth system stability within safe ESBs.

*Intergenerational justice (I2a and I2b)*

Intergenerational justice examines relationships and obligations between generations, such as the legacy of greenhouse gas emissions or ecosystem destruction for youth and future people. Achieving intergenerational justice requires recognizing the potential long-term consequences of short-term actions and associated trade-offs and synergies across time. We define two types of intergenerational justice: (between past and present; I2a) whether actions of past generations have minimized significant harm to current generations and (between present and future; I2b) the responsibility of current generations to minimize significant harm to future generations.

*Intragenerational justice: between countries, communities and individuals (I3)*

Intragenerational justice includes relationships between present individuals, between states (international), among people of different states (global) and between community members or citizens (communitarian or nationalist). Intersectional justice considers multiple and overlapping social identities and categories (for example, gender, race, age, class and health) that underpin inequality, vulnerability and the capacity to respond. Achieving intragenerational justice means minimizing significant harm caused by one country to another, one community to another and one individual to another.

widespread exposure to significant harm, which will lead to greater impacts when vulnerable populations are exposed[3]. Methods and Supplementary Information have details on how just boundaries are determined. The just (NSH) boundaries described here are necessary but not sufficient conditions for Earth system justice, which must also enable access to resources for all[26] and distributional and procedural fairness[22]. A foundation that enables minimum access to water, food, energy and infrastructure for all humans alongside a safe and just (NSH) ESB ceiling of maximum allowed human pressure on biophysical domains could constitute a safe and just 'corridor' over time[4,22] (Fig. 1).

Our assessment builds upon and advances beyond previous research and science-based political consensus, such as the Planetary Boundaries (PBs) framework[27], doughnut economics[28] and the Sustainable Development Goals[29] in the following ways. (1) We define just ESBs for avoiding significant harm using the same units as the safe ESBs for the same domains and propose that actors use the stricter of the safe and just boundaries to inform target setting. The PBs identify only safe biophysical boundaries. The social goals related to access to or harm from natural resources adopted in Agenda 2030, doughnut economics and other approaches[28,30–32] are not quantified in comparable units or examine only the consequences of human activities on the Earth system, not

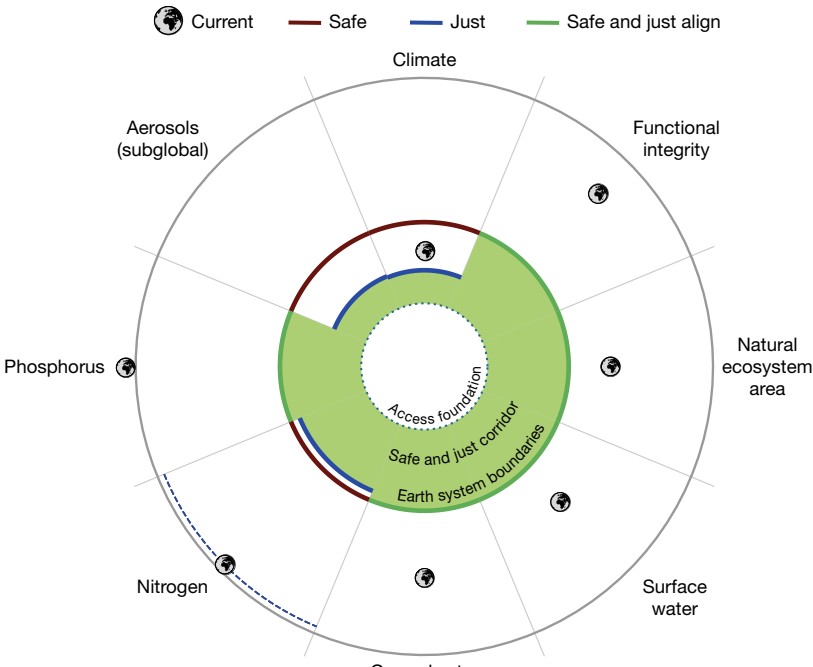

**Fig. 1 | Proposed safe and just (NSH) ESBs.** Visualization of safe ESBs (dark red), just (NSH) ESBs (blue), cases where safe and just (NSH) boundaries align (green) and current global states (Earth icons). Radial axes are normalized to safe ESBs. Headline or central estimate global boundaries (Table 1) are plotted to support comparison with the current global state, but we emphasize that we have also defined subglobal boundaries and multiple likelihood levels for many domains (Table 1). For aerosols, however, we display the subglobal boundaries to compare safe and just boundaries. For nitrogen, we plot with a dashed blue line the boundary quantification for harm from nitrate in groundwater while noting that the just boundary must also incorporate safe considerations via eutrophication, leading to a more stringent safe and just boundary. Minimum access to water, food, energy and infrastructure for all humans (dotted green line) could constitute the foundation of a safe and just 'corridor' (green filled area), but we do not quantify this foundation here. Alternative visualizations are presented in Extended Data Fig. 1.

harm to humans from Earth system change. Articulating sociopolitical notions, such as Earth system justice, and converting their implications into biophysical units can enable a better understanding of the space within which humans can function. (2) We define global and subglobal ESBs in most domains. The PBs' emphasis on the global scale can be inappropriate for the assessment and management of domains such as the biosphere[33] and fresh water[34–37]. (3) We set boundaries at multiple likelihood levels for Earth system states. (4) Tipping element assessments in climate, biosphere and other Earth system domains are key, although not exclusive, evidence for our ESBs. Recent PB assessments instead emphasize risks related to the departure from Holocene ranges of Earth system variability[38].

## Quantifying ESBs

For each Earth system domain, we first quantify safe boundaries for maintaining Earth system resilience, with multiple levels of likelihood reflecting uncertainty or variability in the exact position of the boundary. Adhering to these safe boundaries implements our 'interspecies justice and Earth system stability' criterion (I1 in Box 1) and will safeguard future generations against significant harm from Earth system change (intergenerational justice; I2b in Box 1), but it may not avoid significant harm to current generations, particularly vulnerable populations (I2a and I3 in Box 1). Hence, (1) we propose that some boundaries be made more stringent to protect present generations and ecosystems; (2) we complement safe boundaries with local-level standards to protect present generations and ecosystems; and (3) if the boundary is likely to cause considerable difficulties for present generations, we propose that it is complemented with policies that account for distributive justice. We also assess the current state of the Earth system with respect to each safe and just ESB.

## Climate

We identify safe ESBs for warming (Fig. 1 and Table 1) based on minimizing likelihoods of triggering climate tipping elements; maintaining biosphere and cryosphere functions; and accounting for Holocene (<0.5–1.0 °C) and previous interglacial (<1.5–2 °C) climate variability (Supplementary Methods). Some climate tipping points, such as circulation collapse or Amazon dieback, have high uncertainty or low confidence in their dynamics and potential warming thresholds[16], but the complementary palaeoclimate and biosphere analyses independently support the safe climate ESB assessment. Cryosphere function includes maintaining permafrost in the northern high latitudes, permanent polar ice sheets and mountain glaciers and minimizing sea ice loss. We find that global warming beyond 1.0 °C above pre-industrial levels, which has already been exceeded[9], carries a moderate likelihood of triggering tipping elements, such as the collapse of the Greenland ice sheet or localized abrupt thawing of the boreal permafrost[16]. One-degree Celsius global warming is consistent with the safe limit proposed in 1990[39] and the PB of 350 ppm $CO_2$ (ref. 27). Above 1.5 °C or 2.0 °C warming, the likelihood of triggering tipping points increases to high or very high, respectively (high confidence in Extended Data Table 1). Biosphere damage and the risk of global carbon sinks becoming carbon sources, potentially triggering further climate feedbacks, increase substantially[40]. We conclude that stabilizing at or below a safe ESB of 1.5 °C warming avoids the most severe climate impacts on humans and other species, reinforcing the 1.5 °C guardrail set in the Paris Agreement on Climate Change.

Assessment of significant harm from climate change suggests the need for a stricter just (NSH) boundary. At 1.0 °C global warming, tens of millions of people were exposed to wet bulb temperature extremes (Fig. 2), raising concerns of inter- and intragenerational justice. At 1.5 °C

## Table 1 | Proposed safe and just (NSH) ESBs (visualized in Fig. 1)

| Domain: state variable | Relevant Earth system change | Safe ESB subglobal (local/regional) | Safe ESB globally aggregated | Just (NSH) ESB | Safe and just ESB | Current global state |
|---|---|---|---|---|---|---|
| Climate: global mean surface temperature change since pre-industrial (1850–1900) | Climate tipping points; exceed interglacial range; biosphere functioning | Global climate boundary set to avoid regional tipping points and biome degradation | Likelihood of passing tipping points: low, 0.5–1.0 °C; moderate, >1.0 °C; high, >1.5 °C; very high, >2.0 °C | Exposure to additional significant harm: moderate, 0.5–1 °C; high, 1–1.5 °C; very high, >1.5 °C | 1.0 °C at high exposure to significant harm | 1.2 °C |
| Biosphere: natural ecosystem area | Loss of climate, water, biodiversity NCP | Critical natural ecosystems need to be preserved or restored | >50–60% natural ecosystem area (depending on spatial distribution) | Align with safe boundary plus ensure distributional justice | >50–60% (upper end) depending on distribution | 45–50% natural ecosystem area |
| Biosphere: functional integrity | Loss of multiple local NCP | >20–25% of each 1 km$^2$ under (semi-)natural vegetation; >50% in vulnerable landscapes; at <10%, few NCP remain | 100% of land area satisfies local boundary | Align with safe boundary | >20–25% of each 1 km$^2$ under (semi-)natural vegetation | One third (31–36%) of human-dominated land area satisfies ESB |
| Water: surface water flows | Collapse of freshwater ecosystems | <20% magnitude monthly surface flow alteration | 100% of land area satisfies local boundary (sums to 7,630 km$^3$ per year global flow alteration budget) | Align with safe plus World Health Organization and United Nations Environment Programme quality standards | Regional and global safe ESBs | 66% of global land area satisfies ESB annually (3,553 km$^3$ per year global alterations) |
| Water: groundwater levels | Collapse of groundwater-dependent ecosystems | Annual drawdown does not exceed average annual recharge | 100% of land area satisfies local boundary (sums to 15,800 km$^3$ per year global drawdown) | Align with safe plus World Health Organization and United Nations Environment Programme quality standards | Safe ESB (and ensure recovery) | 53% of global land area satisfies ESB (15,700 km$^3$ per year annual drawdown) |
| Green water[38] (previous assessment) | Not assessed | Monthly root-zone soil moisture deviates from Holocene variability | <10% of ice-free land area exceeds boundary | Not assessed | Not assessed | 18% |
| Nutrient cycles: nitrogen | Surface water and terrestrial ecosystem eutrophication | <2.5 (1–4) mg N l$^{-1}$ in surface water; <5–20 kg N ha$^{-1}$ per year in terrestrial ecosystems (biome dependent) | Surplus, <61 (35–84) Tg N per year; total input, <143 (87–189) Tg N per year | Align with local safe plus drinking water (<11.3 (10–11.3) mg NO$_3$–N l$^{-1}$; globally, <117 (111–117) Tg N per year) and any available air pollution (for example, NH$_3$) standards | Local ESBs; and global surplus, 57 (34–74) Tg N per year | Surplus, 119 Tg N per year; total input, 232 Tg N per year |
| Nutrient cycles: phosphorus | Surface water eutrophication | <50–100 mg P per m$^3$ | Surplus, <4.5–9 Tg P per year; mined input, <16 (8–17) Tg P per year | Align with local safe boundary to avoid eutrophication | Local and global safe ESBs | Surplus, ~10 Tg P per year; mined input, ~17 Tg P per year |
| Atmosphere: aerosol loading | Monsoon systems | <0.25–0.50 AOD | Annual mean interhemispheric AOD difference: <0.15 | Align with safe plus <15 µg per m$^3$ mean annual PM$_{2.5}$; other levels of exposure to significant harm in Supplementary Table 11 | <15 µg per m$^3$ PM$_{2.5}$ plus regional and global safe ESBs | 0.05 annual mean interhemispheric AOD difference |

warming, more than 200 million people, disproportionately those already vulnerable, poor and marginalized (intragenerational injustice), could be exposed to unprecedented mean annual temperatures[41], and more than 500 million could be exposed to long-term sea-level rise (Fig. 2 and Methods). These numbers of people harmed vastly exceed the widely accepted 'leave no one behind' principle[29] and undermine most of the Sustainable Development Goals. Moreover, past emissions have already led to significant harm, including extreme weather events, loss of habitat by Indigenous communities in the Arctic, loss of land area by low-lying states and sea-level rise or reduced groundwater recharge from changing glacial melt systems[3]. Irreversible impacts from cryosphere and biosphere tipping elements that are committed by anthropogenic greenhouse gas emissions in the coming decades but which unfold over centuries or millennia also threaten intergenerational justice (Supplementary Methods). We conclude that if exposure of tens of millions of people to significant harm is to be avoided, the just (NSH) boundary should be set at or below 1.0 °C. Since returning within this boundary may not be achievable in the foreseeable future, adaptations and compensations to reduce sensitivity to harm and

vulnerability will be necessary. During the 2022 United Nations Climate Change Conference (COP-27), developing countries indeed focused actively on issues of adaptation, loss and damage.

**Biosphere**

For the biosphere, we identify safe ESBs for two complementary measures of biodiversity: (1) the area of largely intact natural ecosystems and (2) the functional integrity of all ecosystems, including urban and agricultural ecosystems (Table 1). Maintaining areas of largely intact natural ecosystems is necessary for securing the Earth system functions on which all humans, other species (I1 in Box 1) and Earth system stability depend, including stocks and flows of carbon, water and nutrients and halting species extinction (Earth system nature's contribution to people (NCP) via Earth system functions). Based on climate, water and species conservation model outcomes, we propose a safe ESB of 50–60% (medium confidence in Extended Data Table 1) of global land surface covered by largely intact natural areas to maintain Earth system NCP (Table 1 and Supplementary Methods). This range uses the current area of natural land cover as a minimum value while indicating

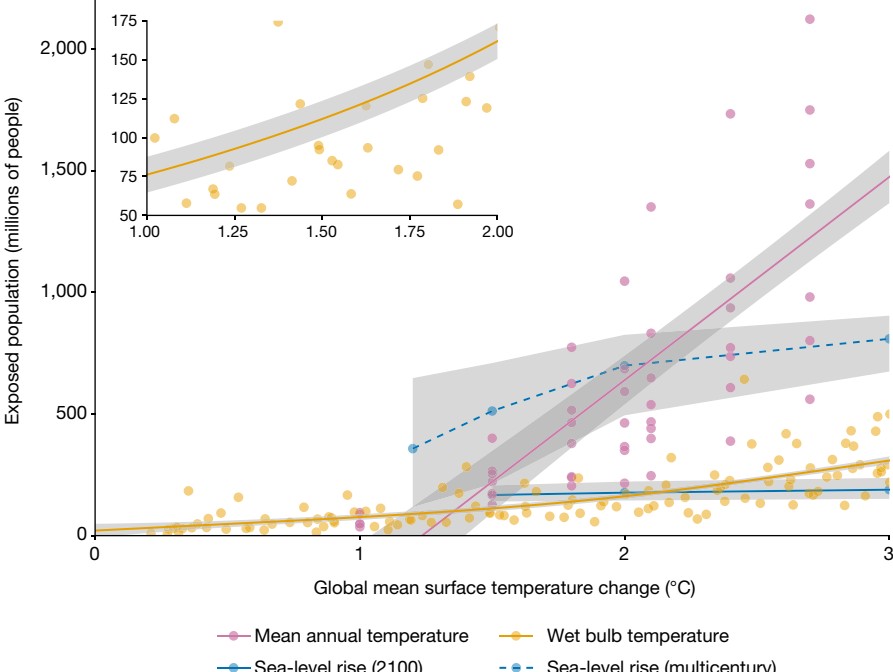

**Fig. 2 | Exposure to significant harm from climate change at different levels of warming.** We examine the exposure of the 2010 global population to mean annual temperatures above 29 °C (purple; linear fit, $P < 0.01$), wet bulb temperatures of 35 °C for an average of at least 1 day per year (orange; quadratic fit, $P < 0.01$) and future sea-level rise (blue; linear interpolation). Sea-level rise is calculated for 2100 (blue solid) and multi-centennial (blue dashed; linear interpolation) responses to a given temperature stabilization by 2100, representing near-term impacts and long-term equilibria, respectively. The inset shows the magnification of wet bulb temperature in the range 1–2 °C. Shading indicates one s.e.

the need to restore largely intact natural areas. The exact safe boundary depends strongly on the demand for specific ecological functions (which in turn depend, for example, on the remaining carbon emissions to be sequestered) and on the spatial distribution of the largely intact natural area across ecoregions and ecosystems. Studies generally indicate that up to 60% of the terrestrial earth surface area may be needed, with some extending up to 80% (Supplementary Methods). Natural ecosystem areas comparable with the 50–60% terrestrial ESB are needed in the ocean to maintain carbon sequestration and minimize additional marine species extinction[42]. Biome-scale boundaries may be more stringent: for example, to protect tropical forest biomes due to their contribution to climate stability and moisture recycling. If allocation and coordination of restoration efforts are less than optimal, the required minimum area will be larger. If these boundaries are transgressed, tipping points involving loss of biome-scale functional integrity and associated NCP may be triggered, including increases in species extinction rates.

Adherence to our proposed safe ESB for the area of largely intact natural ecosystems should minimize harm to future generations (I2b in Box 1) by securing biosphere contributions to all life support through a stable and resilient Earth system and localized NCP provided by largely intact nature. However, achieving justice for current generations (I2a and I3 in Box 1) may require a stricter boundary because the safe ESB does not account for the current uneven distribution of largely intact natural ecosystems needed to support local livelihoods[43], especially in poor or Indigenous communities[44,45]. Some people and countries may directly benefit from policies to maintain or increase natural ecosystem area[46], while others may face opportunity costs[47]. Hence, to ensure just distribution of largely intact natural ecosystems, a just (NSH) boundary may need to be set at the upper end of the 50–60% safe range, as allocation will be less than optimal for achieving the functions the lower boundary was optimized for. We emphasize that natural ecosystem area includes all largely intact natural areas and not only those currently

requiring conservation attention; it does not imply protection that excludes human habitation and sustainable use.

Functional integrity is the capacity of urban, agricultural or other human-modified ecosystems to provide ecological functions and their contributions to people at landscape scale, complementing the Earth system NCP provided by large-scale intact natural ecosystem areas. We analyse what minimum amount, quality and distance of natural habitat and seminatural habitat are needed to maintain local terrestrial NCP provision, including pollination, pest and disease control, water quality regulation, soil protection, natural hazards mitigation and recreation. We identify that at least 20–25% diverse seminatural habitat including native species in each square kilometre in human-modified lands is needed to support the provisioning of multiple local NCP[48]. The exact amount and quality required differ based on landscape type, climate and topography; the amount can range up to 50% in some landscapes vulnerable to natural hazards, such as steep slopes or highly erodible soils. This boundary applies to fine scales, currently proposed as 1 km², because NCP are not transferable (for example, erosion or landslide can only be avoided by natural cover on the same slope) and are often provided or supported by non-mobile or limited mobility species (for example, foraging ranges of pollinating or pest-regulating insects are limited to a few hundred metres). About two thirds of human-dominated land area (approximately 40% of total land area) has insufficient functional integrity (Supplementary Methods), and large areas are showing symptoms of resilience loss[49], requiring regenerative practices to restore local and Earth system functions.

The safe boundary for functional integrity reduces future exposure to significant harm (intergenerational justice). Loss of functional integrity in agricultural ecosystems and cities below the safe boundary would reduce food productivity, ecosystem capacity to mitigate natural hazards, pollution and nutrient losses and increase reliance on harmful pesticides and biocides and capacity to choose alternate land uses (intragenerational justice). The dependence on these services is

often higher in regions with more vulnerable communities. Specific interventions that secure functional integrity are highly local and are best implemented under local authority, knowledge and leadership[50], with policy interventions often needed to ensure that marginalized groups are not further disempowered but are given the space to use their knowledge and approaches to participate in such processes[51].

## Water

For fresh water, we propose two spatially defined safe ESBs based on subglobal boundaries that can be aggregated to the global scale: (1) a flow alteration ESB for surface water and (2) a drawdown ESB for groundwater (Table 1). Flow alteration in rivers is one of the key drivers of freshwater biodiversity loss[52], leading to declines in freshwater biodiversity that outpace those of terrestrial and marine systems[53] and in large-scale NCP, such as coastal and inland fisheries, on which millions of people depend[54,55]. Local-scale flow-ecology analyses are often used to establish environmental flow needs to define safe levels of flow alteration for individual watersheds[56]. These local-scale assessments could provide the basis for spatially explicit safe boundaries but are absent across most of the world[57]. In their absence, we propose that a presumptive subglobal safe ESB of 20% alteration (increase or decrease) of monthly surface water flows compared with the prevailing natural flow regime be met in all rivers globally (medium confidence in Extended Data Table 1). This ESB leaves 80% of flows unaltered to meet environmental needs[58,59], assuming that required water quality standards are also met. The ESB is supported by empirical studies showing that flow alterations within 20% support native fish species and flow alteration beyond this level strongly affects biodiversity and ecosystem structure and function[60,61] (Supplementary Methods has additional references supporting the use of this threshold). The global ESB for surface water is that 100% of all land area meets the subglobal boundary by limiting alterations of flows by 20% in all rivers in the world. Meeting the global ESB sums to a global alteration budget of 7,630 km[3] per year (Supplementary Methods; with high confidence in Extended Data Table 1). Globally aggregated river flow alterations are currently less than this figure; however, we are outside the global ESB because the subglobal safe ESB is only met for 66% of land area (Table 1) and less than half of the global population (Supplementary Methods). These results are consistent with recent analyses of water scarcity, which highlight the challenge of meeting environmental flow requirements to support ecosystem services, such as fisheries production, while ensuring there is sufficient water for human needs[57,62].

Groundwater aquifers contribute to base flows in many river systems and directly sustain wetlands and terrestrial vegetation. Unsafe levels of groundwater extraction occur when drawdown exceeds replenishment rates, impacting groundwater-dependent ecosystems and in some instances, leading to land subsidence and irreversible aquifer loss[12,63,64]. Given the temporal nature of groundwater recharge and discharge and a lack of widespread consistent data on historical aquifer levels, we propose that the safe ESB for annual groundwater drawdown for all aquifers be the average annual recharge, with groundwater considered safe if drawdown is less than recharge. The subglobal safe ESB is met for a given aquifer when local drawdown does not exceed average annual recharge. The global ESB for groundwater is that the subglobal ESB is met for all aquifers around the world. For the 2003–2016 period, the global sum of average annual recharge is approximately 16,000 km[3] per year (Table 1 and Supplementary Methods; with high confidence in Extended Data Table 1). The groundwater extraction that may safely occur within this boundary naturally varies across the planet and, where possible, should be defined based on local-scale monitoring, although broad trends can also be determined via satellite remote sensing[65]. We estimate that we are currently outside the global ESB because groundwater levels in 47% of basins are currently in decline (Table 1).

Our justice analysis of the safe ESBs for surface and groundwater highlights the challenges of (1) multi-level distribution, (2) water insecurity and (3) water quality. The regional surface and groundwater ESBs are generally in the long-term interests of surrounding communities, as they conserve future fresh water (intergenerational justice: I2b in Box 1). Where depleted aquifers have already caused significant environmental impacts[66], groundwater extraction should urgently be reduced, and recharge areas should be protected to restore aquifers to safe levels (NSH to present generations: I2a and I3 in Box 1). Minimizing significant harm to current generations also requires the following. (1) Accounting for multi-level distribution indicates the allocation of allowed alterations between communities, sectors or nations sharing the water body, whether directly or indirectly via virtual water. This allocation is particularly challenging where the safe ESB requires drastic reductions in water use. (2) Minimizing exposure to significant harm should account for water insecurity in different regions of the world. For example, harm associated with poor water sanitation and hygiene conditions disproportionately impacts the health of young children in low-income countries[67], particularly in Sub-Saharan Africa and South Asia[68]. (3) Minimizing exposure to significant harm implies addressing surface water quality guidelines for human use[69], not just an allocation of water quantity. At a minimum, water needs to be safe for consumption and irrigation, meaning that acceptable standards for faecal coliforms and salinity must be met. We align our just (NSH) ESBs for water with the safe ESBs while noting that adhering to the boundaries would considerably restrict current use and will require policies to ensure distributive justice.

These proposed surface and groundwater ESBs are independent of green water stocks. Green water stocks are critical for maintaining the atmospheric water cycle, which regulates seasonal precipitation levels[34]; can support a significant proportion of global agricultural production[70] with less impact on aquatic ecosystems than blue water use[71]; and are closely related to the biosphere ESBs. A recent assessment[38] proposed a spatially explicit green water boundary to ensure hydrological regulation of terrestrial ecosystems, climate and biogeochemical processes by defining a maximum allowed deviation (drying or wetting) of soil moisture levels from mid-Holocene conditions. The state variable for green water is defined as the percentage of ice-free land area that in any month has root-zone soil moisture levels outside the 95th percentile of the local baseline variability. The boundary value is set at 10%, corresponding to the median departure level from mid-Holocene conditions. We include this green water boundary in our set of safe ESBs (Table 1), but we limit our inter- and intragenerational justice analysis (I2 and I3 in Box 1) to surface and ground blue water.

## Nutrients

We set safe ESBs for agricultural nitrogen (N) and phosphorus (P) surpluses for minimizing eutrophication of surface water and terrestrial ecosystems due to runoff, leaching and atmospheric N deposition via ammonia and nitrogen oxide emissions (Table 1). We propose safe global-scale ESBs of 61 (35–84) Tg N per year for agricultural nitrogen surplus[72] and 4.5–9.0 Tg P per year for cropland soil phosphorus surplus[73,74] (medium confidence in Extended Data Table 1). These ESBs are based on recent papers[72,74] calculating subglobal and global agricultural nutrient losses, surpluses and inputs from critical N and P concentrations in water and air beyond which eutrophication occurs (Methods, Table 1 and Supplementary Methods). These ESBs primarily relate to agriculture, which accounts for approximately 90% of anthropogenic N/P inputs to the Earth system[72,75]. Our ESBs are based on agricultural surpluses and losses[72,74], although for comparison with previous PB quantifications (Supplementary Methods), we also provide corresponding global inputs assuming current N/P use efficiency. These recent studies also account for non-agricultural sources, assuming they remain at current levels, and the redistribution of nutrients from over-fertilized to under-fertilized regions (Supplementary Methods).

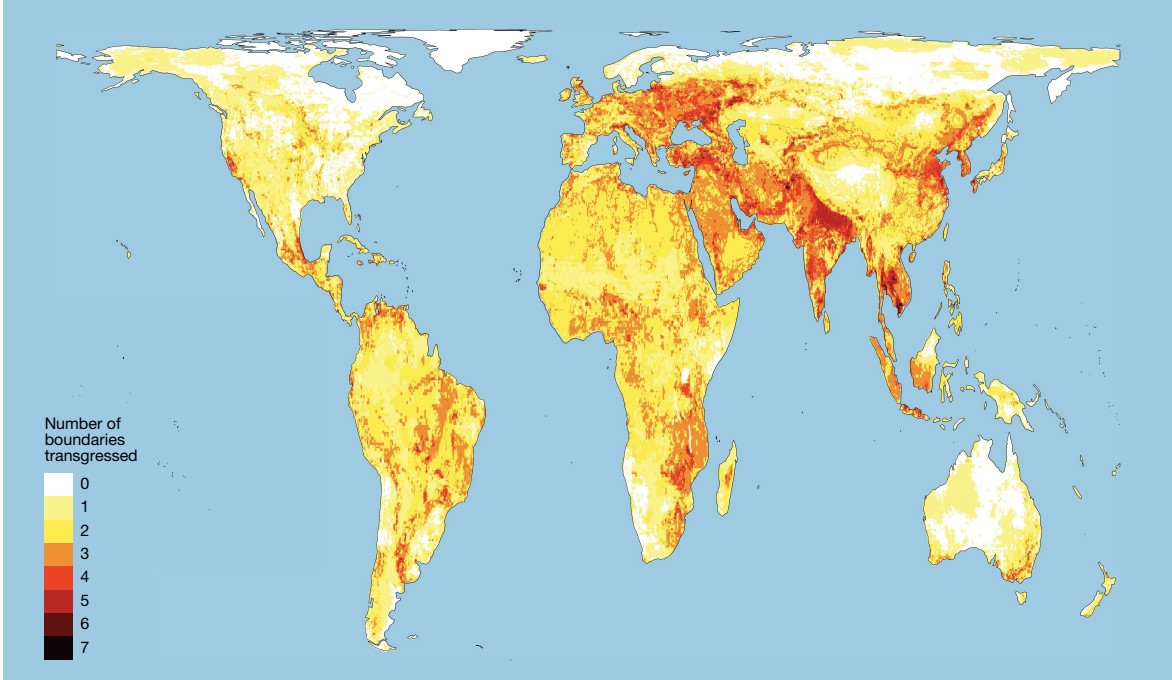

**Fig. 3 | Hotspots of current ESB transgressions.** The number of subglobal climate (two local exposure boundaries), functional integrity, surface water, groundwater, nitrogen, phosphorus and aerosol safe and just ESBs currently transgressed by location. No more than seven of these eight metrics have their ESBs transgressed in any one pixel. Since climate is a globally defined ESB, we use wet bulb temperatures of over 35 °C for at least 1 day per year and low-elevation coastal zones (<5 m) exposed to sea-level rise as proxies for local climate transgression while acknowledging that the impacts of climate change are far more diverse. We also emphasize that exposure of a location does not necessarily imply responsibility for causing or addressing these environmental impacts. We invite the reader to investigate the consequences of different boundary values using the code in the code availability information.

Elevated N and P concentrations cause harm through the consequences of eutrophication on ecosystems and their services, such as fishery collapse, toxic compounds released by algal blooms[72,76] and the health impacts of air pollution from ammonia-derived aerosols[77]. Harm can also occur from drinking surface or groundwater with elevated nitrate concentrations[78] but at a higher level than the safe N concentration for surface water eutrophication. We therefore align the just (NSH) ESBs for subglobal N and subglobal and global P with their safe boundaries, as human harm from nutrient cycle disruption is primarily driven by environmental degradation. Accounting for significant harm from groundwater nitrate tightens the global N boundary slightly to 57 (34–74) Tg N per year (Supplementary Methods). These ESBs should be complemented by standards for local air and water pollution for N and water pollution for P. Additional justice considerations include lack of access to N and P fertilizers, which can threaten food security especially for low-income communities and countries[76], and extraction of phosphate rock, which is a limited resource currently underpinning food production but exposes poor and marginalized communities to mining waste, destroyed land and human rights abuses[76,79].

## Aerosol pollution

For aerosols, we propose a safe ESB defined by the interhemispheric difference in aerosol optical depth (AOD) (Table 1) based on evidence that a rising North/South Hemisphere difference can trigger regional-scale tipping points and cause substantial adverse effects on regional hydrological cycles, in addition to the existing PB of 0.25–0.50 AOD based on regional considerations[27]. We consider AOD differences and their potential impacts arising from natural emissions, anthropogenic emissions and stratospheric aerosol injection (solar geoengineering). Observational data for the West African monsoon rainfall[80] and climate modelling studies for the Indian monsoon[81] have identified potential shifts in the location of the Intertropical Convergence Zone triggered by differences in sulfate AOD between the Northern and Southern Hemispheres[81]. Observational studies on the impacts of interhemispheric AOD difference on the Indian monsoon are lacking, but observations based on past volcanic eruptions and climate modelling studies show that an increased concentration of reflecting aerosols in one hemisphere leads to precipitation decreasing in the same hemisphere's tropical monsoon regions while increasing in the opposite hemisphere[80,82,83]. Observed changes in the South Asian monsoon have well-understood mechanisms (Supplementary Information) that are consistent with the effects of interhemispheric AOD difference[84]. The volcanic eruptions of El Chichon in the 1980s (AOD difference of 0.07) and Katmai (AOD difference of 0.08) provide empirical examples[80], while model-simulated AOD differences of 0.1 and approximately 0.2 lead to declining precipitation in tropical monsoon regions[85]. Interhemispheric AOD difference and its impact on shifts in tropical precipitation are sensitive to the aerosol particle size and the latitudinal and altitudinal distribution of reflecting aerosols[86]. Considering this and the range of these studies (approximately 0.05–0.20 of additional AOD difference), we assess that these shifts may become disruptive if the interhemispheric AOD difference, currently approximately 0.05[87] on average and approximately 0.1 in the boreal spring and summer[87], exceeds 0.15 (low confidence in Extended Data Table 1) due to air pollution[85] or geoengineering-related aerosol asymmetries[81,85] (Supplementary Methods).

Significant harm to human health from exposure to aerosols, such as particulate matter (PM), suggests a more stringent just (NSH) boundary based on local air pollution standards[88]. PM and other aerosols are associated with respiratory illnesses and premature deaths as well as heart problems and debilitating asthma[89]. We select a just (NSH) boundary of 15 μg per m$^3$ mean annual exposure to PM$_{2.5}$ to avoid a high likelihood of significant harm from aerosols (Table 1 and Supporting Information) based on World Health Organization

2021[88] guidelines (Table 1) and European Union and US Environmental Protection Agency air quality standards[90,91]. Such local and regional guidance is needed because $PM_{2.5}$ characteristics, such as toxicity, are highly place and source specific. Eighty-five percent of the world population is currently exposed to $PM_{2.5}$ concentrations beyond this boundary[92], and exposure to ambient $PM_{2.5}$ is estimated to cause 4.2 million deaths annually[89], with vulnerable groups being affected disproportionately more while polluting less[93]. Air pollution scenarios based on globally successful stringent mitigation and pollution control show reductions in affected populations, but areas of high air pollution might remain[94]. A 15 µg per m³ $PM_{2.5}$ concentration translates[95,96] to an AOD of approximately 0.17, indicating that the just (NSH) boundary for aerosols is more stringent than the safe regional boundary (0.25–0.50) (Table 1).

## Novel entities and other pollutants

We acknowledge the risks to Earth system stability and human well-being from other air and water pollutants, for which there are already well-accepted guidelines[88], and the emerging threats from novel entities, new forms of existing substances and modified life forms that are geologically or evolutionarily novel and could have large-scale unwanted geophysical or biological impacts on the Earth system[27,97]. Evidence on the diverse risk potentials of novel entities, such as microplastics, 'forever chemicals', antibiotics, radioactive waste, heavy metals or other emerging contaminants, for Earth system function and human health and food security is increasing, but knowledge gaps on the scale and scope of potential impacts remain[98]. Persson et al.[97] reported that humanity has crossed the PB for novel entities, although data limitations and quantification are challenging even for the known novel entities. The differentiated impacts of novel entities already witnessed today across different populations and the long lifetimes of these substances raise clear intragenerational and intergenerational justice concerns[97,98].

## Current state

Seven of the eight global-scale safe and just ESBs that we quantified have already been crossed (Fig. 1 and Table 1). Transgression of ESBs is spatially widespread, with two or more safe and just ESBs transgressed for 52% of the world's land surface, affecting 86% of the global population (Fig. 3). Some communities experience many ESB transgressions, with four or more ESBs transgressed for 28% of global population but only 5% of global land surface (Fig. 3). Spatial hotspot transgressions are therefore concentrated in regions of higher population density, raising major intragenerational justice concerns.

## Toward a safe and just future

We defined and quantified safe and just (NSH) ESBs for sustaining the global commons that regulate the state of the planet, protect other species, generate NCP, reduce significant harm to humans and support inclusive human development (Fig. 1 and Table 1). Because exceeding safe boundaries results in widespread significant harm, our just and safe ESBs align for surface water, groundwater, functional integrity, natural ecosystem area, phosphorus and nitrogen. Meeting these boundaries without transformation, however, could significantly harm current generations. In two cases, aerosols and climate, the just boundaries are more stringent than the safe boundaries, which indicates that people experience significant harm before that Earth system domain is destabilized.

We identified subglobal ESBs, which, in many domains, are the relevant scale for action to avoid loss of Earth system stability and minimize exposure to significant harm, and global ESBs, which are reference points for monitoring human impacts at the Earth system scale. Nations, cities, businesses and other key actors need to set and achieve science-based targets for reducing their environmental impacts based

on translation of the safe and just ESBs to actor fair shares[99]. Climate is the only ESB that has a relatively well-established and implemented methodology[100,101], with methodologies for other domains under development[101,102]. We emphasize that our ESBs complement, not over-ride, environmental restrictions for specific local settings: for example, stricter biosphere boundaries for carbon-dense ecosystems or targeted conservation efforts for protecting endangered or emblematic species. We also acknowledge that other actors may choose to implement targets based on other likelihood levels than those we have highlighted (Fig. 1 and Table 1): for example, a lower risk tolerance than the high risk of passing tipping points associated with a 1.5 °C safe boundary.

We offer our ESBs as an integration of social and natural sciences for further refinement, in the spirit that the PBs were proposed over a decade ago[103]. Seven of the eight globally quantified ESBs have been crossed and at least two local ESBs in much of the world have been crossed, putting human livelihoods for current and future generations at risk. Nothing less than a just global transformation across all ESBs is required to ensure human well-being. Such transformations must be systemic across energy, food, urban and other sectors, addressing the economic, technological, political and other drivers of Earth system change, and ensure access for the poor through reductions and reallocation of resource use. All evidence suggests this will not be a linear journey; it requires a leap in our understanding of how justice, economics, technology and global cooperation can be furthered in the service of a safe and just future.

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

## Methods

This work is an output of the Earth Commission, an independent international scientific assessment initiative hosted by Future Earth (https://earthcommission.org/). The synthesis presented here builds on recent work of the Earth Commission; other scientific literature, such as the PBs; intergovernmental reports, such as those of the IPCC; and World Health Organization guidelines. As the science component of the Global Commons Alliance (https://globalcommonsalliance.org/), the Earth Commission's theory of change includes providing our results on ESBs to the Science-Based Targets Network, the Systems Change Lab and Earth HQ.

While we acknowledge that any scientific assessment will involve some subjectivity, we have taken several steps to ensure the scientific rigour of our ESBs. (1) Our analysis is founded on a rigorous evidence base (Safe ESBs and Supplementary Methods). (2) Where possible, we determine ESBs at multiple likelihood levels (for climate change, 0.5 °C for low likelihood of passing climate tipping points, 1 °C for moderate likelihood and so on) (Table 1). (3) The nomination process for the Earth Commission and its working groups was an independent process managed by Future Earth (Ethics and inclusion statement). (4) We report the confidence in our ESB assessments (Safe ESBs and Extended Data Table 1).

### Safe ESBs

We used two main groups of approaches to setting safe ESBs: a 'multiple elements' approach and a 'spatial aggregation' approach. We describe these methods here in general terms, with technical details available in Supplementary Methods. These boundaries are aimed at protecting Earth system stability and life-support systems for as many species as possible, but they may not protect all species or all humans today, as further elaborated in our justice analysis.

For climate and biosphere, we assessed critical thresholds for a range of 'elements' relevant to each Earth system domain through literature review and modelling.

- For climate, we based our data on those found in a recent assessment of climate tipping elements[16] combined with evidence on biosphere and cryosphere function and palaeoclimate variability (Supplementary Methods).
- For functional integrity, we synthesized the literature on the area needed to secure local NCP, including pollination, pest and disease control, water quality regulation, soil protection, natural hazards mitigation, and physical and psychological experiences (Supplementary Methods).
- For natural ecosystem area, we examined the Earth system NCP of carbon stocks, water flows and habitat for avoiding species extinction (Supplementary Methods).

From these sets of thresholds, we determined boundaries that avoid triggering climate tipping elements or maintain multiple local or Earth system NCPs at different levels of likelihood. To set the climate boundaries, we also used temperature ranges of previous Quaternary interglacials and temperature ranges that maintain biosphere and cryosphere functioning (Supplementary Methods).

For water and nutrients, we identified subglobal boundaries relevant to these systems and then converted them into global boundaries using models or simple aggregation.

(1) For surface water flows, we used an emerging consensus in the literature to set boundaries on the alterations (increase or decrease) to local-scale surface water flows that protect freshwater ecosystems and fisheries (Supplementary Methods) and applied this to the global land surface area. While the safe alterations can be summed to a global alteration budget, to ensure aquatic ecosystem protection, the safe ESB is best implemented and interpreted according to the subglobal boundary. To derive the safe levels of monthly flow alteration volumes for all land area globally, we analysed water balance model (WBM) runs coupled with the TerraClimate dataset of monthly climate forcings (Supplementary Methods has further information).

(2) For groundwater, our approach is based on preventing declines in local aquifer levels by setting the maximum safe average annual drawdown equal to the average annual recharge (Supplementary Methods). We estimated the annual groundwater recharge and drawdown for all land surface areas using Gravity Recovery and Climate Experiment satellite data covering the period from 2003 to 2016 coupled with data from the Global Land Data Assimilation National Oceanic and Atmospheric Administration Land Surface Model L4 v.2.1 (Supplementary Methods has more detailed information).

(3) For nitrogen, we used three regional environmental boundaries: significant disruption to freshwater ecosystems (from total N runoff), groundwater potability (from nitrate leaching) and terrestrial ecosystems (from atmospheric N deposition due to ammonia and nitrogen oxide emissions) across wide areas based on critical concentration limits for each. We mainly relied on a recent study[72] following up previous works[74,104,105] that extended the approach of the original PBs[27,103]. This study used the Integrated Model to Assess the Global Environment (IMAGE) model to derive subglobal boundaries for critical nitrogen losses, surpluses and inputs based on critical concentrations in air and water and then aggregated these into global boundaries (Supplementary Methods has further information).

(4) For phosphorus, we relied on recent work that used literature-derived critical concentrations for avoiding eutrophication from P runoff to estimate global boundaries for P mined input and surplus based on a global budget calculation, taking into account P recycling, human excreta, soil and sediment retention, and global nutrient rebalancing[74,106].

Our approach for the safe aerosol boundaries does not fit neatly into these two categories because we used different methods for the subglobal and global boundaries. Our subglobal safe boundary uses the PB assessment of AODs that avoid tipping of regional monsoon systems. Our global assessment uses recent literature on the consequences of interhemispheric differences in aerosol concentrations on the global monsoon system (Quantifying ESBs and Supplementary Methods have further information).

As a reference for a 'safe' Earth climate system state, we used the interglacial Holocene epoch (that is, the state of the Earth system since the last Ice Age some 11,700 years ago[107,108]. The Holocene's exceptionally stable global climate system (oscillating <0.5–1 °C from the global pre-industrial 14 °C mean surface temperature)[107] and its configurations of global hydrology, primary production of biomass, biogeochemical cycling and Earth system NCP were the fundamental prerequisites for human development as we know it[7]. We argue that only within a Holocene-like interglacial climate can Earth continue to support human well-being, subject to consumption behaviours and population size. There is no evidence that billions of humans and complex societies can thrive in other known climates, such as a glacial ice age or 'Hothouse Earth'[7].

We identified boundaries at multiple levels of likelihoods to reflect underlying scientific uncertainties and variabilities. These uncertainties included epistemic uncertainty in the boundary value for a specific Earth system process or component, such as a tipping element; variability in a boundary value across different places; and uncertainty when aggregating multiple subglobal boundaries into a global boundary. In some cases, these levels are presented with qualitative descriptors of each likelihood level; in other cases, they are presented as a central estimate with an uncertainty range, depending on the available evidence.

Some of our boundary quantifications use assessments of tipping elements since triggering tipping can endanger Earth system stability.

Tipping elements commonly undergo changes that are abrupt (that is, faster than the forcing), large and difficult to reverse[109], although a particular tipping element may not display all three characteristics simultaneously (for example, table 4.10 in ref. 9). We identified boundaries based on tipping elements that accelerate or lock in change in the same Earth system component or process, such as climate tipping accelerating further climate change or triggering the inevitable loss of an ice sheet, or that trigger a tipping element in another Earth system domain, such as phosphorus concentration reaching a level that triggers eutrophication and disruption of freshwater ecosystems (Table 1).

## Safe ESBs: confidence levels

We also assessed the levels of confidence in our safe boundaries (Extended Data Table 1). 'Confidence' in this context can be read as 'degree of certainty in' or 'confidence in the validity of' a specific ESB quantification. We use the same scheme for assessing and communicating confidence as the IPCC[110,111], which sets out two components: (1) robustness of the evidence base, judged as limited, medium or robust, considering its type, amount, quality and consistency and (2) degree of scientific agreement across the peer-reviewed literature and among the members of each Earth Commission Working Group, judged as low, medium or high. Based on these two dimensions, five qualifiers can be used to express the level of confidence in a particular ESB quantification: very low, low, medium, high and very high. This self-assessment is an expert judgement based on our understanding of the available literature.

## Just (NSH) ESBs

We adopt an Earth system justice lens[22] for both intrinsic and instrumental reasons. We show that some safe ESBs are not strong enough to protect humans and other species today and that we cannot achieve and live within the safe ESBs if inequality is high and resources are unjustly distributed. The evidence from behavioural experiments in public goods provision shows that perceptions of fairness significantly alter the outcomes of such experiments. In particular, individuals in disadvantageous positions insist on fairness even at the risk of large losses by doing so; such experiments suggest that climate change mitigation may not be achieved if rich countries are not perceived as pulling their weight[112,113]. In common pool resource experiments, rising income inequality leads to a downward spiral of resource overexploitation and scarcity[114]. In such experiments, viewing the problem in terms of fairness can lead to norms that motivate restraining from harvesting[115]. A justice analysis is all the more needed as all science emerges from the value systems that apply in that domain, although these are often not made transparent.

Within the context of our Earth system justice approach[22], we use three justice criteria or the '3Is': interspecies justice and Earth system stability (I1)[17], intergenerational justice[18] (I2) and intragenerational justice (I3). Our research into interspecies and multispecies justice reveals details regarding the scholarly approaches to these concepts, but there have been no attempts to operationalize these concepts deductively. In our research, we have combined interspecies justice with Earth system stability because Earth system instability undermines non-human species and inductively identified, through domain-specific (for example, climate, biosphere and aerosol loading) approaches, boundaries based on existing scholarship and the logic of that domain. Intergenerational justice refers to the justice between past and present generations (I2a) and between present and future generations (I2b). In general, although not always, our ESBs meet the I2b criteria because they protect future generations but not the present (I2a). Intragenerational justice (I3) combines justice between countries[19], communities and individuals through an intersectional lens[20]. In balancing between the different justice criteria, we recognize that protecting future generations may impose many trade-offs with the use of resources today and that promoting intragenerational justice will also raise difficult issues regarding how to share resources, risks and responsibilities.

Our concept of harm derives from the justice literature and connects to the terms impact and risk used in the assessment literature. For example, IPCC defines[116] risk as the potential for adverse consequences for human or ecological systems, including to lives; livelihoods; health and well-being; economic, social and cultural assets; infrastructure; services; and ecosystems. These risks are a result of exposure (the presence of people or other assets in regions of Earth system change or hazards, such as populations living near sea level) and of vulnerability (the propensity or disposition to be adversely affected, such as the poor who live in precarious homes or health status). Impact is defined by IPCC as realized risk or consequences. Our harm estimates are mostly based on exposure at different levels of Earth system change.

We recognize four caveats in the justice approach applied in this paper. (1) While staying within the just boundaries as set in this paper is crucial to avoid harm to significant sections of the human population, they are by no means guaranteeing just outcomes. Since just ends can be achieved with unjust means, meeting these boundaries without transformation could significantly harm current generations. (2) While harm to humans is caused in part by increased exposure to biophysical changes, we recognize that harm is also a function of people's social–economic vulnerability and lack of adaptive capacities. This is beyond the scope of the present paper. (3) Our high levels of aggregation preclude systematic analysis of distributional justice issues in terms of which social subgroups are most harmed under what scenarios. (4) We do not explicitly address possible trade-offs between the three justice criteria. For example, policy instruments for achieving 'I1' may well undermine 'I3' (for example, limit access to resources for marginal people). Hence, we call for redistribution, liability and compensation.

Each safe ESB has been dealt with slightly differently, with some domains looking at when the system crosses tipping points (for example, climate change), others arguing that tipping points were crossed in the past and trying to recreate boundaries that allow species and systems to function (for example, surface water) and still others taking existing constraints into account in doing so (for example, groundwater). Although the proposals from a safe (and I1) approach fulfil I2b in that they makes space for future generations of humans, they may not guarantee safety for humans today (I2a; for example, climate change; hence, we call for more stringent targets), do not address local human exposure to pollutants (for example, air pollution; hence, we complement with local standards) or may limit access to resources (hence, calling for redistribution[26], liability, compensation and so on). Finally, while I2a has an explicit temporal dimension, intragenerational justice has an explicit spatial dimension and focuses on whether all people have access to minimum resources and services[26]; how scarce resources are divided or shared between countries, communities and people and the varied justice issues that arise per domain; how environmental risks are spread worldwide and who is most exposed (through, for example, mapping exposure and vulnerability) and how responsibilities are shared between different actors.

To calculate the population exposed to different levels of climate change (Fig. 2), we draw on literature for exposure to sea-level rise at different levels of warming, as well as our own calculations of extreme heat based on output of global models. We acknowledge that these include a limited number of the possible impacts of climate change. (1) Projections of sea-level rise need to account for dynamic processes of different complexity and for various spatiotemporal scales. In particular, the immediate response of several sea-level rise contributors (such as ice sheets and inland glaciers) to global warming is only marginal due to their high inertia but can be orders of magnitude higher on centennial timescales. Therefore, to draw a meaningful connection between selected temperature levels and triggered sea-level rise, recent literature[117,118] has resorted to a twofold approach. The transiently realized sea-level rise throughout the twenty-first century is assessed by pooling Shared Socioeconomic Pathway and Representative Concentration Pathway scenarios by

their end-of-century stabilization temperature. Those pools (for example, all scenarios that end up at $2 \pm 0.25$ °C) are used to drive localized models of sea-level rise, resulting in estimates for sea-level rise at 2100 for different end-of-century warming stabilization levels[117,119]. Additionally, these twenty-first century projections can be complemented with multi-centennial estimates since long-term sea-level rise is governed by the equilibria of the cryosphere elements and ocean thermal expansion[120]. In the next step, assessing exposure on these different timescales would require population projections, which are available for the twenty-first century but futile for longer timescales. For consistency, we therefore refer to a recent study that quantifies the number of people currently (baseline from that paper: 2010 population of 6.8 billion people) inhabiting land that is subject to inundation by end of this century or on a multi-centennial timescale, without accounting for potential adaptation through migration, coastal defences and so on[117].

(2) Wet bulb temperature ($T_W$) exposure was calculated for the historical time period of 1979–2014 and the Shared Socio-Economic Pathway 2-4.5 future scenario for 2015–2100. Wet bulb temperature was calculated following the Davies-Jones[121] method. Global gridded temperature and relative humidity data with a grid spacing of $1.25° \times 1.25°$ at 6-h intervals were downloaded from a bias-corrected global dataset[122] based on 18 models from the Coupled Model Intercomparison Project Phase 6 and the European Centre for Medium-Range Weather Forecasts Reanalysis 5 dataset. We aggregated the data to create a maximum daily $T_W$ dataset and then interpolated this to match the $1° \times 1°$ grid spacing of the spatially explicit data for the 2020 population distribution (most recent available, global total 7.7 billion people) from the UN WPP-Adjusted Population Count, v.4.11 (ref. 123). We then calculated the wet bulb exposure by summing up the population count for all cells with at least 1 day with a maximum $T_W > 35$ °C. The $T_W$ threshold of 35 °C was chosen as it is often considered to be a human physiological limit of tolerance to heat stress. The human body is unable to cool itself beyond $T_W = 35$ °C (ref. 124,125). An average 1 day per year over this temperature per year is therefore a conservative indicator in assessing human exposure to heat stress, which does not account for annual variability. We then plotted the total number of people exposed to 1 day with a maximum $T_W > 35$ °C in a year against the mean annual global warming associated with that year to construct an exposure–temperature response curve.

(3) We calculate the number of people displaced from the human climate niche[8] at different levels of warming, following the method of Lenton et al.[41]. The number of people exposed to mean annual temperatures greater than 29 °C was calculated for different global mean temperature increases under four different Shared Socio-Economic Pathways. We used the downscaled spatially explicit output from the Coupled Model Intercomparison Project phase 6 available from the WorldClim v.2.0 database at 0.0833° (approximately 10-km) resolution (available at https://worldclim.org). The exposed population is based on a 2010 population of 6.9 billion with spatial distribution as given by the History Database of the Global Environment 3.2 database[126]. The mean annual temperature threshold of 29 °C was chosen as it is beyond what humans have historically been exposed to[8].

To calculate current subglobal ESB transgressions (Fig. 3), we use data for the above wet bulb and low-elevation coastal zones[127] as proxies for climate impacts, biosphere functional integrity (Supplementary Methods), surface water and groundwater (Supplementary Methods), exceedance of local safe and just nitrogen surplus and phosphorus concentration (Supplementary Methods) and $PM_{2.5}$ concentrations[128]. For population, we used the UN WPP-Adjusted Population Count v.4.11 (ref. 123).

There are many uncertainties and limitations in this justice analysis. Lack of sufficient data on humans, communities and countries worldwide harmed by biophysical degradation is a key constraint. There is also considerable uncertainty regarding impacts on current generations, future generations, and specific countries and communities. In this paper, we also do not quantify issues of access[26], explore the implications of access for the safe and just corridor or discuss why it is difficult to meet issues of access without transforming our governance systems.

## Ethics and inclusion statement
Earth Commissioners were selected by the Future Earth Advisory Committee following an open call for nominations with consideration for balancing gender, geographical region and expertise to the extent possible. Members of working groups were selected by the working group co-leads following an open call and approved by the Earth Commission, with attention paid to balancing gender, geographical region and expertise to the extent possible.

## Reporting summary
Further information on research design is available in the Nature Portfolio Reporting Summary linked to this article.

## Data availability
The data supporting Figs. 2 and 3 are available at https://doi.org/10.6084/m9.figshare.22047263.v2 and https://doi.org/10.6084/m9.figshare.20079200.v2, respectively. We rely on other published datasets for the climate boundary[16], N boundary[72] (model files are at https://doi.org/10.5281/zenodo.6395016), phosphorus[73,74] (scenario breakdowns are at https://ora.ox.ac.uk/objects/uuid:d9676f6b-abba-48fd-8d94-cc8c0dc546a2, and a summary of agricultural sustainability indicators is at https://doi.org/10.5281/zenodo.5234594), current N surpluses[129,130] (the repository at https://dataportaal.pbl.nl/downloads/IMAGE/GNM) with the critical N surplus limit[72] subtracted, and estimated subglobal P concentration in runoff based on estimated P load to freshwater[131] and local runoff data[132,133]. Current functional integrity is calculated from the European Space Agency WorldCover 10-metre-resolution land cover map (https://esa-worldcover.org/en). The safe boundary and current state for groundwater are derived from the Gravity Recovery And Climate Experiment (http://www2.csr.utexas.edu/grace/RL06_mascons.html) and the Global Land Data Assimilation System (https://disc.gsfc.nasa.gov/datacollection/GLDAS_NOAH025_3H_2.1.html). More information is available in 'Code availability' and Supplementary Methods. Source data for Fig. 2 are provided with this paper.

## Code availability
The code used to produce Figs. 2 and 3 are available at https://doi.org/10.6084/m9.figshare.22047263.v2 and https://doi.org/10.6084/m9.figshare.20079200.v2, respectively. The code used to make the nutrient Earth system boundary layers in Fig. 3 is available at https://doi.org/10.5281/zenodo.7636716. The code used to make the surface water layer in Fig. 3 and derive the subglobal Earth system boundaries for surface water is available at https://doi.org/10.5281/zenodo.7674802. The code to estimate current functional integrity is available at https://figshare.com/articles/software/integrity_analysis/22232749/2. The code to derive the groundwater layer in Fig. 3 and derive the total annual groundwater recharge is available at https://doi.org/10.5281/zenodo.7710540.

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

**Acknowledgements** This work is part of the Earth Commission, which is hosted by Future Earth and is the science component of the Global Commons Alliance. The Global Commons Alliance is a sponsored project of Rockefeller Philanthropy Advisors, with support from the Oak Foundation, MAVA, Porticus, the Gordon and Betty Moore Foundation, the Tiina and Antti Herlin Foundation, William and Flora Hewlett Foundation and the Global Environment Facility. The Earth Commission is also supported by the Global Challenges Foundation and the Frontiers Research Foundation. Individual researchers were supported by the European Research Council (Grant on Climate Change and Fossil Fuel 101020082 to J.G. and Advanced Grant grant ERC-2016-ADG 743080 to J. Rockström), the Open Society Foundations (J.F.A. and T.M.L.), the Australian Government (Australian Research Council Future Fellowship FT200100381 to S.J.L. and Australian Research Council Discovery Early Career Researcher Award DE230101327 to C.N.) and the Swedish Research Council Formas (Grant 2020-00371 to S.J.L.).

**Author contributions** J. Rockström, J.G., D.Q., X.B., G.B., S.E.B., F.D., K.E., C.G., N.K., T.M.L., D.M.L., N.N., D. Obura, D.v.V., P.H.V. and R.W. conceptualized the work. J.F.A., L.S.A., D.I.A.M., D.C., L.G., S.H., T.M.L., S.L., A.M., D. Ospina, K.P., C.R., B.S., J.S., B.S.-K., T.T., C.Z., E.M.B., S.B., W.B., P.G., L.H., L.J., C.N., S.P., J. Rocha, M.S., L.S.-U., W.d.V., C. Xiao, C. Xu, X.X., N.Z.-C. and X.Z. gathered and analysed data. J. Rockström, J.G., D.Q., S.J.L., X.B., G.B., S.E.B., F.D., K.E., C.G., N.K., T.M.L., D.M.L., N.N., D. Obura, D.v.V., P.H.V., R.W., J.F.A., L.S.A., D.I.A.M., D.C., L.G., S.H., S.L., A.M., D. Ospina, K.P., C.R., B.S., J.S., B.S.-K., T.T., C.Z., E.M.B., P.G., C.N., L.S.-U., W.d.V. and X.Z. wrote the paper. S.J.L. coordinated writing.

**Funding** Open access funding provided by Stockholm University.

**Competing interests** The authors declare no competing interests.

**Additional information**
**Correspondence and requests for materials** should be addressed to Johan Rockström or Steven J. Lade.

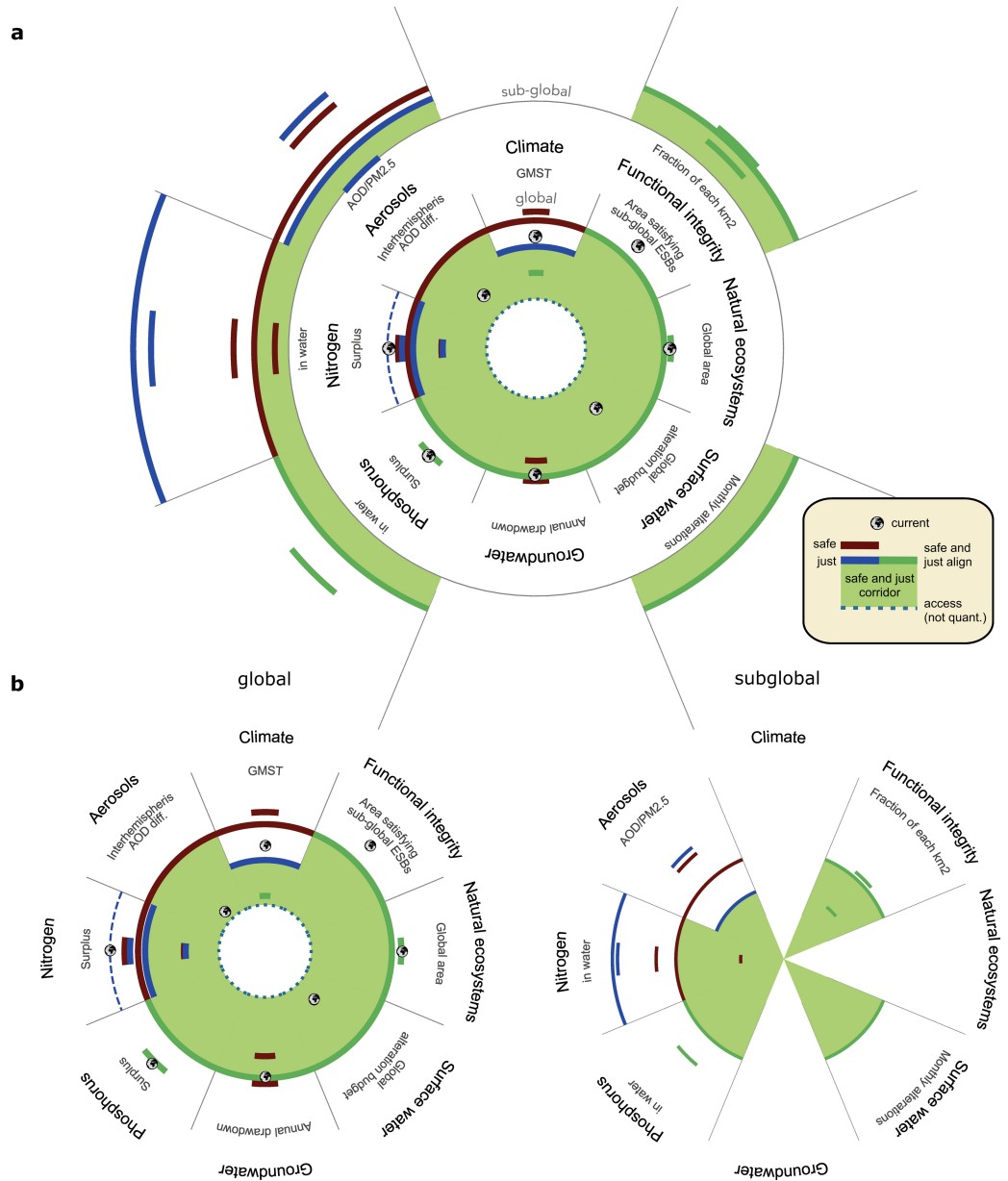

**Extended Data Fig. 1 | Alternative visualizations of safe and just Earth system boundaries (Fig. 1).** Concentric (**a**) and parallel (**b**) visualizations of global (**a**, inner circle; **b**, left circle) and sub-global (**a**, outer circle; **b**, right circle) safe and just ESBs. Colours are as in Fig. 1. Global rings (**a**, inner circle; **b**, left circle) show current global states; a single current state cannot be defined sub-globally. Short concentric lines (that extend across less than the full width of a wedge) represent alternative likelihood levels (safe) or levels of exposure (just NSH) (Table 1).

**Extended Data Table 1 | Assessment of levels of confidence in each domain's safe Earth system boundaries**

| Earth system domain | Overall confidence | Robustness of evidence base | Degree of scientific agreement |
|---|---|---|---|
| Climate | *High confidence* | Given the uncertainty on some evidence streams, for example, paleoclimate reconstructions of Holocene or Eemian maximum temperatures[107,108,118], or on the assessment of climate tipping points in the 1-2°C range, overall this is taken as a *medium evidence robustness*. | There is *high evidence consistency* and *agreement* in the scientific literature of the Earth system being fundamentally altered with respect to climate change between 1 and 2°C. |
| Biosphere: natural ecosystem area | *Medium confidence* | There is *limited evidence* in the scientific literature addressing the relationship between area of natural ecosystems and Earth system functions and services (the majority of studies use the same method, and the range of estimates remains similar), with several conservation planning studies that relate largely intact natural area to extinction risks. | Among experts the *level of agreement is high* regarding the proposed ranges. |
| Biosphere: Functional integrity | *Medium confidence* | Highly variable robustness of the evidence across NCP: *robust evidence* for pollination, pest and disease control and water quality regulation, *medium evidence* for soil protection, *limited evidence* for natural hazards (variable quantity of studies, generally diverse methods used, and high consistency in findings). | *Medium degree of scientific agreement* in the literature. |
| Water: surface and ground | *Medium confidence* in the sub-global safe ESB for surface water flow. *High confidence* on the globally aggregated surface water and groundwater volumes. | The underlying flow data used in this analysis is derived from a well-documented and verified water balance model[132,133]. Output flow results are shown to be consistent with global discharge estimates and agricultural water consumption found in the scientific literature for both similar models and observed datasets (Table S3). Taken together, these suggest a *robust evidence base*. | *Scientific agreement* is considered *high*, based on the assessed literature and opinion of working group experts. The sub-global safe surface water flow boundary is recognised as a suitable boundary in the absence of detailed flow-ecology relationships[59] and it has been adopted elsewhere in global-scale assessments (Supplementary Methods). |
| Nutrients | *Medium confidence* with respect to terrestrial systems. Due to substantial literature uncertainty and limited global modelling, there is *low confidence* for a nutrients ESB on ocean systems and hence none provided in this assessment. | *Medium-to-limited evidence robustness* due to a modest quantity of papers. | *Medium-to-high evidence consistency* and *agreement* in the literature on proposed global N/P boundary values. |
| Aerosols | *Low confidence* | There is high confidence in the physical mechanism by which the aerosol emissions from NH would influence the tropical monsoons[84]. However, the uncertainty in the quantification of the interhemispheric AOD difference could be large due to aerosol-cloud interactions. Therefore, we assess *low level of evidence*, in terms of quantity and consistency in findings. | *Medium degree of scientific agreement* in the scientific literature assessed and among working group experts |

For more information see Methods. The *robustness of evidence* and *degree of agreement* of all ESB quantifications are based on the assessment of available literature and working group experts' views.

# Reporting Summary

## Statistics

For all statistical analyses, confirm that the following items are present in the figure legend, table legend, main text, or Methods section.

| n/a | Confirmed | |
|---|---|---|
| ☐ | ☒ | The exact sample size (*n*) for each experimental group/condition, given as a discrete number and unit of measurement |
| ☒ | ☐ | A statement on whether measurements were taken from distinct samples or whether the same sample was measured repeatedly |
| ☒ | ☐ | The statistical test(s) used AND whether they are one- or two-sided *Only common tests should be described solely by name; describe more complex techniques in the Methods section.* |
| ☒ | ☐ | A description of all covariates tested |
| ☒ | ☐ | A description of any assumptions or corrections, such as tests of normality and adjustment for multiple comparisons |
| ☐ | ☒ | A full description of the statistical parameters including central tendency (e.g. means) or other basic estimates (e.g. regression coefficient) AND variation (e.g. standard deviation) or associated estimates of uncertainty (e.g. confidence intervals) |
| ☒ | ☐ | For null hypothesis testing, the test statistic (e.g. *F*, *t*, *r*) with confidence intervals, effect sizes, degrees of freedom and *P* value noted *Give P values as exact values whenever suitable.* |
| ☒ | ☐ | For Bayesian analysis, information on the choice of priors and Markov chain Monte Carlo settings |
| ☒ | ☐ | For hierarchical and complex designs, identification of the appropriate level for tests and full reporting of outcomes |
| ☒ | ☐ | Estimates of effect sizes (e.g. Cohen's *d*, Pearson's *r*), indicating how they were calculated |

*Our web collection on statistics for biologists contains articles on many of the points above.*

## Software and code

Policy information about availability of computer code

| Data collection | No software used for data collection. |
|---|---|
| Data analysis | Software used includes: R version 4.1.3 (Figures 2 and 3), Google Sheets (accessed in 2022, to track papers for functional integrity literature review), Google Earth Engine (accessed in 2022, for current state of functional integrity), Python v3.7.6 using Jupyter notebook web-based interactive computing platform v 6.2.0 (development of environmental flows and surface water boundaries raster datasets), QGIS v3.22.9 (for data review of surface water boundary), Water Balance Model (accessed in 2022), MATLAB R2018A (for processing the GRACE data and GLDAS model output for the groundwater boundary). |

For manuscripts utilizing custom algorithms or software that are central to the research but not yet described in published literature, software must be made available to editors and reviewers. We strongly encourage code deposition in a community repository (e.g. GitHub). See the Nature Portfolio guidelines for submitting code & software for further information.

## Data

Policy information about availability of data

All manuscripts must include a data availability statement. This statement should provide the following information, where applicable:

- Accession codes, unique identifiers, or web links for publicly available datasets
- A description of any restrictions on data availability
- For clinical datasets or third party data, please ensure that the statement adheres to our policy

> Please see data availability statement in manuscript.

## Human research participants

Policy information about studies involving human research participants and Sex and Gender in Research.

| | |
|---|---|
| Reporting on sex and gender | N/A |
| Population characteristics | N/A |
| Recruitment | N/A |
| Ethics oversight | N/A |

Note that full information on the approval of the study protocol must also be provided in the manuscript.

# Field-specific reporting

Please select the one below that is the best fit for your research. If you are not sure, read the appropriate sections before making your selection.

☐ Life sciences      ☐ Behavioural & social sciences      ☒ Ecological, evolutionary & environmental sciences

For a reference copy of the document with all sections, see nature.com/documents/nr-reporting-summary-flat.pdf

# Ecological, evolutionary & environmental sciences study design

All studies must disclose on these points even when the disclosure is negative.

| | |
|---|---|
| Study description | Methodologies for each domain is included in text, see Methods and Supplementary Methods. These methodologies varied by domain, including relying upon recently published literature review, recently published modelling, new literature review, and new analysis based on well-established datasets. We here include the new literature review for the functional integrity boundary as an example. |
| Research sample | Web of Science |
| Sampling strategy | Search terms as described in Supplementary Methods, Table S2. |
| Data collection | Awaz Mohamed ran the searches. |
| Timing and spatial scale | Limited to papers between 2010 and 2021 (see Supplementary Methods). |
| Data exclusions | Papers were filtered in several steps (Supplementary Methods), including "Of those [402 potentially eligible] studies we conducted a full text review to confirm eligibility which was established if the reference contained measures of habitat quantity within 1 km at the landscape scale with measures appearing either directly in the source text, tables, supplementary information or in figures." |
| Reproducibility | The functional integrity boundary for several NCPs were studied and compared (Table S2). |
| Randomization | N/A |
| Blinding | N/A |

Did the study involve field work?      ☐ Yes      ☒ No

# Reporting for specific materials, systems and methods

We require information from authors about some types of materials, experimental systems and methods used in many studies. Here, indicate whether each material, system or method listed is relevant to your study. If you are not sure if a list item applies to your research, read the appropriate section before selecting a response.

## Materials & experimental systems

| n/a | Involved in the study |
|-----|----------------------|
| ☒ | Antibodies |
| ☒ | Eukaryotic cell lines |
| ☒ | Palaeontology and archaeology |
| ☒ | Animals and other organisms |
| ☒ | Clinical data |
| ☒ | Dual use research of concern |

## Methods

| n/a | Involved in the study |
|-----|----------------------|
| ☒ | ChIP-seq |
| ☒ | Flow cytometry |
| ☒ | MRI-based neuroimaging |

