## [Peer Review File · Nature]

Manuscript Title: Safe and just Earth system boundaries

Reviewer Comments & Author Rebuttals

Reviewer Reports on the Initial Version:

Referee #1 (Remarks to the Author):

This manuscript updates the planetary boundaries, which now are named earth system boundaries. The two main new developments is the assessment of the safe boundaries at the sub-global scale and linking them to the previous global boundaries and the adoption of Earth system justice criteria (intergenerational, intragenerational and interspecies) to make them "just". I think in some aspects this is an improvement of the previous planetary boundaries, but this improvement is more of a refinement than a completely novel proposal. There is a lot of work behind the new concept as can be seen in the supplementary material, but I fill that without substantial revision the paper may contribute to the uncritical reinforcement of the planetary boundary ideas. Below I explain in more detail.

General Comments

1. There has been many criticisms of the planetary boundaries concept. See for instance a review by Biermann and Kim (2020) *Ann. Rev. Env. Resour.* <https://doi.org/10.1146/annurev-environ-012320-080337>. This paper seems to mostly ignore the criticisms, both in the introduction and in the discussion. This is a missed opportunity. First, because one of the developments in this paper (the connection between sub-global and global boundaries) addresses one of the main issues with the previous concept (the inability to downscale the concept). This issues is alluded to in l.138-141, but more should be said. Second, because it continues to frame the boundaries as scientifically determined instead of recognising that they are to a large extent judgement calls by a group of experts, and that a different group of experts, or even a group of policy-makers, may want to make a different call.
2. There has been a lot of discussion on whether tipping points or thresholds can be consistently found in ecological systems, see for instance Hillebrand, H. et al. *Nat. Ecol. Evol.* <https://doi.org/10.1038/s41559-020-1256-9>. So defining 50-60% of global surface as natural ecosystem area is, in my opinion, mainly a judgement call. But this is never stated as such, but instead as justified by the literature (SM). Similar statements can be made about most of the boundaries - they are inspired by some scientific studies but they are not determined by science but by a group of experts opinions, and there are large uncertainties around them.
3. It would be interesting to quantify those uncertainties and summarise them in Figure 1. They are likely to differ across the different boundaries and this is in itself interesting.
4. I was not fully convinced on the soundness of the "just" analysis with the three Earth System Criteria. They seem very general principles that require some more detailed socio-economic analysis

to be applied to the determination of boundaries. Overall I found it difficult to understand how these principles were applied (the supplementary methods did not clarify my doubts). For instance, there are statements like in I. 290 "their implementation should be complemented with justice concerns around distribution and quality", or I.2390-240 "additional interventions often needed" which seem vague.

5. It is not clear what is meant by intact natural ecosystems. In I.213 it is stated that they do imply protection excluding human inhabitation and sustainable use. Later it is clarified that indigenous populations are "ok" in these ecosystems. However, there may be other forms of sustainable use which do lead to the tipping points which are addressed by this boundary, and depending on how one defines "intact natural ecosystems" we may not have surpassed it.

6. In the water boundary I was surprised not to see a mention to the barrier effects of dams. Changing of the flow volume and timing are important disturbances to river ecosystems, but the barrier effects on migrating species are very serious and I would like to see it somehow mentioned or acknowledged.

Specific Comments

I. 249 Can you briefly elaborate on what empirical evidence supports the 20% flow alteration principle?

I. 312-315. It would be nice to have a simple intuitive explanation of the N/P limits.

I. 425-425. This statement on capping the top percentiles of the world's richest is not backed with any data/analysis. Would be interesting to have an idea of what exact percentile we are talking about and how that cap would work.

Figure 2. Exposure is not necessarily impact. This distinction should be highlighted in the text.

Figure 3. I don't fully understand the criteria used for climate here (one day >35C). Maybe consider other criteria being used in the last IPCC assessment?

SM 1.2.1 The 15% CBD restoration criteria was a policy goal and not really "scientifically justified". And the Strassburg study does not really look at thresholds.

SM. 2.4.2.1 The paragraph that starts with "However, we note that extra attention must be given to I2". I really don't understand how current generation needs to have costs so that future generations can benefit from functional integrity. I think a goal of 10-25% of natural elements in agricultural landscapes would also benefit current generations through ecosystem services.

SM. I missed a section 1.2.2 on "Functional Integrity"

Referee #2 (Remarks to the Author):

I read the manuscript several times, more often than in general, but always with a similar conclusion: I miss the innovation of this manuscript and suggest to decline it.

This manuscript reads like a review paper with 144 references to be found in the main text and methods plus the references in the SI. The overall method has not really changed to what has already been published or is not new at all. What makes the difference to former publications? Setting a global alteration budget instead of global sum of water abstractions or consumption does finally not lead to something new. In global scale modelling, a limiting alteration of flow of 20% is (often) used as an indicator for environmental flow requirements. We know that this indicator is too broad and not specific enough to mimic aquatic ecosystem needs. There are more suitable approaches available on the global scale to determine environmental flow requirements (e.g. Pastor et al. 2014). Liu et al. 2021 compared the effect of different EFR approaches on water scarcity which cannot be neglected. However, it is difficult to find a suitable indicator which quantifies the ESB of a renewable resource. It is feasible when accounting for water quality in terms of quantities (consideration of loads and concentration) instead of sanitation information solely.

Referee #3 (Remarks to the Author):

Comments on NATURE-2022-06-09372A

The manuscript examines a series of Earth System Boundaries (ESB) related to fundamental processes governing the future of the planet. More specifically, the manuscript assesses what could be the safe and just values for each ESB and to what extent these safe and just boundaries have been exceeded both globally and regionally. Safe boundaries are defined in the perspective of maintaining the resilience and stability of the Earth System, while just boundaries are defined to minimise exposure to significant harm to humans from Earth System change (in some environmental justice perspective). The manuscript builds upon previous studies – mostly related to the planetary boundary framework – by (i) revising and updating the safe boundaries previously proposed and (ii) providing some just boundaries for each Earth System process considered.

The novelty brought by this manuscript is strong. One noticeable added-value of this manuscript is related to the definition, quantification, and mapping of what could be the just values for each ESB, beyond the previously published safe values. Because this study has a global perspective and because it addresses a large set of Earth System processes – related to climate, the biosphere, water and nutrient cycles, and aerosols – it is likely to meet a global audience.

I consider this manuscript acceptable for publication in Nature due to its large scope, strong novelty and robust methods. However, I consider that significant, major improvements have to be undergone before its publications.

First, the foundations for defining the just boundaries were overall unclear to me. I must say that I

am not a specialist of fairness and justice concepts but I consider that some overall clarifications are needed on this for the broad audience of the Nature journal. More precisely, although I could understand the general rationale, I could not fully understand what the authors mean by using “three Earth system justice criteria (intergenerational, intragenerational and interspecies & Earth system stability) to assess whether adhering to the safe ESBs could protect people from significant harm” (line 109). How is inter-generational justice considered here? Is this somehow connected to long-term impacts of Earth System changes? What does interspecies mean? How Earth System stability is related to justice? How does all this relate to environmental justice? In addition, because just boundaries are defined to minimise exposure to harm from Earth System change, I cannot really understand how this relates to inter-generational justice criteria. Finally, the way just boundaries articulate with safe boundaries is not always clear. For instance, the sentence line 149 (“assuming that all safe boundaries by definition meet our interspecies justice and Earth system stability criteria”) is very unclear. I know that this has been developed in details for each considered Earth System component or process in the Supp Info but I feel more justification is needed in the Main Text.

Second, although this work brings a lot of materials of great importance for guiding human actions in the Anthropocene, I found the “Translation for actors” and “Transformation to live within Earth System boundaries” sections within the Discussion a bit disappointing. In their current status, those sections are quite general and vague, and may lack the appropriate strength to actually translate the study outcomes to the decision-makers. In particular, I consider that some developments are needed about how the outcomes from this study could feed the global debate about green growth, GHG-GDP decoupling, and degrowth as well as about the future for agriculture and food systems.

Third, as requested by the Editor, I provide a series of specific comments about the N and P cycles. Although the corresponding sections are overall fine and the related methods look robust, I consider that several aspects need some improvements:

- Several key aspects of the global N and P cycles are actually missing in this study:
 - o First, nothing is said about N₂O losses and the way they are considered in the definition of the safe and just boundaries. Because N₂O is a long-lived climate pollutant with strong climate warming effects, I would have expected it to be included in the ESB definition. I guess that N₂O is somehow already included within the climate boundary (itself expressed in GMST above pre-industrial level) but nothing is clearly explained about this. Some non-CO₂ budgets exist – in addition to the CO₂ budgets provided by the IPCC – in order to attenuate climate warming and I wonder how these budgets relate to the proposed boundaries for the global N cycle.
 - o Second, I was surprised to read that NH₃ emissions to the atmosphere were only considered in the perspective of N deposition on terrestrial ecosystems. NH₃ air concentrations have also strong effects on human health both directly and indirectly as precursors for aerosols, as acknowledged by the authors on pages 17 and 18 of Supp Info. More details about this and if possible inclusion of the NH₃ effects on harm to humans are expected here.
 - o Third, a major issue about the global P cycle is related to the just use of the globally remaining phosphate rocks. The future of these phosphate rocks is very likely to have severe effects on the global food security for the next coming decades and centuries. I understand that the just definition provided in this study is related to the “harm to humans from Earth System changes” and this differs from the just use of natural resources. However, because food insecurity – partly related to lack of

access to phosphate rock and P fertiliser resources – is a significant harm to humans, I would have expected some clarification about how this phosphate rock issue is considered in this study.

- The way the safe boundaries have been defined for N and P surplus is unclear to me. I understand that the authors have used some boundaries for N and P concentrations in both freshwater and groundwater as well as N emissions to air, and that they related these water concentrations and air emissions to N and P surplus in agricultural soils through some sort of modelling but I could not understand the bases for this modelling approach. Much of the Supp Info about this is actually making reference to other articles whereas I think some clarifications about the compartments and process considered is needed here. More specifically, it is unclear to what extent the authors have accounted for strong soil and climate effects on N and P leaching and erosion globally.
- Finally, details are provided in the Supp Info about how just boundaries have been defined and why they align on safe boundaries for P (and partly for N). However, these details are quite buried in the long Supp Info and may not be accessible to most readers. I recommend instead making this more explicit in the Main Text for both nutrients (e.g, around line 321).

Minor comments

- Much of the 'Methods', in particular about estimate uncertainties and confidence levels are actually results. I think those uncertainty considerations could be moved to the Supp Info while some key methodology details could be moved from the Supp to the Main Text.
- Some clarifications are needed about why NH₃ emissions are considered in this study: while N deposition is mentioned as a considered boundary line 307 it is unclear how it translates into NH₃ emissions line 801.
- Several sentences are unclear, at least for non-native English people such as me. See for instance lines 69, 117-119, 209-210, 299-302, 535-536.
- Supp page 15: what does 'new fixation' mean? I guess it refers to both biological and industrial nitrogen fixation but please clarify.
- Table S6: what does 'current excess' mean? I guess it is value above the natural baseline as in Table S8 but please clarify.

Referee #4 (Remarks to the Author):

Summary of the key results: This study is an assessment and synthesis of evidence in support of a set of critical thresholds for different compartments and cycles of the earth system (e.g. atmosphere, ecosystems, water resources, nutrients essential for life) that are argued to constitute limits for global society and nature to remain safe from harm from climate change and its impacts, while also upholding fairness (justice) among countries/regions and human generations. It is framed as an extension and elaboration of the Planetary Boundaries framework, introduced by partly the same group of authors a decade or so ago. The Planetary Boundaries concept and language has been influential in science-policy work around climate action and sustainability in recent years.

Originality and significance: As noted, this work extends on the Planetary Boundaries framework (PB)

which is already well-established in the climate and sustainability literature. The framework has been criticised on various grounds, but there is no denying it has been found useful by the applied research community, providing a way of linking scientific evidence from impact and modelling studies in a wide range of fields or 'domains' to each other, and to policy goals for climate action (Paris Agreement), sustainability (SDGs) and related areas. The present work is an extension of PB, which limits its originality, but it does include an extensive synthesis of new data and literature of relevance to identifying/quantifying the boundaries, including findings of other major assessments such as IPCC-AR6. Apart from new data, the major novelty probably lies in the juxtaposition of justice alongside danger considerations in setting the proposed boundaries. However, these criteria are already combined in the seemingly very similar conceptual framework of "doughnut economics". The differences between these frameworks are not very clear to me, and may largely boil down to different emphasis – socio-economic factors in "the doughnut", earth system processes in this paper. However, the authors claim there is novelty in that the limits to what is safe and what is just are expressed in the same units (e.g. degrees of warming for climate) in their framework, aiding the setting of policy targets, for example. The inclusion of justice criteria addresses one criticism of the original PB concept, that it fails to explicitly consider opposing impacts on people and nature in different world regions, with the potential for perverse outcomes if policy frameworks are set around a single global target (such as the Paris Agreement's <2 degrees of warming relative to pre-industrial climate).

Overall, I would rate this as a significant piece of work that will influence environmental and sustainability research and science-policy work going forward.

Data & methodology: validity of approach, quality of data, quality of presentation. This is a synthesis and assessment combined with tweaks to an existing conceptual framework, so the methodology mainly comprises identifying, distilling and interpreting relevant literature to come up with numbers for the 'proposed safe and just earth system boundaries' as summarised in Table 1. Details of the literature consulted and principles for assigning values to boundaries are provided in the 55 page Supplementary Methods. This extensive material covers a vast diversity of fields, making it challenging to judge the adequacy of this effort in terms of fairly capturing the state of knowledge within each of the considered domains. It is clear that it reflects a substantial body of underpinning work, and does seem to embrace other recent assessments and synthesis.

Focusing on the fields into which I have the best insight, I looked more closely in the supplement at how boundaries for climate change (including related feedbacks) were defined and quantified. In the Methods the general approach is described thus: "We used two main groups of approaches to setting safe earth system boundaries: a 'multiple elements approach' and a 'spatial aggregation approach'." For climate change, the chosen criterion for deciding what level of change becomes dangerous was whether "multiple tipping points are crossed". I suppose this is the 'multiple elements approach' in action, but the idea seems to come from a paper currently in preprint (Armstrong McKay et al 2021, S1 References). Without consulting this paper, I couldn't find a clear description of the basis for which crossing one tipping point should be considered safe whereas two or more are deemed dangerous. It could be argued that one tipping point is enough to be worried. Figure S1 (seemingly reproduced from the paper above) displays the assessed tipping points with pseudo-uncertainty estimates shown. The safe boundary is identified at 1 degree of warming, since at this level, committed loss of the Greenland ice sheet is the one allowable tipping point. This figure

is useful for understanding, and the data it displays seem broadly credible, acknowledging substantial knowledge gaps around the likelihood and size of many of these climate system features/impacts, or indeed whether they represent true tipping points. For example, (degradation/loss of the) Amazon rainforest due to runaway evapotranspiration-rainfall feedback has been much debated and demonstrated using some models and not by others. There is no consensus this is a real tipping point at all, or if it is, at what level of climate warming it can be expected to kick in. However, I accept an assessment like this has to capture 'best current understanding', and the wide uncertainty bounds around Amazon loss and other features in Figure S1 capture this in a seemingly reasonable way. By the way, some features in this figure are lacking a legend or clear explanation in the figure text, including the "SSP projections" at far right, and the horizontal line and shading around 2-4 degrees.

Conclusions: robustness, validity, reliability. My main concern is probably with the criteria chosen to define what constitutes an earth system boundary, and the premise that identifiable boundaries exist across each of the domains considered. The authors' criteria for the existence of a likely or possible boundary are given on page 3: "To determine safe boundaries, we use an assessment of tipping point risks among local and regional tipping elements ... components or thresholds that regulate the functioning and state of the planet and show evidence of having thresholds at which small additional perturbations can trigger self-reinforcing changes that undermine Earth system resilience". This suggests that the boundaries considered all represent thresholds at which system-internal feedbacks start to occur, or accelerate markedly, pushing the planet into "regime shift" (Supplement page 2) and a less desirable state. The possible existence of such critical thresholds has probably been argued in the past for at least some impacts or phenomena within each of the chosen domains. But, as noted above, the evidence certainly for some of the suggested tipping points in the climate system are sparse and/or contradictory. Most such evidence tends to come from model studies as opposed to observational trends, and different but similar models often show contrasting outcomes. To put it another way, many of these tipping points exist in theory, but we often lack solid evidence that they occur in practice. If tipping points are uncommon in the Earth system, how meaningful is it to identify safe boundaries this side of the tipping point? The authors do not really argue for their belief in boundaries based on the extensive literature they have reviewed. The boundaries are taken as given, and the evidence is deployed to argue for the size/value of the boundary. As a contribution to debate around the resilience of the earth system and its current (Holocene) state, arguments could be made clearer. But perhaps this is too much to expect from one paper.

Clarity and context: lucidity of abstract/summary, appropriateness of abstract, introduction and conclusions. I found the paper well-written and comprehensive, given the complexity and disciplinary breadth of the work.

Referee #5 (Remarks to the Author):

I have reviewed the "Aerosol Pollution" section of the manuscript "Safe and just Earth System Boundaries" by Rockstrom et al. My main comment is that the use of studies analyzing the effects of volcanic aerosol emissions or purposeful stratospheric aerosol

injections to support the use of interhemispheric asymmetry in aerosol optical depth (AOD) as a metric for safe ESB is weak. I understand that a more elaborate and nuanced discussion is provided in the supplementary but this is not reflected in the main text and needs to be captured in a more clear and transparent way. The evidence for the role of interhemispheric asymmetry in AOD on hemispheric precipitation (lines 337-339) needs to be predicated on observations in addition to modeling studies. Further, because of aerosol-cloud interactions, I do not expect the response to “an increased concentration of reflecting aerosols in one hemisphere” to be straight-forward (see Douville et al., 2021; Allen et al., 2015).

Additional specific comments:

L339-340: But we know that air pollutants (particularly aerosols) are declining (e.g., Quaas et al., 2022) and will most likely continue to decline as human health concerns will lead to more stringent controls on air pollution in developing countries. All SSP scenarios except SSP3.70 project declining emissions of short-lived climate forcers (including aerosols) (see Szopa et al., 2021 Figure). In fact, I would argue that the impact of increasing greenhouse gases on precipitation will dominate over that of aerosols in the long-term (see Lee et al., 2021),

L342: all aerosols are short-lived.

References:

Allen, R. J., Evan, A. T., & Booth, B. B. B. (2015). Interhemispheric Aerosol Radiative Forcing and Tropical Precipitation Shifts during the Late Twentieth Century, *Journal of Climate*, 28(20), 8219-8246.

Douville, H., K. Raghavan, J. Renwick, R.P. Allan, P.A. Arias, M. Barlow, R. Cerezo-Mota, A. Cherchi, T.Y. Gan, J. Gergis, D. Jiang, A. Khan, W. Pokam Mba, D. Rosenfeld, J. Tierney, and O. Zolina, 2021: Water Cycle Changes. In *Climate Change 2021: The Physical Science Basis. Contribution of Working Group I to the Sixth Assessment Report of the Intergovernmental Panel on Climate Change* [Masson-Delmotte, V., P. Zhai, A. Pirani, S.L. Connors, C. Péan, S. Berger, N. Caud, Y. Chen, L. Goldfarb, M.I. Gomis, M. Huang, K. Leitzell, E. Lonnoy, J.B.R. Matthews, T.K. Maycock, T. Waterfield, O. Yelekçi, R. Yu, and B. Zhou (eds.)]. Cambridge University Press, Cambridge, United Kingdom and New York, NY, USA, pp. 1055–1210, doi:10.1017/9781009157896.010.

Lee, J.-Y., J. Marotzke, G. Bala, L. Cao, S. Corti, J.P. Dunne, F. Engelbrecht, E. Fischer, J.C. Fyfe, C. Jones, A. Maycock, J. Mutemi, O. Ndiaye, S. Panickal, and T. Zhou, 2021: Future Global Climate: Scenario-Based Projections and Near-Term Information. In *Climate Change 2021: The Physical Science Basis. Contribution of Working Group I to the Sixth Assessment Report of the Intergovernmental Panel on Climate Change* [Masson-Delmotte, V., P. Zhai, A. Pirani, S.L. Connors, C. Péan, S. Berger, N. Caud, Y. Chen, L. Goldfarb, M.I. Gomis, M. Huang, K. Leitzell, E. Lonnoy, J.B.R. Matthews, T.K. Maycock, T. Waterfield, O. Yelekçi, R. Yu, and B. Zhou (eds.)]. Cambridge University Press, Cambridge, United Kingdom and New York, NY, USA, pp. 553–672, doi:10.1017/9781009157896.006.

Szopa, S., V. Naik, B. Adhikary, P. Artaxo, T. Berntsen, W.D. Collins, S. Fuzzi, L. Gallardo, A. Kiendler-Scharr, Z. Klimont, H. Liao, N. Unger, and P. Zanis, 2021: Short-Lived Climate Forcers. In *Climate Change 2021: The Physical Science Basis. Contribution of Working Group I to the Sixth Assessment Report of the Intergovernmental Panel on Climate Change* [Masson-Delmotte, V., P. Zhai, A. Pirani, S.L. Connors, C. Péan, S. Berger, N. Caud, Y. Chen, L. Goldfarb, M.I. Gomis, M. Huang, K. Leitzell, E. Lonnoy, J.B.R. Matthews, T.K. Maycock, T. Waterfield, O. Yelekçi, R. Yu, and B. Zhou (eds.)]. Cambridge University Press, Cambridge, United Kingdom and New York, NY, USA, pp. 817–922, doi:10.1017/9781009157896.008

Quaas, J., Jia, H., Smith, C., Albright, A. L., Aas, W., Bellouin, N., Boucher, O., Doutriaux-Boucher, M., Forster, P. M., Grosvenor, D., Jenkins, S., Klimont, Z., Loeb, N. G., Ma, X., Naik, V., Paulot, F., Stier, P., Wild, M., Myhre, G., and Schulz, M.: Robust evidence for reversal of the trend in aerosol effective climate forcing, *Atmos. Chem. Phys.*, 22, 12221–12239, <https://doi.org/10.5194/acp-22-12221-2022>, 2022.

Referee #6 (Remarks to the Author):

This article deals with significant matters of real importance, but I do not believe it is in publishable form at this time. While there is, in my view, a strong need for work (indeed a sustained programme) of this sort, aiming to cross the divide between the social and natural sciences in climate matters (though the piece does not restrict itself to climate), this article does not currently do this in an altogether clear or persuasive manner.

The article is very ambitious and full of interesting and suggestive observations and intuitions. It opens numerous research paths, many of which, it seems to me, could comprise research projects in themselves. The distinction between ‘safe’ and ‘just’ boundaries is particularly intriguing and could be pursued in greater detail (though, as will become clear below, I am unpersuaded that it quite works in this piece).

I am not well equipped to assess the physical science claims made here, and will set those aside. I am better positioned to respond to the ‘social science’ side of this piece, on which I will concentrate.

A principal drawback of the piece as submitted is that multiple terms are invoked without much discussion, analysis, or consistency. A broad series of weighty concepts are introduced that are not adequately defined or explained—Anthropocene, ‘social goals’, justice (intergenerational, intragenerational, and interspecies), equity, ‘global commons’, ‘common heritage’—some of which are subsequently abandoned. Arguments and sources sometimes seem cherry-picked, with their implications or relevance not fully digested or explained (what is the ‘global commons’—where does it start and stop, and how is it to be managed?; what does the term ‘Anthropocene’ add to the analysis?). Insofar as there is discussion of these matters, it is late (403-446), incomplete, and limited in scope.

Several apparently key-terms are stapled onto the root term 'earth systems' ('-boundaries', '-stability', '-change', '-resilience', '-justice'). These cross between the biophysical and the sociopolitical—as the authors note—but without any clarity as to how this can be done (ie how to relate physical 'system boundaries' to a sociopolitical notion such as 'justice'). But this is key to the article, a central point of which is precisely to expand a set of measurements from the natural sciences into the social.

For example, at 135-8 the authors articulate their contribution in terms of providing 'safe' and 'just' boundaries in the 'same units':

"The [planetary boundaries] identify only safe biophysical boundaries. Social goals quantified in other approaches are not expressed in comparable units or examine only the consequences on the Earth system of human activities, not harm from Earth system change." (See also 63-71).

If this is the objective, however, I fear the article is not altogether successful. To invoke an 'earth system' is not enough in itself to bridge between natural and social sciences—and it is not clear that the term ('earth system') survives the transition intact. Is the 'biophysical' earth system the same 'system' as the 'justice' one? Which kind of system does 'resilience' refer to? ('Both' would not be a clear answer here ... As the SSPs show – notably SSP1 vis-à-vis SSP5 – it is not necessarily the case that the form 'resilience' takes will be identical in both cases.) The complexities raised by this question are largely unaddressed in this piece.

Justice and safety

As noted, the piece introduces an intriguing distinction between 'safe' and 'just' boundaries, the former referring to the biophysical, the latter to the sociopolitical. This should prove to be a fertile distinction, but in the present piece, I don't believe it quite works. Safe for whom; just for whom? The association of 'safety' with 'tipping points' would seem to indicate that 'safe' is invoked at species level: a planet entering uncontrolled feedback processes is not 'safe' for the human species per se. This may or may not be true—and could certainly be grounded more firmly—but one can accept its hypothetical / rhetorical force. 'Just', then, would operate at a level where the species itself is not (necessarily) at risk, but different groups and communities within it are at more or less risk than others (I am extrapolating here: this is not crystal clear from the article).

If this is right, then much depends on knowing who benefits and who loses in any given scenario—and whether we can assess any one outcome as more 'just' than another. This is a fiendishly difficult exercise and, unsurprisingly, the piece largely shies away from undertaking it. However, the result is that we don't really know why any one outcome would be more 'just' than another in most cases—except, possibly, in broad aggregate terms. (But I am sure the authors are aware that aggregate—ie utilitarian—modes of justice are controversial and contested.)

However my assumption may be wrong: it sometimes feels as though the notion of 'just' boundaries are, like 'safe' boundaries, also invoked at species-level. If so, it seems doubtful that 'justice' is the right term at all, since justice is a fundamentally relational, inter-human, notion.

To expand on the point, the piece speaks (at 109) of three justice criteria: intergenerational, intragenerational and interspecies (although 109 is in fact grammatically hard to follow – see my note below). But none are explained. If the piece is relying on these three criteria, some sort of discussion of how they are supposed to function, and the inter-relation between them, is probably needed. There is a vast literature on ‘intergenerational’ justice, for example. Moreover, an ‘intergenerational’ and ‘intragenerational’ analysis need not result in the same, or even in non-contradictory, conclusions. The ‘interspecies’ criterion too needs fleshing out: it is not at all intuitive—if it is (as it seems) intended to apply between human and all other species, it begs the question as to what humans have to offer other species at all. Unfortunately, words like ‘justice’ are not transparent: they have different meanings in different contexts, and can be used, quite correctly, in very inconsistent ways: the semantic import of a word like ‘justice’ cannot be assumed.

Some consideration is given to this broad set of problems towards the end of the piece, between lines 403 and 446. This is hugely helpful, but I don’t believe it is sufficient, both because the (late) acknowledgement of the existence of this (significant) concern is not in itself enough to neutralise it, and because the discussion is so limited in scope. The cautionary note struck in (most of) that passage seems correct, but it also seems out of synch with the relatively assertive tone deployed in the earlier passages. Plus, I doubt it is plausible to refer to something as ‘just’ without first being upfront as to one’s own position on ‘justice’, which is (understandably) avoided here.

The relative unwillingness to grapple with these kinds of questions in any depth leads to some surprising assertions. So, for example, the authors write at 443-6: "We ... need to address legal, economic, infrastructural, political and social barriers to transformation and promote international collaboration and inclusive governance by multiple actors that practices, incentivises and legislates for a safe and just future". This is a political sentence with all the politics sucked out: it is nearly impossible to identify a ‘we’ that can coherently sit as agent in this sentence, and no hint is given as to how any such ‘we’ could in fact exercise such agency, requiring as a precondition, as it would, an initial global consensus on the set of immensely complicated matters identified just beforehand. To my ear, at this juncture, the somewhat less rigorous ‘social science’ side of the piece seems to have become unmoored from the more rigorous natural science side.

Other imprecise terms

The problem of inadequate definition and precision of terms arises frequently. So, for example, in the first passage cited above, the term ‘social goals’ seems key, but it is hard to parse: what does it refer to? Whose social goals? Again these are matters of significant dispute, with irreconcilable views existing across countries and disciplines (a point implicitly made at 408-410). The whole idea of ‘social goals’ at global level is fraught for all sorts of reasons, not least the absence of a governing body capable of defining what they might be and how to achieve them. Even at national level this kind of problem can prove intractable (a point implicitly made at 437-9).

For the purposes of an article such as this (and following the IPCC’s lead), the SDGs might be adequate as proxy ‘social goals’ and indeed they get a mention (132-4, 175, 420-4). This is not a perfect solution as the SDGs deliberately avoid laying out either the policies that might achieve them or the trade-offs and politics these may entail. In any case, if the article wished to rely on the SDGs—

as having at least international imprimatur—this would still require some active engagement with the substance of the SDGs themselves which does not take place here.

Also in the first passage cited above (which I am using as exemplary), much would seem to depend on the ‘approaches’ for which citation is to footnotes 22, 24-26 (Raworth, Van Vuuren, Hickel, O’Neill et al). These are solid citations—Raworth, Hickel and Van Vuuren are distinguished authors whose work is important and well-known—but it is not obvious how they relate to one another (they do very different kinds of work) nor how they can ground the ‘social goals’ referred to here without quite a bit more explanation.

For this reader the use of several other terms besides—change, transformation, resilience, stability, and ‘equitable’—is also hard to follow.

From this perspective, the core criterion settled on—‘no significant harm’ (NSH)—is emblematic. It is unclear what is meant by this expression, but it seems exceedingly unlikely that, for example, an immediate stop to greenhouse gas emissions (as apparently contemplated) would not result in ‘significant harm’ to many people in much of the world. Indeed arguably, at least a billion or so already live in circumstances entailing ‘significant harm’: what happens to these people if our mitigation efforts outstrip any substitute—or (more likely), if a new ‘green economy’ is largely limited to the rich world?

The authors are alive (see 380-381) to the differing equity stakes for current/future generations in regard to rapid v slower mitigation action, which also applies to local v. global action, but the whole matter remains very vague. For example, (180-184):

"We conclude that to minimise exposure of hundreds of millions of people to significant harm, the just (NSH) boundary should be set at or below 1.0°C. Since returning within this boundary may not be achievable in the foreseeable future, adaptations and compensations to reduce sensitivity to harm and vulnerability will be necessary."

The implication here is that some sort of trade-off is needed between the incremental rises above 1°C to which we are now committed, and the amount and kind of ‘adaptation and compensation’ needed in response. As a matter of ‘justice’, this trade-off is complex and steeped in politics—though this is arguably where the action is (or ought to be) in climate negotiations. Distributional questions like these are theoretically complex because our response tends to depend on our presuppositions about just distributions (as noted at 408-410). They are also practically complex, as they depend on the kinds of institutions in place now and how these might change. The authors do recognise the centrality of these questions (at 408-412, 426-7 and 437-446), but it seems clear that they regard them as falling outside their scope of analysis. The question, then, is how to approach/manage them.

Recommendation

I imagine this article can make a significant contribution, but suggest it should curb its ambition somewhat, play to its strengths and state its limits upfront, indicate clearly its interest in connecting

the natural and social sciences, and clarify the scope of its intervention—focusing on the ‘biophysical’, drawing in the sociopolitical, but noting that whereas ‘safe’ boundaries raise the problem of a broader ‘justice’ calculus, the latter will require significantly more programmatic work. The piece might build on the remark at 459, that it is merely a first step in a larger project attempting to knit ‘social’ and ‘physical’ sciences together. Indeed, I wonder if a little zooming out at that point would be useful too: the challenge is not merely empirical, it is also epistemological.

I am unsure whether this piece is R&R given the currently wide recourse to an overbroad and underdetermined vocabulary. However, if R&R were recommended, it would require, it seems to me, at a minimum a much closer discussion of what is intended by ‘justice’, the three justice criteria, and ‘social goals’, ideally cognizant of the differences between the registers, assumptions, and evidentiary bases between the natural and social sciences. Some of the less central vocabulary (Anthropocene, global commons) could be removed and other terms more precisely defined. The link to the SDGs could be more focused and grounded. I would also tentatively suggest the discussion at 403-446 be moved up and expanded, and the nuance and circumspection in that passage injected throughout the analysis.

I additionally include below a number of more or less ad hoc reactions to specific passages in the first pages of this article.

63: It is not clear why the article begins with the words ‘In the Anthropocene.’ Is there something specific about being ‘in the anthropocene’ that entails the rest of the sentence? This seems unlikely, but if it is the case, the authors ought to tell us why.

77: ‘Humanity is 70 years into the Anthropocene’. The authors won’t need me to point out that the process of dating the ‘anthropocene’ appears to have run aground – essentially as it has apparently run into a performative contradiction: it is unclear how we can determine a geological stratigraphic signature within a 70 year timespan. Moreover the 70-year date intends to ground the anthropocene in radioactive signatures to do with nuclear tests, which does not appear among the authors’ concerns, whereas intensive fossil fuel use, which does, rather aligns with 160 years of activity. I fear the opening sentence is neither clear nor epistemologically sound: it doesn’t kick the article off well.

80: ‘undermine all life-support systems’. This is extremely vague and feels like hyperbole. All life-support systems? Everywhere? For all forms of life? There is no clear citation here but I doubt this sentence can be correct.

84: ‘a precondition for equitable human development is a stable and resilient Earth system’. Whereas I imagine this sentence may be correct, it is not easy to follow. What is meant by ‘equitable’ here? Under what conditions would ‘human development’ be ‘equitable’ and have they ever been met (I imagine this is a question of geographical scope).

85-6: ‘without equitable human development a stable Earth system may not be possible’. I have no idea what this is supposed to mean or why it would be supposed to be true? It crosses between the physical and social sciences without explanation. I am all for ‘equitable human development’ but I

don't think we achieve it by presuming it is needed for a 'stable earth system'. Again there is no attempt at grounding or citation.

109: 'we use three Earth system justice criteria (intergenerational, intragenerational and interspecies & Earth system stability)'. An Oxford comma may help with this grammatically troubled sentence, but is the implication that 'intragenerational and interspecies' is one 'justice' criterion and 'earth system stability' another? Or does 'interspecies' go with 'earth system stability'? Very hard to follow what either refers to – and does not become clearer.

127: 'Earth system justice'. This is an interesting and ambitious term – which I assume is intended by the authors as itself a contribution to debate. Perhaps the piece needs to be foregrounded and carefully defined at the outset. What is 'earth system justice'? The text then says 'which must also enable access to resources for all and distributional and procedural fairness'. I understand the article is not attempting to enter into these criteria – but it is worth noting that these are immense but very vague ambitions, imposing a very high (if laudable) bar. Might the authors not worry that if the objective is unattainable in practice, the overarching argument becomes that much easier to dismiss?

149-50: 'assuming that all safe boundaries by definition meet our interspecies justice and Earth system stability criteria'. This seems a rather significant assumption, certainly in the former case (the latter sounds tautological?)!

174-5: 'the widely accepted "leave no one behind" principle'. I am not aware that there is such a 'widely accepted' principle: to the contrary, most policies by most states seems to assume the reverse. At a minimum a citation would be helpful here.

179: 'committed by greenhouse gas emissions' a typo I assume.

Referee #7 (Remarks to the Author):

In this paper, the authors of Earth Commission propose five safe and just Earth system boundaries (ESBs) for climate, the biosphere, freshwater, nutrients, and air pollution. These proposed ESBs represents new specific targets of decarbonation and conservations in additional to the Paris Agreement of keeping the planet from warming more than 1.5°C above pre-industrial levels.

My major concern of the paper is the proposed ESB for the biosphere with "50-60% of global land surface covered by intact natural areas", which needs a global scale of land reallocation and environmental governance without precedent in human history (Ellis, 2019). In addition, to achieve "50-60% natural areas" requires displacing land from agriculture, which has great consequences for food security and can cause significant harm for some nations or sub-populations (Mehrabi et al., 2018). Therefore, this "50-60% natural areas" ESB would cause significant harm for vulnerable populations and is not a just (no-significant-harm) ESB.

My other concern is the proposed ESB of annual mean interhemispheric AOD difference <0.15 with regards to the aerosol forcing on global monsoon precipitation. The two cited references (citation 74-75) do not provide direct estimate of monsoon precipitation reduction with the scenarios of mean interhemispheric AOD difference <0.15 . More importantly, the monsoon precipitation responses are highly sensitive to latitudinal distribution of aerosol forcing (Fasullo et al., 2019). In fact, citation 75 reports that “the summer monsoon precipitation over India decreases by about 21% for 15° N and 29% for 30° N injections” (Krishnamohan and Bala, 2022). Therefore, the ESB of annual mean interhemispheric AOD difference might be larger or smaller than 0.15 depending on the latitudinal distribution of AOD in each hemisphere.

Reference:

Ellis, E.C. (2019). To Conserve Nature in the Anthropocene, Half Earth Is Not Nearly Enough. *One Earth* 1, 163-167.

Fasullo, J.T., Otto-Bliesner, B.L., and Stevenson, S. (2019). The Influence of Volcanic Aerosol Meridional Structure on Monsoon Responses over the Last Millennium. *Geophysical Research Letters* 46, 12350-12359.

Krishnamohan, K.S., and Bala, G. (2022). Sensitivity of tropical monsoon precipitation to the latitude of stratospheric aerosol injections. *Climate Dynamics* 59, 151-168.

Mehrabi, Z., Ellis, E.C., and Ramankutty, N. (2018). The challenge of feeding the world while conserving half the planet. *Nature Sustainability* 1, 409-412.

Author Rebuttals to Initial Comments:

Response to reviews for Rockström et al., *Safe and just Earth system boundaries*

We thank the editor and reviewers for their constructive comments that have improved the manuscript (included in this document in **bold text**).

- Below, please find our responses to the reviewers' comments (in regular text) and detailed descriptions of the corresponding changes we have made to the manuscript (in underlined text).
- We have attached versions of the manuscript and Supporting Methods with and without tracked changes. Please note that line numbers mentioned in our responses below refer to the version of the manuscript *with* tracked changes. Reference numbers in the main text are correct but may appear out of order due to how our referencing software manages tracked changes. This will be corrected in any final submission.

Table of Contents (referee expertise as listed by the Editor)

Editor	2
Referee #1: biodiversity, habitats	6
Referee #2: hydrology	13
Referee #3: nutrient cycles	15
Referee #4: carbon cycle, climate	23
Referee #5: aerosol, climate	27
Referee #6: justice	30
Referee #7: cryosphere, climate	44

Referee #1: biodiversity, habitats

This manuscript updates the planetary boundaries, which now are named earth system boundaries. The two main new developments is the assessment of the safe boundaries at the sub-global scale and linking them to the previous global boundaries and the adoption of Earth system justice criteria (intergenerational, intragenerational and interspecies) to make them "just". I think in some aspects this is an improvement of the previous planetary boundaries, but this improvement is more of a refinement than a completely novel proposal. There is a lot of work behind the new concept as can be seen in the supplementary material, but I [feel] that without substantial revision the paper may contribute to the uncritical reinforcement of the planetary boundary ideas. Below I explain in more detail.

General Comments

1. There has been many criticisms of the planetary boundaries concept. See for instance a review by Biermann and Kim (2020) *Ann. Rev. Env. Resour.* <https://doi.org/10.1146/annurev-environ-012320-080337>. This paper seems to mostly ignore the criticisms, both in the introduction and in the discussion. This is a missed opportunity. First, because one of the developments in this paper (the connection between sub-global and global boundaries) addresses one of the main issues with the previous concept (the inability to downscale the concept). This issue is alluded to in l.138-141, but more should be said.

Response: We highlight the sub-global advance, as well as other ways in which we advance over previous work, in the Introduction (lines 162-163). We have now added citations to previous critiques that our advances address (Heistermann et al.; Biermann and Kim, line 163), in particular that Earth system boundaries account for sub-global scales and local level health standards. We also elaborate on the sub-global advance at lines 468-473.

Second, because it continues to frame the boundaries as scientifically determined instead of recognising that they are to a large extent judgement calls by a group of experts, and that a different group of experts, or even a group of policy-makers, may want to make a different call.

Response (see also response to editor above): We acknowledge that any scientific assessment will involve some subjectivity. However, we have taken several steps to ensure the ESBs are scientifically rigorous. Below, we describe these steps and our edits to clarify them in the manuscript.

- Our analysis is founded on a rigorous evidence base. For example, the sub-global boundaries for water, N and P are based on well-established empirical results that we then computationally scale up into global boundaries. Internationally established health standards inform many of our just boundaries such as N and aerosols. Other boundaries

are based on systematic reviews, such as climate tipping points. We report the results of these reviews in a way that avoids judgement calls – see next point.

- A major advance in this manuscript is that, where possible, we use scientific analysis to determine multiple likelihood levels for safe boundaries. For example, for climate change we identify boundaries at 0.5°C for low likelihood of passing multiple climate tipping points; 1°C for moderate likelihood; 1.5°C for high likelihood and 2°C for very high likelihood (also supported by other evidence streams from biosphere functioning and palaeoclimate variability). For communication purposes, we then select one of these levels (1.5°C for the safe climate boundary). We acknowledge that other actors may have different risk tolerances and choose other boundaries. Our analysis (as summarised in Table 1) gives them the evidence to make these decisions.
 - Lines 162-163: added “at multiple likelihood levels”
 - Lines 473-476 (Discussion 2nd para): added “We also acknowledge that other actors may choose to implement targets based on other likelihood levels than those we have highlighted (Table 1, Figure 1); for example a lower risk tolerance than the high risk of passing multiple tipping points associated with a 1.5°C safe boundary.”
- We self-assess the confidence in our boundaries, and find that most are only ‘medium’.
 - We have now highlighted our confidence self-assessment more prominently, including: elevating our self-assessment of confidence into Extended Data Table 2 and including the confidence level when we report each boundary in the Results section
- Our nomination process for the Earth Commission and its Working Groups, which was managed independently by Future Earth, has facilitated inclusion across gender, geography and scientific expertise (see Ethics and Inclusion Statement).
- To summarise all these points, we have added to the top of the Methods section (lines 972-978):
 - “We acknowledge that any scientific assessment will involve some subjectivity. We have taken several steps to ensure the scientific rigour of our ESBs: (1) our analysis is founded on a rigorous evidence base (see below and Supplementary Methods); (2) where possible, we determine ESBs at multiple likelihood levels (for climate change, 0.5°C for low likelihood of passing multiple climate tipping points, 1°C for moderate likelihood, etc.; Table 1); (3) the nomination process for the Earth Commission and its Working Groups was an independent process managed by Future Earth (see Ethics and Inclusion Statement); and (4) we report the confidence in our ESB assessments (see below and Extended Data Table 2).”

2. There has been a lot of discussion on whether tipping points or thresholds can be consistently found in ecological systems, see for instance Hillebrand, H. et al. Nat. Ecol. Evol. <https://doi.org/10.1038/s41559-020-1256-9>. So defining 50-60% of global surface as natural ecosystem area is, in my opinion, mainly a judgement call. But this is never stated as such, but instead as justified by the literature (SM). Similar statements can be

made about most of the boundaries - they are inspired by some scientific studies but they are not determined by science but by a group of experts opinions, and there are large uncertainties around them.

Response:

- On tipping points: We support the assessment by Lade et al. (<https://www.nature.com/articles/s41559-021-01481-5>) that Hillebrand et al.'s analysis is actually consistent with the existence of widespread tipping points. Nevertheless, the presence of tipping points is not required to set boundaries. We also use evidence on other, reversible, degradations in functions to set boundaries. In the case of the natural ecosystems area boundary, we use evidence on the largely intact area required to maintain well-functioning carbon, water and nutrient cycles and to halt species extinctions including the preservation of wild biodiversity.
 - We have added text to clarify these points: "We do not exclusively rely on tipping points for setting safe ESBs, however, and the ESBs should not be interpreted as representing tipping points." (lines 112-113)
- On "judgement calls" generally across our work: Please see our response to the previous comment above.
- On the "judgement call" for biosphere in particular: We have added extra text to clarify the evidence from which we arrived at the 50-60% boundary:
 - Main text, lines 232-238: "Based on climate, water and species conservation model outcomes, we propose a safe ESB of 50-60% (medium confidence, Extended Data Table 2) of global land surface covered by largely intact natural areas to maintain Earth system NCP (Table 1, Supplementary Methods). This range uses the current area of natural land cover as a minimum value while indicating the need to restore largely intact natural areas. The exact safe boundary depends strongly on the demand for specific ecological functions (which in turn depend, for example, on the remaining carbon emissions to be sequestered) and on the spatial distribution of the largely intact natural area across ecoregions and ecosystems. Studies generally indicate that up to 60% of the terrestrial earth surface area may be needed, with some extending up to 70% (Supplementary Methods)."
 - See also new text in Supplementary Methods section 1.2.1.
- On uncertainties: We acknowledge that there are large uncertainties associated with many of our quantifications. We report these uncertainties (in some cases characterised as likelihood levels, see response to previous comment) in Table 1. We additionally perform a qualitative self-assessment of confidence in our results in line with IPCC definitions.
 - The results of this confidence analysis are now presented in Extended Data Table 2 and more prominently throughout the Results section.

3. It would be interesting to quantify those uncertainties and summarise them in Figure 1. They are likely to differ across the different boundaries and this is in itself interesting.

Response: As described in our response to the previous comment, we have quantified uncertainties associated with each boundary (Table 1). We visualise these uncertainties/likelihoods in Extended Data Figure 1 (the short concentric lines), but after much internal debate found they clutter the figure too much for presentation in our main Figure 1. We welcome advice on this decision and alternative ideas for visualising uncertainties.

4. I was not fully convinced on the soundness of the "just" analysis with the three Earth System Criteria. They seem very general principles that require some more detailed socio-economic analysis to be applied to the determination of boundaries. Overall I found it difficult to understand how these principles were applied (the supplementary methods did not clarify my doubts). For instance, there are statements like in I. 290 "their implementation should complemented with justice concerns around distribution and quality", or I.2390-240 "additional interventions often needed" which seem vague.

Response: Thank you for raising these concerns. We have made extensive revisions to clarify our justice analysis. Please see new text in the Introduction (lines 117-126), Results (lines 174-182), Methods (lines 1129-1186) and a new table, Extended Data Table 1. Please see our responses to Reviewer 6 for further information. We have also edited the specific lines that the reviewer noted above:

- Lines 349-350 (formerly line 290): "We align our just (NSH) ESBs for water with the safe ESBs, while noting that adhering to the boundaries would considerably restrict current use and will require policies to ensure distributive justice."
- Lines 283-286 (formerly lines 239-240): "Specific interventions that secure functional integrity are highly local and are best implemented under local authority, knowledge and leadership, with policy interventions often needed to ensure that marginalised groups are not further disempowered but rather are given the space to use their knowledges and approaches to participate in such processes."

5. It is not clear what is meant by intact natural ecosystems. In I.213 is stated that they do imply protection excluding human inhabitation and sustainable use. Later it is clarified that indigenous populations are "ok" in these ecosystems. However, there may be other forms of sustainable use which do lead to the tipping points which are addressed by this boundary, and depending on how one defines "intact natural ecosystems" we may not have surpassed it.

Response:

- We have added a definition of natural ecosystem area to the new Glossary in the Supplementary Methods; thank you for the prompt to clarify this term.
 - Text added to Glossary: "Largely intact ecosystems are those where the ecological function and species composition largely resembles those of undisturbed ecosystems. Light human use that does not significantly affect these functions, for example low density population subsistence use by indigenous communities, is not excluded."

- We consistently indicate that intact natural ecosystems *do not* imply protection excluding human habitation and sustainable use; we have modified the sentence to make this clear.
 - Modification to main text (line 256-259): “We emphasise that natural ecosystem area includes all largely intact natural areas and not only those currently requiring conservation attention, and therefore does not imply protection that excludes human habitation and sustainable use.”

6. In the water boundary I was surprised not to see a mention to the barrier effects of dams. Changing of the flow volume and timing are important disturbances to river ecosystems, but the barrier effects on migrating species are very serious and I would like to see it somehow mentioned or acknowledged.

Response: We thank the reviewer for this comment and we agree this is an important aspect to discuss. We have added an additional paragraph to the Supplementary Methods (section 1.3, two paragraphs above start of 1.3.1) focused on the barrier effects of dams, and how they impact both species migration and sediment and nutrient fluxes.

Specific Comments

I. 249 Can you briefly elaborate on what empirical evidence supports the 20% flow alteration principle?

Response: There is substantial literature that supports the 20% flow alteration principle – unfortunately too much to succinctly cover in the main text. Instead, we have added multiple paragraphs reviewing this evidence to the Supplementary Information (Section 1.3, paragraphs 3-6).

I. 312-315. It would be nice to have a simple intuitive explanation of the N/P limits.

Response: We have rephrased and rearranged this paragraph to make the basis of the global N and P ESBs clearer, in particular their relation to remaining within critical concentration in water & air:

- (Lines 368-380): “We set safe ESBs for agricultural nitrogen (N) and phosphorus (P) surpluses for minimising eutrophication of surface water and terrestrial ecosystems due to runoff, leaching, and atmospheric N deposition via ammonia and nitrogen oxide emissions (Table 1). We propose safe global-scale ESBs of 61 TgN/yr for agricultural nitrogen surplus and 4.5-9.0 TgP/yr for cropland soil phosphorus surplus, based on calculating subglobal and global agricultural nutrient losses, surpluses, and inputs that remain within critical N and P concentrations in water and air (Methods, Supplementary Methods). These ESBs primarily relate to agriculture, which accounts for ~90% of anthropogenic N/P inputs to the Earth system. Our ESBs are based on recent work that focuses on agricultural surplus and losses, though for comparison with previous planetary boundary quantifications we also provide corresponding global inputs

assuming current N/P use efficiency. These recent studies also account for non-agricultural sources, assuming they remain at current levels, and the redistribution of nutrients from over-fertilised to under-fertilised regions (Supplementary Methods)."

I. 425-425. This statement on capping the top percentiles of the world's richest is not backed with any data/analysis. Would be interesting to have an idea of what exact percentile we are talking about and how that cap would work.

Response: We are now in a position to add quantitative results recently published by some of us (Rammelt et al., see reference in main text). We have added (lines 504-506): "the impacts of consumption by the world population's richest 1-4% equals the impact that would come from achieving access to resources for those living below the minimum²³" A discussion of how that cap would work is beyond the scope of this article.

Figure 2. Exposure is not necessarily impact. This distinction should be highlighted in the text.

Response: We agree with the importance of this distinction. We highlight throughout the text the relationship between exposure and vulnerability. See for example:

- Lines 142-144: "We focus on assessing levels of Earth system change leading to widespread exposure to significant harm, which will lead to greater impacts when vulnerable populations are exposed."
- Lines 592-593: "Analysis of significant harm needs to integrate how it is exacerbated by poverty and vulnerability"
- We have also added items on exposure, vulnerability and harm/impact to the Glossary in the Supporting Information to further clarify this relationship.

Figure 3. I don't fully understand the criteria used for climate here (one day >35C). Maybe consider other criteria being used in the last IPCC assessment?

Response: Wet-bulb temperatures over 35°C is an increasingly accepted indicator for human heat stress, including by IPCC WG2. Wet-bulb temperature is a method of measuring temperature that incorporates humidity, with direct impacts on human health. We have added to the manuscript:

- (line 1222) A reference to the IPCC chapter that supports this metric, in addition to the paper that the IPCC itself cites (Im et al. 2017)
- Additional clarification of the "one day" metric (lines 1222-1224): "An average one day per year over this temperature per year is therefore a conservative indicator to assessing human exposure to heat stress, which does not account for annual variability."

SM 1.2.1 The 15% CBD restoration criteria was a policy goal and not really "scientifically justified". And the Strassburg study does not really look at thresholds.

Response:

- We do not base our boundary on the CBD's criteria. We mention CBD only as an observation that our independently determined boundary is consistent with it.
 - We have added a note to the Supporting Methods (section 1.2.1, end of second paragraph) to clarify this: "we emphasise that this comparison is an observation and do not use the CBD goals as part of the evidence for this boundary".
- As detailed in previous remarks above, thresholds in the sense of tipping points are not necessary to set boundaries, though for other boundaries we do use tipping point evidence where available.
- We have added extra material to the Supporting Methods (section 1.2.1) to support the natural ecosystem area goal, see response to this reviewer's second comment above.

SM. 2.4.2.1 The paragraph that starts with "However, we note that extra attention must be given to I2". I really don't understand how current generation needs to have costs so that future generations can benefit from functional integrity. I think a goal of 10-25% of natural elements in agricultural landscapes would also benefit current generations through ecosystem services.

Response: Thank you for identifying that this passage was unclear. We have added to SM 2.4.2.1: "In the specific case of integrity loss, both current and future generations of people and other species can benefit from meeting the 20-25% integrity boundary."

SM. I missed a section 1.2.2 on "Functional Integrity"

Response: Our apologies. Some material fell out during the final stages of editing. We have now re-inserted and further elaborated this text (see Supplementary Methods section 1.2.2).

Referee #2: hydrology

I read the manuscript several times, more often than in general, but always with a similar conclusion: I miss the innovation of this manuscript and suggest to decline it.

This manuscript reads like a review paper with 144 references to be found in the main text and methods plus the references in the SI. The overall method has not really changed to what has already been published or is not new at all. What makes the difference to former publications?

Response: Our manuscript contains at least two major advances: (a) incorporation of justice considerations into boundary-setting and (b) specification of many boundaries at sub-global scale. These advances are described in the Introduction of the manuscript (lines 150-167).

Point (a), specifying sub-global boundaries, is an important advance because it overcomes many critiques of the surface water planetary boundary, for example that a global boundary is not a hydrologically or operationally relevant scale.

- We now reference some of these previous critiques including Gleeson et al., Zipper et al., Biermann and Kim, and Heistermann (line 162).

This paper is a synthesis of research across many domains of the Earth system, which we acknowledge gives the impression of a review paper. Some of the research we rely on has been previously published, while some is new to this paper. Furthermore, the act of drawing together boundaries from so many different biophysical domains under a novel framework allows for a systematic appraisal of the risks posed by undermining our planet's biophysical systems.

Setting a global alteration budget instead of global sum of water abstractions or consumption does finally not lead to something new. In global scale modelling, a limiting alteration of flow of 20% is (often) used as an indicator for environmental flow requirements. We know that this indicator is too broad and not specific enough to mimic aquatic ecosystem needs. There are more suitable approaches available on the global scale to determine environmental flow requirements (e.g. Pastor et al. 2014). Liu et al. 2021 compared the effect of different EFR approaches on water scarcity which cannot be neglected. However, it is difficult to find a suitable indicator which quantifies the ESB of a renewable resource. It is feasible when accounting for water quality in terms of quantities (consideration of loads and concentration) instead of sanitation information solely.

Response: Our interpretation of the eco-hydrological literature is that the general findings are moving away from EFRs such as Pastor et al. towards a stricter 20% alteration requirement. Pastor et al. allow up to 60% alteration during the high flow season: levels of flow alteration that have been shown to have significant impacts on aquatic ecosystems and the ecosystem services they provide. In the most recent planetary boundary for water, Gerten et al. (see Supporting Methods reference list) adopt the approach of Pastor et al. and themselves sum these local ESBs into an indicative global budget that can be compared with previous planetary boundary quantifications.

In light of the reviewer's concerns, we have:

- Added multiple paragraphs to the Supporting Methods (section 1.3, paragraphs 3-6) to detail the evidence supporting the 20% flow alteration requirement. We show that the less stringent requirements of Pastor et al., such as up to 60% alteration during the high flow season, allow significant impacts on aquatic ecosystems and the ecosystem services they provide. Pastor et al. themselves acknowledged "The choice of environmental flow methods for our study was limited to hydrological methods due to lack of data on ecosystem responses to flow alterations for most river basins of the world."
- Acknowledged in the main text that a 20% requirement is an imperfect rule that is only designed for use where local assessments are unavailable (lines 294-300): "Local-scale flow-ecology analyses are often used to establish environmental flow needs to define safe levels of flow alteration for individual watersheds. These local-scale assessments could provide the basis for spatially explicit safe boundaries, however, unfortunately they are absent across most of the world. In their absence, we propose that a presumptive sub-global safe ESB of 20% alteration (increase or decrease) of monthly surface water flows compared to the prevailing natural flow regime be met in all rivers globally."

Referee #3: nutrient cycles

The manuscript examines a series of Earth System Boundaries (ESB) related to fundamental processes governing the future of the planet. More specifically, the manuscript assesses what could be the safe and just values for each ESB and to what extent these safe and just boundaries have been exceeded both globally and regionally. Safe boundaries are defined in the perspective of maintaining the resilience and stability of the Earth System, while just boundaries are defined to minimise exposure to significant harm to humans from Earth System change (in some environmental justice perspective). The manuscript builds upon previous studies – mostly related to the planetary boundary framework – by (i) revising and updating the safe boundaries previously proposed and (ii) providing some just boundaries for each Earth System process considered.

The novelty brought by this manuscript is strong. One noticeable added-value of this manuscript is related to the definition, quantification, and mapping of what could be the just values for each ESB, beyond the previously published safe values. Because this study has a global perspective and because it addresses a large set of Earth System processes – related to climate, the biosphere, water and nutrient cycles, and aerosols – it is likely to meet a global audience.

I consider this manuscript acceptable for publication in Nature due to its large scope, strong novelty and robust methods. However, I consider that significant, major improvements have to be undergone before its publications.

First, the foundations for defining the just boundaries were overall unclear to me. I must say that I am not a specialist of fairness and justice concepts but I consider that some overall clarifications are needed on this for the broad audience of the Nature journal. More precisely, although I could understand the general rationale, I could not fully understand what the authors mean by using “three Earth system justice criteria (intergenerational, intragenerational and interspecies & Earth system stability) to assess whether adhering to the safe ESBs could protect people from significant harm” (line 109). How is inter-generational justice considered here? Is this somehow connected to long-term impacts of Earth System changes?

Response: Yes. In the revised manuscript, we consider both the impacts of past generations on current generations (I2a) and current generations on future generations (I2b) due to long-term impacts of Earth system changes. We have now

- Elevated the introduction of intergenerational justice (and the other justice criteria) to the opening paragraphs of the manuscript (lines 117-126): “Second, we use three criteria to assess whether adhering to the safe ESBs could protect people from significant harm: ‘interspecies justice and Earth system stability’ (I1); ‘intergenerational justice’ between past and present generations (I2a) and present and future generations (I2b); and ‘intragenerational justice’ (I3) between countries, communities and individuals through an

intersectional lens. These criteria sit within a wider Earth system justice framework that goes beyond planetary and issue-related justice to take a multi-level transformative justice approach focusing on ends (boundaries and access levels) and means (Supplementary Methods).”

- Designed a new table that will appear in the online main text (Extended Data Table 1) that explains how we apply the justice criteria. For intergenerational justice, this includes a separate explanation for each Earth system domain.

What does interspecies mean? How Earth System stability is related to justice? How does all this relate to environmental justice?

Thank you for the prompt to clarify this point. Interspecies justice concerns the natural world other than humans. We couple together ‘interspecies justice’ and ‘Earth system stability’ in criterion I1 because we assume that our safe Earth system boundaries deliver interspecies justice. We have added new text at:

- Lines 1131-1135: “Our research into interspecies and multispecies justice reveals details regarding the scholarly approaches to these concepts but there have been no attempts to operationalize these concepts deductively. In our research we have combined interspecies justice with Earth system stability and inductively identified, through a domain-specific (e.g. climate, biosphere, aerosol loading) approach, boundaries based on existing scholarship and logic of that domain (Extended Data Table 1).”

We base our Earth system justice framework on environmental justice, planetary justice, and other similar frameworks. We now clarify this point at:

- Lines 123-126: “These criteria sit within a wider Earth system justice framework that goes beyond planetary and issue-related justice to take a multi-level transformative justice approach focusing on ends (boundaries and access levels) and means (Supplementary Methods).”
- For further information, please also see our recently accepted manuscript on Earth system justice (cited in the manuscript; for full text see link on the first page of this reviewer document).

In addition, because just boundaries are defined to minimise exposure to harm from Earth System change, I cannot really understand how this relates to inter-generational justice criteria.

We hope that our new Extended Data Table 1 helps illuminate how we applied the intergenerational justice (justice “between generations”) criterion. Taking climate as an example from that table: current generations experience harm because of the emissions of past generations (I2b). Future generations will experience harm if current generations do not keep climate within an Earth system boundary.

Finally, the way just boundaries articulate with safe boundaries is not always clear. For instance, the sentence line 149 (“assuming that all safe boundaries by definition meet our interspecies justice and Earth system stability criteria”) is very unclear.

Thank you for the prompt to clarify. We have rewritten this sentence to clarify that we take the safe boundaries developed for each Earth system domain to satisfy the first justice criterion (I1):

- Lines 174-175: “Adhering to these safe boundaries implements our ‘interspecies justice and Earth system stability’ criterion (I1, Extended Data Table 1)...”

I know that this has been developed in details for each considered Earth System component or process in the Supp Info but I feel more justification is needed in the Main Text.

Response: Thank you for your comments. As indicated in the responses immediately above, we have added much more justification in the main text, including

- New sentences in the Introduction (lines 117-126)
- New sentences in Results (lines 174-182)
- New paragraphs in Methods (lines 1129-1186)
- A new table (Extended Data Table 1)

Second, although this work brings a lot of materials of great importance for guiding human actions in the Anthropocene, I found the “Translation for actors” and “Transformation to live within Earth System boundaries” sections within the Discussion a bit disappointing. In their current status, those sections are quite general and vague, and may lack the appropriate strength to actually translate the study outcomes to the decision-makers. In particular, I consider that some developments are needed about how the outcomes from this study could feed the global debate about green growth, GHG-GDP decoupling, and degrowth as well as about the future for agriculture and food systems.

Response: In line with similar statements by the Editor, we have severely shortened and rewritten these sections (see lines 478-516 for new versions of these paragraphs). We now discuss the future of agricultural and food systems (lines 514-516) and refer to the need to transform away from a growth paradigm (line 502), but we feel the green growth / degrowth debate is too nuanced to enter into during the short space we have available here.

Third, as requested by the Editor, I provide a series of specific comments about the N and P cycles. Although the corresponding sections are overall fine and the related methods look robust, I consider that several aspects need some improvements:

- **Several key aspects of the global N and P cycles are actually missing in this study:**
 - o **First, nothing is said about N₂O losses and the way they are considered in the definition of the safe and just boundaries. Because N₂O is a long-lived climate pollutant with strong climate warming effects, I would have expected it to be included in the ESB definition. I guess that N₂O is somehow already included within the climate boundary (itself expressed in GMST above pre-industrial level) but nothing is clearly explained**

about this. Some non-CO2 budgets exist – in addition to the CO2 budgets provided by the IPCC – in order to attenuate climate warming and I wonder how these budgets relate to the proposed boundaries for the global N cycle.

Response:

Reviewer 3 is correct to note that N₂O losses are important to consider given their role in climate change and that it is included within the climate ESB, because we quantify climate by global warming, of which N₂O is one of the drivers. However, we did not state this clearly in the paper, and so have now added text on this to the Supporting Methods. Our ambition is that in future work interactions between the ESBs will be more explicit, including whether incorporating N₂O-climate interactions reduces the N ESB, and the impact of increased carbon sequestration driven by N deposition. In anticipation of this analysis, we note that the warming effect from anthropogenic N₂O emissions is uncertain since it may partly be compensated by the cooling effect of additional carbon sequestration in forests induced by enhanced N deposition (as discussed by Schulte-Uebbing et al., 2022).

- (Supplementary Methods section 1.4.1, first paragraph): “N₂O emissions also contribute to climate change (with 273 times the global warming potential of CO₂ over 100 years; Forster et al., 2021), but this effect does not affect setting of the climate ESB, which is specified using temperature change rather than concentrations of greenhouse gases. Our ambition is that inter-boundary interactions will be analysed more explicitly in future iterations. For example, in addition to N₂O emissions increasing radiative forcing, N cycle disruption might also lead to a net-cooling effect via increased carbon sequestration [de Vries et al., 2017; Erismann et al., 2011].”
- (lines 575. 581-582) “Critical areas for further research on safe ESBs include:... (4) Modelling interactions and feedbacks between ESBs that regulate the state of the Earth system”

o Second, I was surprised to read that NH₃ emissions to the atmosphere were only considered in the perspective of N deposition on terrestrial ecosystems. NH₃ air concentrations have also strong effects on human health both directly and indirectly as precursors for aerosols, as acknowledged by the authors on pages 17 and 18 of Supp Info. More details about this and if possible inclusion of the NH₃ effects on harm to humans are expected here.

Response: Ammonia (and nitrogen oxide) emissions can indeed pose a threat to human health via their contribution to formation of particulate matter, as now discussed in the Supplementary Methods and briefly in the main text (see list of edits below). We did not include a safe boundary for NH₃ emissions in view of air pollution though (alongside a boundary for groundwater harm) as this would require detailed information on meteorology, atmospheric chemistry and presence of other PM-precursors. However, it is likely that critical N deposition levels on nature used in deriving ESBs for the N surplus also protect human health due to NH₃ induced PM_{2.5} effects (Schulte-Uebbing et al. 2022). We have clarified this in the Supplementary Methods, and expanded mention of ammonia-derived aerosols in the main text. In future work it would be important to try to incorporate air pollution in the global harm quantification more explicitly.

Edits:

- Line 383-385: “Elevated N and P concentrations cause harm through the consequences of eutrophication on ecosystems and their services, such as fishery collapse, toxic compounds released by algal blooms and the health impacts of air pollution from ammonia-derived aerosols”
- (Supplementary Methods, section 1.4.1, paragraph beginning “Beyond ecological damage...”): “However, there is currently insufficient work quantifying a limit for N-related air pollution with respect to human health at a global scale, although it is likely that NH₃ emission levels ceilings that protect nature also protect human health from NH₃-induced PM2.5 effects (Schulte-Uebbing et al. 2022). As a result, in this iteration we only include harm from groundwater contamination and ecological impacts for the global harm quantification, but in future work harm for air pollution should also be directly incorporated.”

o Third, a major issue about the global P cycle is related to the just use of the globally remaining phosphate rocks. The future of these phosphate rocks is very likely to have severe effects on the global food security for the next coming decades and centuries. I understand that the just definition provided in this study is related to the “harm to humans from Earth System changes” and this differs from the just use of natural resources. However, because food insecurity – partly related to lack of access to phosphate rock and P fertiliser resources – is a significant harm to humans, I would have expected some clarification about how this phosphate rock issue is considered in this study.

Response: Reviewer 3 is correct that rock phosphate depletion is also a source of indirect harm via the threat to food security, as discussed in more detail in Supplementary Methods section 2.3.4. We’ve clarified that this is considered.:

- Line 391-395: “Additional justice considerations include lack of access to N and P fertilisers, which can threaten food security especially for low-income communities and countries; and extraction of phosphate rock, which is a limited resource currently underpinning food production but exposes poor and marginalised communities to mining waste, destroyed land and human rights abuses”.
- Supplementary Methods (last paragraph of section 1.4.2): “A further consideration is that unlike N, the primary source of P for agriculture is a non-renewable resource (rock phosphate). Rock phosphate depletion presents potential inter-generational harm through inefficient usage prior to the establishment of full nutrient recycling (see SM section 2.4.4), while unequal access due to both poverty and geographic distribution is also a justice issue [Elser & Bennett, 2011].”

• The way the safe boundaries have been defined for N and P surplus is unclear to me. I understand that the authors have used some boundaries for N and P concentrations in both freshwater and groundwater as well as N emissions to air, and that they related these water concentrations and air emissions to N and P surplus in agricultural soils

through some sort of modelling but I could not understand the bases for this modelling approach. Much of the Supp Info about this is actually making reference to other articles whereas I think some clarifications about the compartments and process considered is needed here. More specifically, it is unclear to what extent the authors have accounted for strong soil and climate effects on N and P leaching and erosion globally.

Response: We have added more details to clarify the basis of the boundary quantification and consideration of soil and climate effects, both briefly in the Methods section (e.g. clarifying that sub-global N limits are calculated using the IMAGE model as outlined in the original publication by Schulte-Uebbing et al. (2022), whereas P is a simple global budget calculation) and in more detail in the Supplementary Methods. For further details beyond this necessarily brief summary, readers are referred to the cited papers themselves which are now all fully published and available.

Edits:

- Lines 1020-1025: “We mainly relied on a recent study following up previous works that extended the approach of the original planetary boundaries. This study used the IMAGE model to derive sub-global boundaries for critical nitrogen losses, surpluses and inputs based on critical concentrations in air and water, and then aggregated these into global boundaries (see Supplementary Methods for further information).”
- Lines 1026-1029: “For phosphorus, we relied on recent work that used literature-derived critical concentrations for avoiding eutrophication from P runoff to estimate global boundaries for P mined input and surplus based on a global budget calculation, taking into account P recycling, human excreta, soil and sediment retention, and global nutrient rebalancing”
- Supplementary Methods: see new text in the paragraph immediately above Table S5 and in the two paragraphs above Table S6.

• Finally, details are provided in the Supp Info about how just boundaries have been defined and why they align on safe boundaries for P (and partly for N). However, these details are quite buried in the long Supp Info and may not be accessible to most readers. I recommend instead making this more explicit in the Main Text for both nutrients (e.g, around line 321).

Response: We have added text (to the extent that word limits allow) to clarify that the basis for the just ESB largely aligning with safe ESB for N & P is that harm to humans is mostly via environmental degradation.

- Lines 387-389: “We therefore align the just (NSH) ESBs for sub-global N and sub-global and global P with their safe boundaries, as human harm from nutrient cycles disruption is primarily via environmental degradation.”

Minor comments

- **Much of the ‘Methods’, in particular about estimate uncertainties and confidence levels are actually results. I think those uncertainty considerations could be moved to the Supp Info while some key methodology details could be moved from the Supp to the Main Text.**

Response: Thank you for this helpful proposal. We have:

- Moved the full results of our confidence self-assessment out of Methods into Extended Table Data 2.
- Expanded on key methodology details in Methods for N/P and water (lines 1003-1029; see also responses to comments above and below)

- **Some clarifications are needed about why NH₃ emissions are considered in this study: while N deposition is mentioned as a considered boundary line 307 it is unclear how it translates into NH₃ emissions line 801.**

Response: We have added extra text to the MS to clarify that the atmospheric N deposition to terrestrial ecosystems considered is from both agricultural ammonia emissions and NO_x emissions from industry and transport, where we estimate ‘critical’ agricultural NH₃ emissions under the assumption that (non-agricultural) NO_x emissions stay constant.

Edits:

- Lines 368-370: “We set safe ESBs for nitrogen (N) and phosphorus (P) for minimising eutrophication of surface water and terrestrial ecosystems due to runoff, leaching, and atmospheric N deposition via ammonia and nitrogen oxide emissions (Table 1)”.
- Lines 1017-1025: “For nitrogen, we used three regional environmental boundaries: significant disruption to freshwater ecosystems (from total N runoff), groundwater potability (from nitrate leaching), and terrestrial ecosystems (from atmospheric N deposition due to ammonia and nitrogen oxide emissions) across wide areas, based on critical concentration limits for each.”

- **Several sentences are unclear, at least for non-native English people such as me. See for instance lines 69, 117-119, 209-210, 299-302, 535-536.**

Response: Thank you for identifying these. We have edited the manuscript as listed below.

- Line 69: “The stricter of the safe and just boundaries sets the safe and just ESB”
- Lines 135-137: “We show that at 1°C global warming tens of millions of people have already been exposed to wet bulb temperature extremes, and that at 1.5°C over 500 million people could be exposed to sea level rise and 200 million to unprecedented mean annual temperature”
- Lines 252-254: “Some people and countries may directly benefit from policies to maintain or increase natural ecosystem area, while others may face opportunity costs.”
- Lines 359-364: “The state variable for green water is defined as the percentage of ice-free land area that in any month has root-zone soil moisture levels outside the 95th percentile of the local baseline variability. The boundary is set at 10%, corresponding to the median departure level from mid-Holocene conditions.”

- Lines 951-952: “Short concentric lines (that extend across less than the full width a wedge) represent alternative likelihood levels (safe) or levels of exposure (just NSH) (Table 1).”

• **Supp page 15: what does ‘new fixation’ mean? I guess it refers to both biological and industrial nitrogen fixation but please clarify.**

Response: Reviewer 3 is correct that ‘new fixation’ means both biological and synthetic N fixation by humans, and have clarified this in the Supplementary Methods. Also ‘new fixation’ is a doubling so this was reworded to ‘intentional’ N fixation’. Edits:

- Supplementary Methods, paragraph below Figure S2: “In 2009 the Planetary Boundaries framework first proposed a global safe boundary for N pollution beyond which unacceptable global environmental change occurs, setting the boundary for intentional new N fixation (i.e. N fixation for synthetic fertiliser production and biological fixation by leguminous crops) as a first guess well below current rates (~140 TgN yr⁻¹) at 35 TgN yr⁻¹ (Rockström et al., 2009) (Table S7).”

• **Table S6: what does ‘current excess’ mean? I guess it is value above the natural baseline as in Table S8 but please clarify.**

Response: Reviewer 3 is correct that ‘current excess’ means the magnitude of the indicator’s deviation above the natural baseline, and have clarified the table accordingly (Table S6). Edits:

- Table S6: “Current excess above baseline”

Referee #4: carbon cycle, climate

Summary of the key results: This study is an assessment and synthesis of evidence in support of a set of critical thresholds for different compartments and cycles of the earth system (e.g. atmosphere, ecosystems, water resources, nutrients essential for life) that are argued to constitute limits for global society and nature to remain safe from harm from climate change and its impacts, while also upholding fairness (justice) among countries/regions and human generations. It is framed as an extension and elaboration of the Planetary Boundaries framework, introduced by partly the same group of authors a decade or so ago. The Planetary Boundaries concept and language has been influential in science-policy work around climate action and sustainability in recent years.

Originality and significance: As noted, this work extends on the Planetary Boundaries framework (PB) which is already well-established in the climate and sustainability literature. The framework has been criticised on various grounds, but there is no denying it has been found useful by the applied research community, providing a way of linking scientific evidence from impact and modelling studies in a wide range of fields or 'domains' to each other, and to policy goals for climate action (Paris Agreement), sustainability (SDGs) and related areas. The present work is an extension of PB, which limits its originality, but it does include an extensive synthesis of new data and literature of relevance to identifying/quantifying the boundaries, including findings of other major assessments such as IPCC-AR6. Apart from new data, the major novelty probably lies in the juxtaposition of justice alongside danger considerations in setting the proposed boundaries. However, these criteria are already combined in the seemingly very similar conceptual framework of "doughnut economics". The differences between these frameworks are not very clear to me, and may largely boil down to different emphasis – socio-economic factors in "the doughnut", earth system processes in this paper. However, the authors claim there is novelty in that the limits to what is safe and what is just are expressed in the same units (e.g. degrees of warming for climate) in their framework, aiding the setting of policy targets, for example. The inclusion of justice criteria addresses one criticism of the original PB concept, that it fails to explicitly consider opposing impacts on people and nature in different world regions, with the potential for perverse outcomes if policy frameworks are set around a single global target (such as the Paris Agreement's <2 degrees of warming relative to pre-industrial climate). Overall, I would rate this as a significant piece of work that will influence environmental and sustainability research and science-policy work going forward.

Response: Thank you for your assessment of our manuscript as a "significant piece of work that will influence environmental and sustainability research and science-policy work going forward". In response to your and other reviewers' comments, we have clarified how our work's treatment of justice advances over the doughnut framework. In addition to quantifying just and safe boundaries in the same units, we examine significant harm from Earth system change which doughnut economics does not do:

- “Social goals related to access to or harm from natural resources quantified in Agenda 2030, doughnut economics and other approaches are not quantified in comparable units or examine only the consequences on the Earth system of human activities, not harm from Earth system change” (lines 155-158)

Data & methodology: validity of approach, quality of data, quality of presentation. This is a synthesis and assessment combined with tweaks to an existing conceptual framework, so the methodology mainly comprises identifying, distilling and interpreting relevant literature to come up with numbers for the ‘proposed safe and just earth system boundaries’ as summarised in Table 1. Details of the literature consulted and principles for assigning values to boundaries are provided in the 55 page Supplementary Methods. This extensive material covers a vast diversity of fields, making it challenging to judge the adequacy of this effort in terms of fairly capturing the state of knowledge within each of the considered domains. It is clear that it reflects a substantial body of underpinning work, and does seem to embrace other recent assessments and synthesis.

Focusing on the fields into which I have the best insight, I looked more closely in the supplement at how boundaries for climate change (including related feedbacks) were defined and quantified. In the Methods the general approach is described thus: “We used two main groups of approaches to setting safe earth system boundaries: a ‘multiple elements approach’ and a ‘spatial aggregation approach’.” For climate change, the chosen criterion for deciding what level of change becomes dangerous was whether “multiple tipping points are crossed”. I suppose this is the ‘multiple elements approach’ in action, but the idea seems to come from a paper currently in preprint (Armstrong McKay et al 2021, S1 References). Without consulting this paper, I couldn’t find a clear description of the basis for which crossing one tipping point should be considered safe whereas two or more are deemed dangerous. It could be argued that one tipping point is enough to be worried. Figure S1 (seemingly reproduced from the paper above) displays the assessed tipping points with pseudo-uncertainty estimates shown. The safe boundary is identified at 1 degree of warming, since at this level, committed loss of the Greenland ice sheet is the one allowable tipping point. This figure is useful for understanding, and the data it displays seem broadly credible, acknowledging substantial knowledge gaps around the likelihood and size of many of these climate system features/impacts, or indeed whether they represent true tipping points. For example, (degradation/loss of the) Amazon rainforest due to runaway evapotranspiration-rainfall feedback has been much debated and demonstrated using some models and not by others. There is no consensus this is a real tipping point at all, or if it is, at what level of climate warming it can be expected to kick in. However, I accept an assessment like this has to capture ‘best current understanding’, and the wide uncertainty bounds around Amazon loss and other features in Figure S1 capture this in a seemingly reasonable way. By the way, some features in this figure are lacking a legend or clear explanation in the figure text, including the “SSP projections” at far right, and the horizontal line and shading around 2-4 degrees.

Response: We're pleased that Reviewer 4 broadly supports our approach to quantifying a safe ESB for climate.

- We'd like to note that the supporting paper for the climate tipping elements assessment, which was accepted in *Science* when we submitted our manuscript, has now been published in *Science*. As indicated in the text (line 186-188), our safe climate ESB is also supported by literature on palaeoclimate variability and biosphere functioning, which we also compare with the IPCC's latest 'Reasons for Concerns' in the SM. As such, we rely not just on the climate tipping points assessment, but a wide basis of climate science.
- We also agree with the reviewer that of course "one tipping point is enough to be worried". We have chosen multiple climate tipping points as one of our criteria to delimit safe, however, due to the resulting systemic threats to Earth system stability. We have now added the text below and thank the reviewer for identifying that this point was not sufficiently clarified in the previous version of the manuscript.
 - Added text (line 188-190): Triggering multiple climate tipping points, whether individually or through cascades, would systemically threaten the stability of the Earth's climate and biosphere.
- We also note that we identify multiple safe boundaries corresponding to different likelihood levels in Table 1. We have chosen to highlight 1.5°C, for high likelihood of passing multiple tipping points, but a lower risk tolerance would lead to the 1°C boundary that the reviewer highlighted.
- We have also revised Figure S1 now that the paper it is based on has been published, and provided further details in the caption.

Conclusions: robustness, validity, reliability. My main concern is probably with the criteria chosen to define what constitutes an earth system boundary, and the premise that identifiable boundaries exist across each of the domains considered. The authors' criteria for the existence of a likely or possible boundary are given on page 3: "To determine safe boundaries, we use an assessment of tipping point risks among local and regional tipping elements ... components or thresholds that regulate the functioning and state of the planet and show evidence of having thresholds at which small additional perturbations can trigger self-reinforcing changes that undermine Earth system resilience". This suggests that the boundaries considered all represent thresholds at which system-internal feedbacks start to occur, or accelerate markedly, pushing the planet into "regime shift" (Supplement page 2) and a less desirable state. The possible existence of such critical thresholds has probably been argued in the past for at least some impacts or phenomena within each of the chosen domains. But, as noted above, the evidence certainly for some of the suggested tipping points in the climate system are sparse and/or contradictory. Most such evidence tends to come from model studies as opposed to observational trends, and different but similar models often show contrasting outcomes. To put it another way, many of these tipping points exist in theory, but we often lack solid evidence that they occur in practice. If tipping points are uncommon in the Earth system, how meaningful is it to identify safe boundaries this side of the tipping point? The authors do not really argue for their belief in boundaries based on the extensive literature they have reviewed. The boundaries are taken as given, and the

evidence is deployed to argue for the size/value of the boundary. As a contribution to debate around the resilience of the earth system and its current (Holocene) state, arguments could be made clearer. But perhaps this is too much to expect from one paper.

Response: As mentioned in our response immediately above, we do not only consider tipping points in our quantification of ESBs. The full sentence quoted by the reviewer reads (emphasis added):

“To determine safe boundaries, we use an assessment of tipping point risks among local and regional tipping elements, *evidence on declines in Earth system functions, and analyses of historical variability*” (lines 106-108)

and elsewhere:

“Tipping element assessments in climate, biosphere and other Earth system domains are *key, though not exclusive, evidence for our ESBs*” (lines 164-167).

As such, we do not suggest that the ESBs “**considered all represent thresholds at which system-internal feedbacks start to occur**”, merely that for some (but not all) ESBs such behaviour is identifiable in the Earth system and is considered when setting the safe ESB. To clarify that we don't rely solely on tipping points in our analyses, we have inserted the following sentence:

- “We do not exclusively rely on tipping points for setting safe ESBs, however, and the ESBs should not be interpreted as representing tipping points” (lines 112-113).

We also recognise that the evidence for the presence and position of tipping points is varied, as the reviewer points out. This uncertainty is reflected in our self-assessment of medium evidence robustness for the safe climate boundary (now moved into Extended Data Table 2 in response to other reviewers' comments).

Clarity and context: lucidity of abstract/summary, appropriateness of abstract, introduction and conclusions. I found the paper well-written and comprehensive, given the complexity and disciplinary breadth of the work.

Response: Thank you for the compliment!

Referee #5: aerosol, climate

I have reviewed the "Aerosol Pollution" section of the manuscript "Safe and just Earth System Boundaries" by Rockstrom et al. My main comment is that the use of studies analyzing the effects of volcanic aerosol emissions or purposeful stratospheric aerosol injections to support the use of interhemispheric asymmetry in aerosol optical depth (AOD) as a metric for safe ESB is weak. I understand that a more elaborate and nuanced discussion is provided in the supplementary but this is not reflected in the main text and needs to be captured in a more clear and transparent way. The evidence for the role of interhemispheric asymmetry in AOD on hemispheric precipitation (lines 337-339) needs to be predicated on observations in addition to modeling studies. Further, because of aerosol-cloud interactions, I do not expect the response to "an increased concentration of reflecting aerosols in one hemisphere" to be straight-forward (see Douville et al., 2021; Allen et al., 2015).

Response: Thank you for pointing to the fact that we did not sufficiently highlight the observational insights on the AOD difference in the main text as they were only provided in the Supplementary Methods. We have adjusted the sentence to discriminate between the empirical evidence for the West African monsoon rainfall and the climate modelling studies that provide insights for the Indian monsoon patterns. There are uncertainties related to aerosol-cloud interactions in understanding the effects of aerosols on the monsoon systems [Douville et al., 2021]. However, observations based on past volcanic eruptions and climate modelling studies show that an increased concentration of reflecting aerosols in one hemisphere leads to precipitation decreasing in the same hemisphere's tropical monsoon regions while increasing in the opposite hemisphere [Haywood et al., 2013; Liu et al., 2016; Zuo et al., 2019]. We have added this.

Text edited:

- Lines 401-411: "Observational data for the West African monsoon rainfall and climate modelling studies for the Indian monsoon have identified potential shifts in the location of the Intertropical Convergence Zone (ITCZ) triggered by differences in sulphate AOD between the Northern and Southern hemispheres. Observational studies on the impacts of interhemispheric AOD difference on Indian monsoon are lacking but observations based on past volcanic eruptions and climate modelling studies show that an increased concentration of reflecting aerosols in one hemisphere leads to precipitation decreasing in the same hemisphere's tropical monsoon regions while increasing in the opposite hemisphere. Observed changes in the South East Asian monsoon have well understood mechanisms (Supplementary Information) that are consistent with the effects of interhemispheric AOD difference."
- We have added a paragraph based on IPCC on this to the Supplementary Information (second-last paragraph of section 1.5): "The observed reductions in South East Asian monsoon precipitation in the second half of the 20th century and Sahel drought in the 1980s are explained by a southward shift of the ITCZ due to inter-hemispheric temperature differences related to NH cooling caused by increased NH aerosol

emissions. Further, over South Asia, East Asia and Sahel, increases in monsoon precipitation due to warming from GHG emissions were counteracted by decreases in monsoon precipitation due to cooling from human-caused NH aerosol emissions over the 20th century.”

Additional specific comments:

L339-340: But we know that air pollutants (particularly aerosols) are declining (e.g., Quaas et al., 2022) and will most likely continue to decline as human health concerns will lead to more stringent controls on air pollution in developing countries. All SSP scenarios except SSP3.70 project declining emissions of short-lived climate forcers (including aerosols) (see Szopa et al., 2021 Figure). In fact, I would argue that the impact of increasing greenhouse gases on precipitation will dominate over that of aerosols in the long-term (see Lee et al., 2021),

Response: We agree with the reviewer that many scenarios, such as the SSPs quoted by the reviewer, reflect reductions in air pollution. Yet,

- 1) The global aerosol ESB could become relevant in view of some solar radiation management approaches, even when anthropogenic air pollution is solved/reduced. We have thus added this (see list of changes below).
- 2) We suggest ESB irrespective of the current trend as they should never be exceeded.
- 3) Strong reductions in air pollution are possible/technically feasible - as historical experience and many scenarios of future developments show - but strong political will was and will be needed to implement those pollution control measures, so they are not guaranteed (Amann et al 2020, <https://royalsocietypublishing.org/doi/full/10.1098/rsta.2019.0331>; Rao et al 2016, <https://www.sciencedirect.com/science/article/pii/S0959378016300723>).
- 4) Short-lived climate pollutants, such as black carbon as contributing to PM2.5, might go down in many scenarios, such as quoted by the reviewer (Szopa et al., 2021), but
 - a) black carbon is a small part of PM2.5: for example 5-10% [see Yu et al 2014, <https://aaqr.org/articles/aaqr-14-11-0a-0295>]
 - b) Regional analysis of potential air pollution futures consistent with the SSPs (as the SSPs cover climate change mitigation and adaptation challenges and do not specify pollution control in detail) show that in some regions, such as Asia but also Middle East & Africa, the majority of SSP/RCP combinations leave a substantial share of population exposed to air pollution above our suggested limits in 2050 - even under strong climate mitigation scenarios which come with co-benefits for air pollution measures [Rao et al, 2016, Figure 5] (see list of changes below).
- 5) With regards to justice: we go into detail in the Supplementary Methods for which we have now added a new sentence in the main text (see below additions)
 - a) On regional and local level, which is relevant for our assessment of just Earth system boundaries, we are far from seeing sufficient progress to stay within our suggested limits, see for example (WHO, 2021b). This is also noted in the cited reference (Quaas et al, 2022): “In contrast, over Southeast Asia, especially India,

aerosol retrievals from satellites show continuing increases throughout the period”. To consider this, we have added the following to the SI:

- i) Air pollutants have been declining in many regions around the globe since 2000, however, the aerosol concentrations over Southeast Asia have been rising during this period [Quaas 2022].
- b) Great inequality exists in terms of who pollutes and who suffers from air pollution [Rao et al. 2021, <https://www.nature.com/articles/s41893-021-00744-0>].

Changes to main text:

- Lines 415-419: “Considering this and the range of these studies (~0.05 to 0.20 of additional AOD difference), we assess that these shifts may become disruptive if the interhemispheric AOD difference, currently approximately 0.05, exceeds 0.15 due to air pollution or geoengineering-related aerosol asymmetries (Supplementary Methods).”
- Lines 427-433: Such local and regional guidance is needed because PM_{2.5} characteristics such as toxicity are highly place- and source-specific. Eighty-five percent of the world population is currently exposed to PM_{2.5} concentrations beyond this boundary and exposure to PM_{2.5} is estimated to cause 4.2 million premature deaths annually, with vulnerable groups being affected disproportionately more while polluting less. Air pollution scenarios based on globally successful stringent mitigation and pollution control show reductions in affected populations but areas of high air pollution might remain.

L342: all aerosols are short-lived.

Response: We deleted “short-lived” (line 422).

Referee #6: justice

This article deals with significant matters of real importance, but I do not believe it is in publishable form at this time. While there is, in my view, a strong need for work (indeed a sustained programme) of this sort, aiming to cross the divide between the social and natural sciences in climate matters (though the piece does not restrict itself to climate), this article does not currently do this in an altogether clear or persuasive manner.

Thank you for the comprehensive feedback on our manuscript. We are in broad agreement with your critiques. We have done our best to clarify our position on justice in response to your concerns within the limited words available. We identify the following five overarching critiques. Please see responses to your detailed comments below for specific changes made to the manuscript.

1. inadequate definition of Earth System justice: We have now included a clearer definition of Earth system justice and what it means;
2. inadequate explanation of our justice criteria (the “3 I” approach): We have now explained this better in the paper and how we have operationalized it.
3. some discussion needed of No significant harm: We have explained our definition, the difficulties in drawing a cut-off line between significant and no significant harm, and the simple way in which we have now tried to deal with this challenge.
4. inadequate embedding in the 2030 Agenda on SDGs and vague referencing to social goals: We now explicitly refer to the SDGs and how we engage with them.
5. need for more reflexivity in terms of the strengths and weaknesses of our approach. One of our strengths is our attempt to operationalize justice in quantitative terms. However this ambitious goal comes with many limitations. We have strengthened our “cautionary tone” in the manuscript (see responses below) and also now describe in the manuscript four general caveats to the justice approach applied in this paper (lines 1162-1172):
 1. “While staying within the just boundaries as set in this paper is crucial to avoid harm to significant sections of the human population, they are by no means guaranteeing just outcomes. Since just ends can be achieved with unjust means, meeting these boundaries without transformation could significantly harm current generations.
 2. While harm to humans is caused in part by increased exposure to biophysical changes, we recognize that harm is also a function of people’s social-economic vulnerability and lack of adaptive capacities. This is beyond the scope of the present paper.
 3. Our high levels of aggregation preclude systematic analysis of distributional justice issues in terms of what people/communities/countries are most harmed under what scenarios:
 4. We do not explicitly address possible trade-offs between our three justice criteria. For example, policy instruments for achieving ‘I1’ may well undermine ‘I3’ (e.g. limit access to resources for marginal people). Hence we call for redistribution, liability and compensation in the transformation section.”

The article is very ambitious and full of interesting and suggestive observations and intuitions. It opens numerous research paths, many of which, it seems to me, could comprise research projects in themselves. The distinction between ‘safe’ and ‘just’ boundaries is particularly intriguing and could be pursued in greater detail (though, as will become clear below, I am unpersuaded that it quite works in this piece).

I am not well equipped to assess the physical science claims made here, and will set those aside. I am better positioned to respond to the ‘social science’ side of this piece, on which I will concentrate.

A principal drawback of the piece as submitted is that multiple terms are invoked without much discussion, analysis, or consistency. A broad series of weighty concepts are introduced that are not adequately defined or explained—Anthropocene, ‘social goals’, justice (intergenerational, intragenerational, and interspecies), equity, ‘global commons’, ‘common heritage’—some of which are subsequently abandoned. Arguments and sources sometimes seem cherry-picked, with their implications or relevance not fully digested or explained (what is the ‘global commons’—where does it start and stop, and how is it to be managed?; what does the term ‘Anthropocene’ add to the analysis?). Insofar as there is discussion of these matters, it is late (403-446), incomplete, and limited in scope.

Response: Thank you for the prompt to clarify these terms.

- We have added a glossary of terms to the Supplementary Methods that defines the Anthropocene, our three justice criteria, and global commons, amongst others.
- We have taken your advice in your comments below to be more careful around our use of the term ‘Anthropocene’. The term is relevant because threats to humans from declining Earth system stability have become acute in the Anthropocene.
 - Lines 77-80: “Humanity is well into the Anthropocene, the new geological epoch where human pressures have put the Earth system on a trajectory moving rapidly away from the stable Holocene state of the past 12,000 years, the only state of the Earth system we have evidence of being able to support the world as we know it.”
- We have removed the term ‘common heritage’ from the manuscript
- We have clarified the types of social goals that we refer to in the Introduction (see response to comment below)

Several apparently key-terms are stapled onto the root term ‘earth systems’ (‘-boundaries’, ‘-stability’, ‘-change’, ‘-resilience’, ‘-justice’). These cross between the biophysical and the sociopolitical—as the authors note—but without any clarity as to how this can be done (ie how to relate physical ‘system boundaries’ to a sociopolitical notion such as ‘justice’). But this is key to the article, a central point of which is precisely to expand a set of measurements from the natural sciences into the social.

Response: Thank you for the prompt to clarify these terms. We have now added a glossary to the Supplementary Methods that defines Earth system resilience, Earth system change, and Earth system justice, among others.

Specifically in relation to the concept of Earth system justice (“**how to relate physical ‘system boundaries’ to a sociopolitical notion such as ‘justice’**”): we build on existing justice scholarship – global and planetary, as well as on energy justice, environmental justice, climate justice etc. We are happy to report that an article describing our Earth system justice approach has now been published in which this approach is described in detail. We now cite this article (Gupta et al., Nature Sustainability, 2022) in the manuscript; the accepted fulltext is available via a link on the opening page of this review document. We have also rewritten our discussion of Earth system justice for clarity. Broadly, our narrative is different from existing justice narratives in that it (a) aims to cover local to global or multi-level justice issues; (b) is based on the assumption that incremental justice will not be enough to address Earth system challenges facing us; (c) recognizes that avoiding biophysical tipping points is not enough to prevent harm, especially harm caused by current generations towards future generations.

Furthermore, our article provides a clear methodology for how we “**expand a set of measurements from the natural sciences into the social.**” We

1. assess safe Earth system boundaries (paragraph starting “First, we” on line 104)
2. analyse these safe boundaries using three justice criteria (paragraph starting “Second, we” on line 117)
3. propose alternative just Earth system boundaries where necessary (paragraph starting “Third, we” on line 139)

We have edited these paragraphs to further clarify our approach.

For example, at 135-8 the authors articulate their contribution in terms of providing ‘safe’ and ‘just’ boundaries in the ‘same units’:

“The [planetary boundaries] identify only safe biophysical boundaries. Social goals quantified in other approaches are not expressed in comparable units or examine only the consequences on the Earth system of human activities, not harm from Earth system change.” (See also 63-71).

If this is the objective, however, I fear the article is not altogether successful. To invoke an ‘earth system’ is not enough in itself to bridge between natural and social sciences—and it is not clear that the term (‘earth system’) survives the transition intact. Is the ‘biophysical’ earth system the same ‘system’ as the ‘justice’ one? Which kind of system does ‘resilience’ refer to? (‘Both’ would not be a clear answer here ... As the SSPs show – notably SSP1 vis-à-vis SSP5 – it is not necessarily the case that the form ‘resilience’ takes will be identical in both cases.) The complexities raised by this question are largely unaddressed in this piece.

Response: We have now clarified in the Glossary that, for the purposes of this article, Earth system resilience refers to the resilience of the biophysical Earth system.

We have also rewritten the sentence referring to “the same units” for clarity (lines 152-155): “We define just ESBs for avoiding significant harm using the same units as the safe ESBs for the same domains (for example, global mean surface temperature change for climate) and propose that actors use the stricter boundary to inform target-setting.”

Justice and safety

As noted, the piece introduces an intriguing distinction between ‘safe’ and ‘just’ boundaries, the former referring to the biophysical, the latter to the sociopolitical. This should prove to be a fertile distinction, but in the present piece, I don’t believe it quite works. Safe for whom; just for whom? The association of ‘safety’ with ‘tipping points’ would seem to indicate that ‘safe’ is invoked at species level: a planet entering uncontrolled feedback processes is not ‘safe’ for the human species per se. This may or may not be true—and could certainly be grounded more firmly—but one can accept its hypothetical / rhetorical force. ‘Just’, then, would operate at a level where the species itself is not (necessarily) at risk, but different groups and communities within it are at more or less risk than others (I am extrapolating here: this is not crystal clear from the article).

Response: We argue that our safe boundaries are a necessary but not sufficient condition for justice (lines 66-68 and 144-146). They are not fully ‘safe’ for all species or all people today. Our concept of safe operates at different scales for different boundaries: for climate it is global, but we also specify many safe boundaries at sub-global scales, based on local tipping points; in which ‘safety’ would apply at community or regional scales. We have added a clarification in this spirit to our glossary entry for ‘safe ESBs’ (Supplementary Methods).

If this is right, then much depends on knowing who benefits and who loses in any given scenario—and whether we can assess any one outcome as more ‘just’ than another. This is a fiendishly difficult exercise and, unsurprisingly, the piece largely shies away from undertaking it. However, the result is that we don’t really know why any one outcome would be more ‘just’ than another in most cases—except, possibly, in broad aggregate terms. (But I am sure the authors are aware that aggregate—ie utilitarian—modes of justice are controversial and contested.)

Response: Indeed, in this paper we do not engage in scenario building for establishing winners/losers. However, we do perform spatial analysis of where safe and just boundaries are currently transgressed (Fig 3). As described in the text (lines 452-456): “Transgression of ESBs is spatially widespread with two or more safe and just ESBs transgressed for 52% of the world’s land surface affecting 86% of the global population (Fig 3). Some communities experience many ESB transgressions, with four or more ESBs transgressed for 28% of global population but only 5% of global land surface. Spatial hotspot transgressions are therefore concentrated in regions of higher population density, raising major intragenerational justice concerns.”

However my assumption may be wrong: it sometimes feels as though the notion of ‘just’ boundaries are, like ‘safe’ boundaries, also invoked at species-level. If so, it seems doubtful that ‘justice’ is the right term at all, since justice is a fundamentally relational, inter-human, notion.

Response: The harm experienced by humans due to Earth system change is, of course, in turn caused by human pressures on the Earth system and is therefore a “inter-human notion”. Exploring these responsibilities is crucial and is the subject of our section “Operationalising the ESBs” (lines 478-516). More on that section below.

To expand on the point, the piece speaks (at 109) of three justice criteria: intergenerational, intragenerational and interspecies (although 109 is in fact grammatically hard to follow – see my note below). But none are explained. If the piece is relying on these three criteria, some sort of discussion of how they are supposed to function, and the inter-relation between them, is probably needed. There is a vast literature on ‘intergenerational’ justice, for example. Moreover, an ‘intergenerational’ and ‘intragenerational’ analysis need not result in the same, or even in non-contradictory, conclusions. The ‘interspecies’ criterion too needs fleshing out: it is not at all intuitive—if it is (as it seems) intended to apply between human and all other species, it begs the question as to what human have to offer other species at all. Unfortunately, words like ‘justice’ are not transparent: they have different meanings in different contexts, and can be used, quite correctly, in very inconsistent ways: the semantic import of a word like ‘justice’ cannot be assumed.

Response: Agreed. We hope that we have now clarified this matter by enlarging our explanation of our justice criteria (the “3Is”) and defining our approach to justice more clearly. We also agree that each of the 3Is may give different results, which we have sought to clarify in Extended Data Table 1 and the text. Please see the glossary in the Supporting Methods and main text lines:

- Introduction, lines 117-126: “Second, we use three criteria to assess whether adhering to the safe ESBs could protect people from significant harm: ‘interspecies justice and Earth system stability’ (I1); ‘intergenerational justice’ between past and present generations (I2a) and present and future generations (I2b); and ‘intragenerational justice’ (I3) between countries, communities and individuals through an intersectional lens. These criteria sit within a wider Earth system justice framework that goes beyond planetary and issue-related justice to take a multi-level transformative justice approach focusing on ends (boundaries and access levels) and means (Supplementary Methods).”
- Results, lines 174-182: “Adhering to these safe boundaries implements our ‘interspecies justice and Earth system stability’ criterion (I1, Extended Data 175 Table 1) and will safeguard future generations against significant harm from Earth system change (intergenerational justice: I2a, Extended Data Table 1), but may not avoid significant harm to current generations, particularly vulnerable populations (I2b and I3, Extended Data Table 1). Hence, we: (a) propose that some boundaries be made more stringent to protect present generations and ecosystems, (b) complement safe boundaries with local level standards to protect present generations and ecosystems, (c) if the boundary is

likely to cause considerable difficulties for present generations, propose that it is complemented with policies that account for distributive justice.”

- Methods: New paragraphs at lines 1129-1140 and 1173-1186
- Extended Data Table 1

These justice criteria, and our conception of Earth system justice more generally, is now also described in our recently accepted paper in *Nature Sustainability* (link to full text provided at start of this review document).

Regarding operationalising ‘interspecies justice and Earth system stability’ in particular, our boundary is not derived deductively from philosophical debates on interspecies and multispecies justice because such scholars have scarcely operationalized their analysis in quantitative terms. Instead, we use domain specific reasoning, logic and literature to inductively derive proposals for safe boundaries that promote some degree of interspecies justice and Earth system stability (I1). We argue that in most cases this meets the intergenerational target – because it addresses the impacts of current generations on future generations, but that it does not always address the impacts of past generations on present generations. Three limitations arise here, which are elaborated throughout the text (e.g. lines 1131-1135) and in Extended Data Table 1:

1. Some safe I1 just proposals are not safe enough for humans today – (I2a) (e.g. climate change, hence we call for more stringent targets;
2. Some safe I1 just proposals do not address local human exposure to pollutants (e.g. air pollution, hence we complement with local standards); and
3. Some safe I1 just proposals may limit access to resources (hence we call for redistribution, liability and compensation in a separate paper.).

Some consideration is given to this broad set of problems towards the end of the piece, between lines 403 and 446. This is hugely helpful, but I don’t believe it is sufficient, both because the (late) acknowledgement of the existence of this (significant) concern is not in itself enough to neutralise it, and because the discussion is so limited in scope. The cautionary note struck in (most of) that passage seems correct, but it also seems out of synch with the relatively assertive tone deployed in the earlier passages. Plus, I doubt it is plausible to refer to something as ‘just’ without first being upfront as to one’s own position on ‘justice’, which is (understandably) avoided here.

Response: Thank you for the feedback. We have now sought to introduce this “cautionary” tone into the Introduction. We have also clarified our position on justice throughout the manuscript; see response to the preceding comment.

- New text (lines 100-102): “Our proposed Earth system boundaries are based on existing scholarship, expert judgement, and widely shared norms such as Agenda 2030. They are meant as transparent proposals for further debate and refinement by scholars and the wider society.”

The relative unwillingness to grapple with these kinds of questions in any depth leads to some surprising assertions. So, for example, the authors write at 443-6: "We ... need to address legal, economic, infrastructural, political and social barriers to transformation and promote international collaboration and inclusive governance by multiple actors that practices, incentivises and legislates for a safe and just future". This is a political sentence with all the politics sucked out: it is nearly impossible to identify a 'we' that can coherently sit as agent in this sentence, and no hint is given as to how any such 'we' could in fact exercise such agency, requiring as a precondition, as it would, an initial global consensus on the set of immensely complicated matters identified just beforehand. To my ear, at this juncture, the somewhat less rigorous 'social science' side of the piece seems to have become unmoored from the more rigorous natural science side.

Response: Thank you for your critique of this paragraph. We agree that the monolithic 'we' was unhelpful. We have rewritten the translation and transformation sections (see lines 478-516) to emphasise actors and their actions. These new paragraphs also highlight (i) the important roles of researchers to develop common procedures for translating global budgets derived from ESBies to actors followed by science-based target settings by these actors to reduce anthropogenic pressures and resource use to a sustainable level at a faster rate, and (ii) the need to transform our current systems of governance, values, economics, technology, investment, trade and production to achieve equitable redistribution of finite resources and associated responsibilities

Other imprecise terms

The problem of inadequate definition and precision of terms arises frequently. So, for example, in the first passage cited above, the term 'social goals' seems key, but it is hard to parse: what does it refer to? Whose social goals? Again these are matters of significant dispute, with irreconcilable views existing across countries and disciplines (a point implicitly made at 408-410). The whole idea of 'social goals' at global level is fraught for all sorts of reasons, not least the absence of a governing body capable of defining what they might be and how to achieve them. Even at national level this kind of problem can prove intractable (a point implicitly made at 437-9).

For the purposes of an article such as this (and following the IPCC's lead), the SDGs might be adequate as proxy 'social goals' and indeed they get a mention (132-4, 175, 420-4). This is not a perfect solution as the SDGs deliberately avoid laying out either the policies that might achieve them or the trade-offs and politics these may entail. In any case, if the article wished to rely on the SDGs—as having at least international imprimatur—this would still require some active engagement with the substance of the SDGs themselves which does not take place here.

Also in the first passage cited above (which I am using as exemplary), much would seem to depend on the 'approaches' for which citation is to footnotes 22, 24-26 (Raworth, Van Vuuren, Hickel, O'Neill et al). These are solid citations—Raworth, Hickel and Van Vuuren

are distinguished authors whose work is important and well-known—but it is not obvious how they relate to one another (they do very different kinds of work) nor how they can ground the ‘social goals’ referred to here without quite a bit more explanation.

Response: Indeed, in this passage we use the term ‘social goals’ to refer to a wide range of previous works. Thank you for prompting us to more precisely bound which works we intend for this term to refer to. We now write, “The social goals related to access to or harm from natural resources adopted in Agenda 2030, doughnut economics and other approaches...” (lines 155-157)

With regard to our own work, yes, we have based our social goals on the SDGs as they both reflect growing knowledge and political consensus. Conscious of our word limits, we have made a clear link to the SDGs but do not further discuss them. We now refer more explicitly to the 2030 Agenda and the social goals adopted in this agenda (lines 152, 156 and 209).

For this reader the use of several other terms besides—change, transformation, resilience, stability, and ‘equitable’—is also hard to follow.

Response: Thank you for the prompt to clarify these terms. As described in response to a previous comment, we now include a Glossary where we define all these terms except ‘equitable’, which we have removed from the main text.

From this perspective, the core criterion settled on—‘no significant harm’ (NSH)—is emblematic. It is unclear what is meant by this expression, but it seems exceedingly unlikely that, for example, an immediate stop to greenhouse gas emissions (as apparently contemplated) would not result in ‘significant harm’ to many people in much of the world. Indeed arguably, at least a billion or so already live in circumstances entailing ‘significant harm’: what happens to these people if our mitigation efforts outstrip any substitute—or (more likely), if a new ‘green economy’ is largely limited to the rich world?

Response: We have now defined NSH in the Glossary (Supporting Methods) and discuss there how norms for what constitutes significant harm vary widely. We were not able to agree on a single cut off point, so we have used a combination of a quantitative range of exposure levels, how can we reduce harm locally and globally to current generations, and if that is not possible, how can liability and compensation (e.g. loss and damage) be used to address that. This has been an inductive exercise and had different implications for the different biophysical domains.

- Lines 219-221: “During COP-27 in 2022, developing 219 countries have indeed focused actively on issues of adaptation and loss and damage. Responsibility may 220 differ from domain to domain.”

That our results identify a just (NSH) climate boundary of 1°C for high exposure to significant harm (with other temperatures for other levels of exposure, Table 1) highlights that significant harm from climate change is already widespread.

- Lines 204-205: “At 1.0°C global warming, tens of millions of people were exposed to wet bulb temperature extremes (Figure 2), raising concerns of inter and intragenerational justice.”
- Lines 209-211: “Moreover, past emissions have already led to significant harm, including extreme weather events, loss of habitat by Indigenous communities in the Arctic, loss of land area by low lying states, and sea level rise or reduced groundwater recharge from changing glacial melt systems”

We acknowledge that mitigation of climate change could cause significant additional harm (lines 462-463: “Meeting these boundaries without transformation, however, could significantly harm current generations.”). A justice analysis of pathways for mitigation and adaptation is a priority for future work (line 599: “...Developing transformative pathways to navigate within safe and just ESBs”). We share the reviewer’s intuition, however, that “an immediate stop to greenhouse gas emissions” would result in very high levels of significant harm; we make no recommendation that such a goal is an appropriate response to the fact that exposure to significant harm is already high above 1°C.

The authors are alive (see 380-381) to the differing equity stakes for current/future generations in regard to rapid v slower mitigation action, which also applies to local v. global action, but the whole matter remains very vague. For example, (180-184):

“We conclude that to minimise exposure of hundreds of millions of people to significant harm, the just (NSH) boundary should be set at or below 1.0°C. Since returning within this boundary may not be achievable in the foreseeable future, adaptations and compensations to reduce sensitivity to harm and vulnerability will be necessary.”

The implication here is that some sort of trade-off is needed between the incremental rises above 1°C to which we are now committed, and the amount and kind of ‘adaptation and compensation’ needed in response. As a matter of ‘justice’, this trade-off is complex and steeped in politics—though this is arguably where the action is (or ought to be) in climate negotiations. Distributional questions like these are theoretically complex because our response tends to depend on our presuppositions about just distributions (as noted at 408-410). They are also practically complex, as they depend on the kinds of institutions in place now and how these might change. The authors do recognise the centrality of these questions (at 408-412, 426-7 and 437-446), but it seems clear that they regard them as falling outside their scope of analysis. The question, then, is how to approach/manage them.

Response: Agreed. We are of the opinion that the current damage of climate change worldwide is significant. The conservative “Economist” puts it like this: “Overshooting 1.5 does not doom the planet. But it is a death sentence for some people, ways of life, ecosystems, even countries. To let the moment pass without some hard thinking about how to set the world on a better trajectory would be to sign yet more death warrants”. We believe that even at current levels people are dying and being displaced. In relation to climate change, the problem has always

been that the targets were also set based on what was practicable and would protect future generations and not what would protect current generations. We just want to put forth the hypotheses that to reduce harm from climate change, more stringent boundaries are needed. It may not be achievable, but we have now evidence that there is considerable harm, even if there is considerable uncertainty in drawing the boundary between significant and non-significant harm. From a justice perspective, we cannot do more than raise these issues. We are not trying in this paper to change political institutions but to force some degree of recognition of the existing harm being caused.

Added text:

- Lines 219-221: “During COP-27 in 2022, developing countries have indeed focused actively on issues of adaptation and loss and damage. Responsibility may differ from domain to domain.”
- Lines 463-464: Meeting these boundaries without transformation, however, could significantly harm current generations.
- Lines 486-487: “Each principle carries different, often implicit underlying value judgments and associated biases regarding distributive justice”
- See also new paragraph on transformations, lines 497-516.

Recommendation

I imagine this article can make a significant contribution, but suggest it should curb its ambition somewhat, play to its strengths and state its limits upfront, indicate clearly its interest in connecting the natural and social sciences, and clarify the scope of its intervention—focusing on the ‘biophysical’, drawing in the sociopolitical, but noting that whereas ‘safe’ boundaries raise the problem of a broader ‘justice’ calculus, the latter will require significantly more programmatic work. The piece might build on the remark at 459, that it is merely a first step in a larger project attempting to knit ‘social’ and ‘physical’ sciences together. Indeed, I wonder if a little zooming out at that point would be useful too: the challenge is not merely empirical, it is also epistemological.

Response: Thank you for the advice to state limitations more upfront, which we have endeavoured to follow.

- We have expanded on the remark the reviewer noted (formerly line 459, now lines 585-599), including epistemological and other challenges for future work: “We offer our just (NSH) boundaries and corresponding justice criteria and assessment as a first integration of social and natural sciences for further refinement, in the spirit that the planetary boundaries were proposed over a decade ago. This integration requires addressing empirical and epistemological challenges, including the following. (1) Data and justice analyses are currently limited on harm from different levels of safe Earth system transgression. Assessment is needed on which countries will suffer first, which communities will be exposed first and which groups are more exposed to multiple impacts from biophysical domains, considering the multiple vulnerabilities and varying impacts they face. (2) Analysis of significant harm needs to integrate how it is

exacerbated by poverty and vulnerability, which have been magnified by the Great Inequalities of the Anthropocene. (3) Analysis of how our domain-specific interpretations of interspecies justice (11, Extended Data Table 1) match and link to the growing scholarship on interspecies and multispecies justice. (4) Further articulation of what are acceptable thresholds for no significant harm and how justice approaches to boundaries can be refined and tested to enhance legitimacy. (5) Developing standardised procedures and methods of translation to cities and companies that account for interactions between ESBs with due considerations for justice. (6) Developing transformative pathways to navigate within safe and just ESBs.

- We have added to the Introduction comments in a similar spirit (lines 100-102): “Our proposed Earth system boundaries are based on existing scholarship, expert judgement, and widely shared norms such as Agenda 2030. They are meant as transparent proposals for further debate and refinement by scholars and the wider society.”

I am unsure whether this piece is R&R given the currently wide recourse to an overbroad and underdetermined vocabulary. However, if R&R were recommended, it would require, it seems to me, at a minimum a much closer discussion of what is intended by ‘justice’, the three justice criteria, and ‘social goals’, ideally cognizant of the differences between the registers, assumptions, and evidentiary bases between the natural and social sciences. Some of the less central vocabulary (Anthropocene, global commons) could be removed and other terms more precisely defined. The link to the SDGs could be more focused and grounded. I would also tentatively suggest the discussion at 403-446 be moved up and expanded, and the nuance and circumspection in that passage injected throughout the analysis.

Response: We support the reviewer’s recommendations. As described in our responses to their previous comments above (and below), we have:

- Clarified and expanded open our understanding of justice, the three justice criteria, and “social goals”.
- Removed references to terms where not useful (e.g. Anthropocene at line 63 and ‘equitable’ throughout the manuscript) and defined others (see Glossary)
- Expanded on our link to the SDGs.
- Modified the Introduction to reflect the nuanced tone of the Discussion (formerly lines 403-446). In line with the editor’s recommendations, we have rewritten that passage (now lines 478-516) and condensed, rather than expanded it.

I additionally include below a number of more or less ad hoc reactions to specific passages in the first pages of this article.

63: It is not clear why the article begins with the words ‘In the Anthropocene.’ Is there something specific about being ‘in the anthropocene’ that entails the rest of the sentence? This seems unlikely, but if it is the case, the authors ought to tell us why.

Response: Thank you for identifying this matter. We have removed “In the Anthropocene” from the start of the sentence (line 63).

77: ‘Humanity is 70 years into the Anthropocene’. The authors won’t need me to point out that the process of dating the ‘anthropocene’ appears to have run aground – essentially as it has apparently run into a performative contradiction: it is unclear how we can determine a geological stratigraphic signature within a 70 year timespan. Moreover the 70-year date intends to ground the anthropocene in radioactive signatures to do with nuclear tests, which does not appear among the authors’ concerns, whereas intensive fossil fuel use, which does, rather aligns with 160 years of activity. I fear the opening sentence is neither clear nor epistemologically sound: it doesn’t kick the article off well.

Response: We have removed the reference to 70 years. The sentence now reads “Humanity is now well into the Anthropocene...” (line 77)

80: ‘undermine all life-support systems’. This is extremely vague and feels like hyperbole. All life-support systems? Everywhere? For all forms of life? There is no clear citation here but I doubt this sentence can be correct.

Response: Thank you for identifying this overreach. We have replaced “all” with “critical”: “... undermine critical life-support systems” (line 80)

84: ‘a precondition for equitable human development is a stable and resilient Earth system’. Whereas I imagine this sentence may be correct, it is not easy to follow. What is meant by ‘equitable’ here? Under what conditions would ‘human development’ be ‘equitable’ and have they ever been met (I imagine this is a question of geographical scope).

Response: We have rewritten this sentence - see next comment below.

85-6: ‘without equitable human development a stable Earth system may not be possible’. I have no idea what this is supposed to mean or why it would be supposed to be true? It crosses between the physical and social sciences without explanation. I am all for ‘equitable human development’ but I don’t think we achieve it by presuming it is needed for a ‘stable earth system’. Again there is no attempt at grounding or citation.

Response: We agree this passage was poorly worded. We have now edited it to (lines 87-90): “Given these interdependencies between inclusive human development and a stable and resilient Earth system, an assessment of safe and just boundaries is required that accounts for Earth system resilience and human wellbeing in an integrated framework.”

109: ‘we use three Earth system justice criteria (intergenerational, intragenerational and interspecies & Earth system stability)’. An Oxford comma may help with this grammatically troubled sentence, but is the implication that ‘intragenerational and

interspecies’ is one ‘justice’ criterion and ‘earth system stability’ another? Or does ‘interspecies’ go with ‘earth system stability’? Very hard to follow what either refers to – and does not become clearer.

Response: Thank you for the remark! Our intention is the latter: ‘Interspecies justice and Earth system stability’. We have edited this formulation throughout the manuscript to clarify this point, using inverted commas to bracket the terms.

127: ‘Earth system justice’. This is an interesting and ambitious term – which I assume is intended by the authors as itself a contribution to debate. Perhaps the piece needs to be foregrounded and carefully defined at the outset. What is ‘earth system justice’? The text then says ‘which must also enable access to resources for all and distributional and procedural fairness’. I understand the article is not attempting to enter into these criteria – but it is worth noting that these are immense but very vague ambitions, imposing a very high (if laudable) bar. Might the authors not worry that if the objective is unattainable in practice, the overarching argument becomes that much easier to dismiss?

Response: As mentioned above, we are now happy to be able to cite in the manuscript a companion piece on Earth system justice that has now been accepted for publication in Nature Sustainability. We include a link to the accepted manuscript on the opening page of this response document. Our present manuscript addresses a small though valuable part of the wider concept of Earth system justice elaborated in the accompanying piece.

Within the present manuscript, we have elaborated in the main text on our conception of Earth system justice to the extent that space allows.

- (lines 123-126) “These criteria sit within a wider Earth system justice framework that goes beyond planetary and issue-related justice and the doughnut to take a multi-level transformative justice approach focusing on ends (boundaries and access levels) and means (Gupta et al. 2021; Supplementary Methods)”
- See new table, Extended Data Table 1
- See also our definition of Earth system justice in the Glossary (Supporting Methods).

149-50: ‘assuming that all safe boundaries by definition meet our interspecies justice and Earth system stability criteria’. This seems a rather significant assumption, certainly in the former case (the latter sounds tautological?)!

Response: Indeed, we use the domain-specific safe boundary quantifications as an inductive implementation of interspecies justice. We have edited this sentence for clarity. It now reads (lines 174-178): “Adhering to these boundaries implements our ‘interspecies justice and Earth system stability’ criterion (I1, Extended Data Table 1) and will safeguard future generations against significant harm from Earth system change, but may not avoid significant harm to current generations, particularly vulnerable populations (I2 and I3, Extended Data Table 1).”

This approach is reasonable because the safe boundaries have been set at the levels that are needed to maintain and restore nature. However we recognise that more research is needed here and have added the following text (lines 593-595): “Critical areas for further research on safe and just ESBs include: ...(5) Analysis of how our domain-specific interpretations of interspecies justice (11. Extended Data Table 1) match and link to the growing scholarship on interspecies and multispecies justice;”

174-5: ‘the widely accepted “leave no one behind” principle’. I am not aware that there is such a ‘widely accepted’ principle: to the contrary, most policies by most states seems to assume the reverse. At a minimum a citation would be helpful here.

Response: The “leave no one behind” (LNOB) principle was accepted by 193 countries in Agenda 2030. We have added a citation at line 209. The reviewer is correct that the document is not legally binding.

179: ‘committed by greenhouse gas emissions’ a typo I assume.

Response: We have clarified here that humans commit certain long-term tipping element impacts through present-day GHG emissions.

- Lines 212-214: “The irreversible impacts from cryosphere and biosphere tipping elements that are committed by anthropogenic greenhouse gas emissions in the coming decades but which unfold over centuries or even millennia to come also threaten intergenerational justice”

Referee #7: cryosphere, climate

In this paper, the authors of Earth Commission propose five safe and just Earth system boundaries (ESBs) for climate, the biosphere, freshwater, nutrients, and air pollution. These proposed ESBs represents new specific targets of decarbonation and conservations in additional to the Paris Agreement of keeping the planet from warming more than 1.5°C above pre-industrial levels.

My major concern of the paper is the proposed ESB for the biosphere with “50-60% of global land surface covered by intact natural areas”, which needs a global scale of land reallocation and environmental governance without precedent in human history (Ellis, 2019). In addition, to achieve “50-60% natural areas” requires displacing land from agriculture, which has great consequences for food security and can cause significant harm for some nations or sub-populations (Mehrabi et al., 2018). Therefore, this “50-60% natural areas” ESB would cause significant harm for vulnerable populations and is not a just (no-significant-harm) ESB.

Response: Thank you for raising this important issue. We would like to clarify that:

- Our quantification of ‘no significant harm’ in this paper deals with the *exposure of people to harm from Earth system change*. We acknowledge that insufficient access to resources (such as land for agriculture) can also cause harm, but we have analysed this matter elsewhere (Rammelt et al., <https://www.nature.com/articles/s41893-022-00995-5>). We clarify this distinction in:
 - Our new glossary at the start of the Supplementary Methods: “Harm: negative impact on humans, communities and countries from Earth system change additional to background rates and to changes in vulnerability. For the purposes of this manuscript, we analytically separate matters of harm due to Earth system change from matters of sufficient access to natural resources.”
 - The main text: “We define harm as negative impacts on humans, communities and countries from Earth system change, additional to background rates” (lines 126-127).
- Nevertheless, adhering to the natural ecosystem area ESB is feasible without endangering food supply. We have added the following text to the Supplementary Methods (section 1.2.1, fourth paragraph):
 - “With the current natural ecosystem area at 45-50% (Table 1), restoration within the range 50-60% may be feasible. Agriculture is the largest contributor to loss of natural ecosystem area, and feeding a global population a healthy diet can be achieved without further land expansion (Willett et al. 2019); largely intact natural areas can also be compatible with human use and occupation. Thus access to sufficient food can be negotiated, managed and avoided through practices and policies that increase sustainable production, transition to healthy consumption, reduce loss and waste, and improve fair/just distribution.”
- In fact, our ESBs are consistent with the Convention on Biological Diversity’s restoration goals:

- Supporting Methods section 1.2.1, end of second paragraph: “This boundary value is approximately consistent with the scientific justification underlying the CBD, which prescribes 15% area restoration over the current natural area of 45-50% as living in harmony with nature, we emphasise that this comparison is an observation and do not use the CBD goals as part of the evidence for this boundary”
- We acknowledge that restoration of the biosphere will need to be implemented with careful consideration to local justice. See amended main text at lines 283-286:
 - “Specific interventions that secure functional integrity are highly local and are best implemented under local authority, knowledge and leadership, with additional interventions often needed to ensure that marginalised groups are not further disempowered but rather are given the space to use their knowledge and approaches to participate in such processes.”

My other concern is the proposed ESB of annual mean interhemispheric AOD difference <0.15 with regards to the aerosol forcing on global monsoon precipitation. The two cited references (citation 74-75) do not provide direct estimate of monsoon precipitation reduction with the scenarios of mean interhemispheric AOD difference <0.15. More importantly, the monsoon precipitation responses are highly sensitive to latitudinal distribution of aerosol forcing (Fasullo et al., 2019). In fact, citation 75 reports that “the summer monsoon precipitation over India decreases by about 21% for 15° N and 29% for 30° N injections” (Krishnamohan and Bala, 2022). Therefore, the ESB of annual mean interhemispheric AOD difference might be larger or smaller than 0.15 depending on the latitudinal distribution of AOD in each hemisphere.

Response

Thank you. In the Supporting Methods we have referred to the sensitivity to latitudinal distribution of aerosol forcing. (“It is further recognized that AOD difference and its impact on shifts in tropical precipitation and water availability is sensitive to the aerosol particle size, and the latitudinal and altitudinal distribution of reflecting aerosols emissions (Zhao et al., 2021).” We have moved this to the main text (see changes below), as well as the rationale for putting the ESB at < 0.15 interhemispheric AOD difference, which was previously also in the Supporting Methods. Based on the empirical observations and modelled sulfate injections we assess that an additional interhemispheric difference in AOD of 0.05 to 0.20 could lead to major disruptions to monsoon precipitation in the tropical monsoon regions. Given that the background rate is 0.05 on average and ~0.1 in boreal summer and that latitude and particle size matter, we stick with the lower bound leading to < 0.15 interhemispheric AOD difference. See below for text edits and the manuscript for supporting references.

Text edits

- Lines 401-420: “Observational data for the West African monsoon rainfall and climate modelling studies for the Indian monsoon have identified potential shifts in the location of the Intertropical Convergence Zone (ITCZ) triggered by differences in sulphate AOD between the Northern and Southern hemispheres. [...] The volcanic eruptions of El

Chichon in the 1980s (AOD difference of 0.07) and Katmai (AOD difference of 0.08) provide empirical examples, while modelled aerosol injections with AOD differences of ~0.2 and ~0.1 simulated declined precipitation in tropical monsoon regions. Hemispheric AOD difference and its impact on shifts in tropical precipitation are sensitive to the aerosol particle size, and the latitudinal and altitudinal distribution of reflecting aerosols emissions. Considering this and the range of these studies (~0.05 to 0.20 of additional AOD difference), we assess that these shifts may become disruptive if the interhemispheric AOD difference, currently approximately 0.05 on average and ~0.1 in the boreal spring and summer exceeds 0.15 due to air pollution or geoengineering-related aerosol asymmetries (Supplementary Methods).”

END OF RESPONSE DOCUMENT

Reviewer Reports on the First Revision:

Referee #1 (Remarks to the Author):

The revised version of the manuscript has addressed several of the issues that I and other reviewers have raised about the previous version. Notable improvements include a better explanation of the social "just" boundaries in the main text, the recognition that different thresholds may be chosen by different actors, an improved description of the levels of confidence, and the addition of a glossary. Still, there are some important issues remaining that I outline below.

1. Myself (reviewer 1) and Rev.3 had asked for a better explanation of the N/P global boundaries. I find that the new lines added by the authors in the main text still provide no real insights to why these thresholds were found.
2. I and reviewer 4 questioned the use of tipping points in defining the ESB. The authors reply that they do not rely exclusively on tipping points to set ESBs. However the authors never define what are tipping points. But the idea of some kind of threshold is inherent both to the definition of ESB and to the science guiding the identification of the boundaries itself. It would be important to define these terms more clearly and recognise that some of the tipping points not only have large uncertainty but will not necessarily result on a global "runaway climate" tipping point and/or that they have time scales much beyond the usual policy timescales or even the 50-100 year time scale used in global scenarios.
3. Although the treatment of uncertainty improved, it still seems to emphasise the confidence on the boundary and not so much the range of values for the boundary that would receive a high confidence. In addition, it may seem that this likelihood levels result from some rigorous probability assessment when they are in reality mostly qualitative assessments of certainty from expert opinion.
4. I was a bit surprised to find that the "Ethics and Inclusion Statement" is given as one of the ways to achieve scientific rigour. It's certainly important to improve analysis and to address some unfairness issues in science, but does not automatically lead to more rigour.
5. I still found the review of the criticisms of the ESBs to be a bit in passing and restricted to the ones addressed in this paper. It would be great if the authors could be a bit more self-critical still.
6. The authors continue to confuse the 15% CBD target and somehow fail to address my comments and the ones from rev. 7. First the 15% CBD target is for degraded ecosystems - see CBD documents that state "including restoration of at least 15 per cent of degraded ecosystems". So it's not in relation to the total area of the planet. In addition it just changed to be "30%", so the authors may want to update this value. Finally, I am not sure that any of those papers cite in 1.2.1 really scientifically justifies the 15% target (they were certainly published many years after the target was set, BTW the Riggio ref seems to be missing from the list of references). I continue to find this global ESB on natural ecosystem area a bit poorly defined, particularly without a more rigorous definition of what natural means (the one in the glossary is broad) and a link to a model or product that measures it.

7. The authors have made several improvements in explaining the "just" boundaries in response to my comments and the ones from rev. #6. But I still found Extended Data Table 1 a bit confusing or imprecise. I don't fully understand the key issues for I1 and they really seem to be the "weak link" in this concept. In addition the I2b key issues seem to be really be I2a key issues, while the key issue for I2a is mentioning I2b (?). It seems this table is still amenable to some improvements and simplification.

8. I am not an expert, but I found the answers regarding the issues with the aerosol boundary a bit unconvincing, and wondering if would be better to remove it from the analysis or to to lower the confidence in Ext Table 2 to low confidence.

Referee #2 (Remarks to the Author):

Dear authors,

thank you for all the effort spent on the improvement of the manuscript and Supplementary Material. The overall approach applied and assessment of the individual ESBs become now clearer to the reader due to better descriptions and explanations in most sections. This has unfortunately led to a longer text, but is unavoidable.

I propose to publish the manuscript without further revisions.

Referee #3 (Remarks to the Author):

Thanks for these revisions.

I have gone through the whole revised text and the replies to my comments and to those of other Reviewers and Editor. Overall, these authors have accounted for my comments and addressed the several concerns I had raised. I think the manuscript is acceptable from my side, at least about the nutrient cycling aspects.

Just one minor comment though: although I appreciated that the authors have introduced an Extended data Table 1 about Earth system justice criteria, I found this table quite unclear. I expected from this Table some clarifications about what interspecies, intergenerational and intragenerational would mean. However, the authors have put some quite long text in this Table that makes it quite unclear (especially about interspecies justice). I would encourage the authors to make this table much shorter and strictly focussed on definitions.

Referee #4 (Remarks to the Author):

In this revised version of a paper I previously reviewed, the authors have made revisions that more clearly delineate the scope and logic/methodology of their assessment, i.e. the identification of safe and just boundaries for a number of 'earth system domains' affected by anthropogenic climate change. In my view, the narrative is definitely clearer than before and consequently more likely to be found useful by stakeholders and researchers.

With regard to specific points I raised in my earlier review.

1. The revised text (main manuscript page 2) makes it clearer that the proposed ESBs do not primarily correspond to hypothesised tipping points, also including evidence of gradual decline in earth system functions, beyond historical variability. However, tipping points still retain a lot of emphasis, as evident from the details provided in the supplementary methods for each ESB. For example, the climate and cryosphere ESB is entirely framed in terms of the assessed likelihood of 'passing multiple tipping points' (Table S1 and Figure S1). This tipping points thinking is just one school of thought within climate science, as noted in my previous review, largely based in theoretical analysis and predictions using models. Admittedly physical climate science relies a lot on models because of the large and long scales of the processes and feedback mechanisms involved, but it would be better to be honest and state that tipping points are just one perspective, and largely informed by model-based analysis with concomitant uncertainty.

2. Why "multiple climate tipping points" as opposed to a single tipping point? The revised text proposed in response to this query doesn't make things any clearer. Crossing a single tipping point, e.g. the breakdown of the North Atlantic thermohaline circulation to take one familiar example, could likewise "systematically threaten the stability of the Earth's climate and biosphere".

3. The authors provided no response to my comments questioning the state of consensus around evapotranspiration-mediated degradation/loss of the Amazon rainforest. The supplement still states "Even at present-day warming levels we are already committed to losing minor parts of some tipping elements prior to wider tipping points being reached, for example some outlet glaciers in West Antarctica (Joughin et al. 2014; Rignot et al. 2014) or in the southeastern Amazon rainforest (Gatti et al. 2021), supporting >1°C as being unsafe."

4. The added value of ESBs relative to other frameworks such as planetary boundaries and doughnut economics. I am not an expert, but despite the authors' rebuttal they have not convinced me that the ESB concept adds novelty relative to existing comparable frameworks, taken together. In my previous review, I acknowledged that expressing boundaries for safety and justice in the same units could help decision makers set targets (such as the Paris Agreement's 1.5-2 degrees), but to some extent this could be seen as a trivial advance. The revised text on page 5 seems to imply that existing frameworks do not account for harm from Earth system change. This is contrary to my understanding. For example, SDG 13 on Climate Action. The blurb on SDG 13 on the UN SDG website states "The annual average economic losses from climate-related disasters are in the hundreds of billions of dollars. This is not to mention the human impact of geo-physical disasters, which are 91 percent climate-related, and which between 1998 and 2017 killed 1.3 million people, and left 4.4 billion injured." This is specifically highlighting harm to people from Earth system change as an

argument for climate action.

Referee #5 (Remarks to the Author):

I appreciate that the authors have addressed my comments and provided more clarification and justification of the choice of safe ESB related to aerosols. I have minor comments on the edited text:

- I think this section should be prefaced by stating that the ESB values for aerosol pollution also consider potential risks from geoengineering as this is not communicated anywhere in the paper except the last sentence of the first paragraph of this section.
- The sentence " ~ 0.1 simulated declined precipitation in tropical monsoon regions⁸⁷" reads odd. Is it 0.1 simulated AOD interhemispheric difference leads to declined precipitation in tropical monsoon regions in a modeling study?
- In the sentence - "...we assess that these shifts may become disruptive if the interhemispheric AOD difference, currently approximately 0.0589 on average and ~ 0.1 in the boreal spring and summer⁸⁷ exceeds 0.15...", I don't think citation 87 is the correct reference for the ~ 0.1 AOD difference in boreal and spring and summer based on observations.

Referee #6 (Remarks to the Author):

Thanks for sending this revised article, which I have now examined. This article is certainly improved from the previous version I saw. The language has been tightened in parts giving clearer signposting to the reader and avoiding some ambiguity or possible misunderstanding. For example, the language of 'Anthropocene' has either been removed or nuanced, which is an improvement. I also reiterate my sense that this article engages an immensely important theme and represents a potentially significant intervention, despite its flaws. I would think the piece is, from the perspective of 'justice' much closer to publishable standard now. I do have some concerns, which I lay out below. I am happy to leave a decision on whether or how to incorporate these to the editors. I am also happy for any or none of the below to be passed to the authors as you see fit.

I will not comment on the amendments or text relating to other reviewer's comments, and will focus on the three specific questions you asked, turning to your second question first (as it provides relevant context for the first question):

1. Do you think the discussion regarding justice is now sufficiently transparent and consistent to be acceptable as a first attempt?
2. Do you think the paper accepted by Nature Sustainability entitled 'Earth System Justice needed to identify and live within Earth System Boundaries' supports the authors' justice criteria and the concept of Earth system justice or do you think it has similar limitations?

3. We should be most grateful if you would look at the revised manuscript, and tell us whether you feel that the points raised in the previous round of review have been satisfactorily addressed.

A. Gupta et al (2022) article in Nature Sustainability

The Gupta et al article is interesting and very rich. For me it is a bit of a curiosity, as it sits very much in my own field, but is published in a journal from another discipline; it is too short to qualify as an article in the usual sense (at just under 5600 words not including citations); and it has the feel of a literature review or opinion piece. (Speaking as a journal editor myself, I would be intrigued to understand the peer review process leading to its acceptance.) It is largely devoted to the topic of justice and is considerably richer in its sourcing and deliberations than the present piece. It shows awareness of the many strands of 'justice' debates in contemporary global and climate matters (at lines 166-250), something lacking in the article here under review, including in its revised form.

From the perspective of the present article, the Gupta et al piece has the important function of supplying a much more thorough account of the term 'earth system justice' to which this article can refer. I do find this an exciting and innovative notion and one that can bear significantly more research and development. Although I am unsure it is, as yet, well-founded (see further below), the term's use in the present article is enhanced by the Gupta et al piece, and is somewhat clearer.

That said, the Gupta et al article has, in my view, three significant limitations. The first is that, whereas the piece aims to sketch the main arteries of thinking about climate and global justice it makes little reference to the principal figures in the decades-long debates in these fields. Henry Shue – by any measure a giant in this field – is entirely absent. So are Stephen Gardiner and Dale Jamieson. Meyer and Gosseries, who have done immense work to flesh out the notion of intergenerational justice (alongside Shue), are again entirely absent. Simon Caney, another immense figure, gets a single oblique mention, which cannot do justice (ahem) to the meticulousness of his work. The work of Steve Vanderheiden, Edith Brown Weiss and Peter Singer is also touched upon very briefly; Martha Nussbaum slightly more. The 144 footnotes are very rich, but insofar as they source the 'justice' debates sketched in the text it is at second or third-hand at best in most cases.

This needn't be a problem in itself but it does contribute to the second shortcoming, which is that none of the strands of 'justice' identified is entered into in any detail, and their inconsistencies and inadequacies are not engaged. These are big and difficult debates, and the positions staked out tend to be nuanced and detailed, striving for both consistency and realism. The Gupta piece provides competent sketches, showing a good awareness of the various positions canvassed – but they tend to appear like tinned offerings on a supermarket shelf. The contradictions and distinctions between them are not drawn out: they sit side by side synoptically. The very short length of the piece would tend to militate against any serious consideration of these positions in any case. So, for example, while it is correct to say that justice may be 'relational' and/or 'non-relational' (215-218), very much depends on where we stand on this spectrum. Are our obligations to everyone everywhere the same as those to our compatriots and families? Some may think the answer is an obvious yes (such as the authors of the present piece—though it's unclear), but a long history of politics and practice clearly disagrees. The piece does not note the importance of this distinction, especially to climate debate.

Finally, 'based on the definitions above', the piece arrives at its own preferred definition of justice at lines 254-5, as 'an equitable sharing among all people in the world of nature's benefits, risks and related responsibilities, within safe and just Earth system boundaries to provide universal life support.' Here the authors also introduce the '3Is' that will feature in the present article. However, since no attempt is made to express how their preferred definition is 'based' on the earlier definitions—which were not themselves assessed for their strengths, weaknesses, convergences and contradictions—it is impossible to know how this synthetic position has been arrived at. In fact where certain choices are made (justice is 'non-relational'; 'human exceptionalism' is rejected), they are not argued. The definition comprises an intuitively attractive set of desiderata but it is difficult to know how these would survive close scrutiny in light of the reality of ethical trade-offs, the availability of political and legal institutions, and the history of economic analyses and decisions.

B. Adequacy of revisions on 'justice'

Turning to your first question, then, I do think the piece is more transparent and consistent in this revision, and its case is plausibly helped by the existence of the Gupta et al. piece.

Importantly, the language on justice is more nuanced in many places (at eg 1124-1126). In particular, the 4 caveats at 1161-1171 acknowledge the limits of the study's analysis in a way that I believe is helpful. (I am surprised, though, that this qualification appears very near the end of a lengthy article – whereas the questions addressed will be at the forefront of many readers' minds from the moment the terms are introduced 30 pages earlier (at line 171).)

However, my sense is the '3Is' themselves remain problematic. No attempt is made to explain these terms in the glossary or supplementary materials, nor in their initial statement (line 171). A late paragraph (1128-1139) goes into more detail, but it is both difficult to follow and, I think, superficial. I do appreciate (as I noted before) that these conceptual concerns are too complex to fully manage in the context of this paper -- and so I welcome the caveats at 1161. But unfortunately I doubt the existing qualifications are sufficient to establish terminological security. For example, to define 'interspecies justice' (I1) as 'the relation between humans and nature' (Gupta et al at 243—there is no definition in the present paper) is plainly anthropocentric, a seeming performative contradiction. I am all for 'rejecting human exceptionalism' but it is impossible to know if this is being put forward as a policy prescription? And if so, who is to achieve it and how? On the other hand, 'justice between past and present generations' (I2a)—that is, the question of past responsibility—suffers from the reverse problem. This is a well-known and understood matter of perennial and critical importance in climate policy, yet the present paper largely brushes it aside. (My sense is the paper can retain I2b and I3, because the terms are well established and their usage here does not deviate significantly from intuition—though a reader versed in these matters may find their articulation unsophisticated.)

One approach for now might be:

(i) drop the notion of 'interspecies justice' altogether, or at a minimum clarify how it relates to 'biodiversity' (a well-established term, and one which does not imply that the mere achievement of non-extinction is a kind of 'justice'). Reading the authors' rebuttal, p35, I fear we may be slightly at cross-purposes on this: it is the opacity of this term that I am querying, not the (laudable) objective of preserving species!];

(ii) also remove 12(a) – justice between past and present generations -- which is far too important and complicated to be introduced briefly, as it is here, and then largely ignored. Past responsibility remains a critical component of any notion of ‘equity’ and ‘justice’ in climate policy, its thorniness notwithstanding. Setting it aside because it has been historically difficult begs the question as to why other desiderata laid out here will be any easier to achieve (clearly they will not); and (iii) state upfront that ‘intergenerational’ and ‘intragenerational’ justice are inter-related, and are likely to involve trade-offs. I would think the paper can be a little more sensitive to these trade-offs, especially given the very different local and national stakes in climate policy in both present and future.

If these three suggestions were taken up, they would nuance the text and also table 1, without involving significant further revisions. They can also be integrated easily with the discussion in supp materials 2.2 and table s9. Indeed the latter point (iii) is supported repeatedly in the supplementary methods doc. The authors may further benefit from looking at the work of Henry Shue and Simon Caney (and I also attach an article just now published in the European Journal of International Law entitled ‘Against Future Generations’ that may be of interest).

As a last point on this, my sense is that ‘justice’ is intended in this paper to mean minimising harm in climate actions in a manner that is as fair as possible to all affected constituencies, present and future. From this point of view, the definition of harm matters (see below). For what it’s worth, one gets the sense that the authors often mean ‘equity’ when they say ‘justice’ but perhaps wish to avoid the former term because of its tricky baggage in climate law/policy. Indeed I see a decision was made to remove the term in this revision (whereas, I note the Gupta paper occasionally reverts to ‘equity’ to explain ‘justice’). I would be concerned that, in both philosophy and law, ‘justice’ is not a good substitute for ‘equity’. The latter term is, I think, more easily grasped – a just outcome is not necessarily an equitable (ie fair) one – and stands in any case to suffer the same fate were it to replace the latter, since the problem is not with the concept itself but with the opposition it triggers. I am thus unsure that the removal of ‘equitable’ and ‘equity’ in this revision improves clarity.

C. Response to reviews

To your third question, some of which I have touched on above. I am very grateful for the authors’ rebuttal document which is informative and transparent. It is clear my (and other reviewers’) comments have been read closely and incorporated where the authors felt they could. Reading through the doc, I can see the authors and I are in agreement on much of the substance, that many suggestions made have been considered, and most have been met with relatively straightforward tweaks, which do, in many cases, successfully bracket the complexities and queries that might otherwise arise. There are many minor clarifications that I think help the text. The glossary too helps clarify a number of terms used (I wondered should it be referred to in the main text?).

In particular I found the discussion of ‘significant harm’ interesting and informative, notably in comparing different sources of definition. At the same time that paragraph is not always crystal clear and raises questions that I don’t believe are addressed in the piece. How should one understand ‘no deaths additional to background rates’? This seems to be a matter of drawing a boundary (as to what qualifies as the ‘background rate’), both temporally (now? 1850-1900? 1989?) and spatially

(can we really find a 'global' background rate given the vastly differing conditions around the world)? The causal chains of climate-related mortality are notoriously complex, but it is also true that rapid mitigation policies will also affect any background rate. Is the IPCC figure 10 million to 100 million or just 10 to 100 million? The latter does seem to comport with the IFRC figure, but also seems operationally inutile – whereas 10 million is a very high lower bound. (I note a discussion at bottom page 41 of the supplementary methods that seems relevant here.)

Whereas these matters could potentially be clarified, I can also see the sense of avoiding entering these (thorny) discussions and choosing a broader definition, as is done here using Gupta et al ('Existential and/or irreversible negative impact on countries, communities or people, such as substantial loss of life, livelihood and income, loss of access to nature's contributions to people, loss of land, chronic disease, injury, malnutrition and displacement'). Still, I am not sure this solves the problem. It seems a somewhat impatient response that amounts to relying on intuition. I don't think this is an indefensible posture, but I wonder do we not still need a clearer grasp of the kinds of research needed to get us past intuition on this? Ideas about harm are inherently relational and I wonder whether we can still (in 2023) speak as though hard trade-offs are avoidable in a way we could plausibly in 1991? In sum, my sense is this paragraph on significant harm tries to contain and bracket a potential minefield for the paper, but I don't believe it is altogether successful.

I find the glossary entry on 'global commons' confusing and unhelpful. It's an attractive concept but also an unwieldy one and I would suggest removing it if it is not to be further elaborated.

159-160: New line: 'Articulating sociopolitical notions such as justice and converting them into biophysical units enables a better understanding of the space within which humans can function.' I wonder is 'converting' the right term here? In general, I wonder can 'justice' really be expressed in 'biophysical units'? And, were it possible, would it be desirable? Once one has units, decisions may be based on thresholds; and once we move away from the (admittedly absolutist and unrealistic) thresholds presupposed in this piece, we may find ourselves in a utilitarian dystopia, with life-altering decisions based on arbitrary cut-off points. I would re-examine this language.

I do hope these thoughts are helpful as this piece is shepherded into publication, as I am sure it will be!

Referee #7 (Remarks to the Author):

I am satisfied with the Authors' responses to my comments.

Author Rebuttals to First Revision:

We thank the editor and reviewers for their continued constructive comments on the manuscript. Please find below our responses to the editor and reviewers detailing how we have modified the manuscript.

Referees' comments:

Referee #1 (Remarks to the Author):

The revised version of the manuscript has addressed several of the issues that I and other reviewers have raised about the previous version. Notable improvements include a better explanation of the social "just" boundaries in the main text, the recognition that different thresholds may be chosen by different actors, an improved description of the levels of confidence, and the addition of a glossary. Still, there are some important issues remaining that I outline below.

1. Myself (reviewer 1) and Rev.3 had asked for a better explanation of the N/P global boundaries. I find that the new lines added by the authors in the main text still provide no real insights to why these thresholds were found.

Response: We have further clarified in the manuscript that the N/P boundaries used here were developed in previously published work (lines 354-356). The global boundaries are based on model-based aggregation of critical local N/P concentrations related to eutrophication that are well established in the literature. In-depth descriptions of the rationale and methods are available in the cited papers with summaries provided here in Results, the Method section, and SI.

Given the reviewer's comment below, we emphasise that we are not proposing a "global tipping point" for N/P. Our boundaries "*are based on recent papers calculating subglobal and global agricultural nutrient losses, surpluses, and inputs from critical N and P concentrations in water and air, beyond which eutrophication occurs*" (lines 354-356).

2. I and reviewer 4 questioned the use of tipping points in defining the ESB. The authors reply that they do not rely exclusively on tipping points to set ESBs. However the authors never define what are tipping points. But the idea of some kind of threshold is inherent both to the definition of ESB and to the science guiding the identification of the boundaries itself. It would be important to define these terms more clearly and recognise that some of the tipping points not only have large uncertainty but will not necessarily result on a global "runaway climate" tipping point and/or that they have time scales much beyond the usual policy timescales or even the 50-100 year time scale used in global scenarios.

Response:

- The previous version of our manuscript defined tipping points in the Introduction (lines 105-107) and SI (second paragraph below Table S1). We have now added a definition to the Glossary (start of SI).
- While the science of climate tipping points is improving, we acknowledge that many tipping points indeed have large uncertainty. These uncertainties are summarised in this manuscript with more information available in the cited literature such as Armstrong

McKay et al. (2022). These uncertainties are one reason why we are careful to use multiple lines of evidence to set our climate ESBs. In addition to tipping point science, our climate ESBs are independently supported by paleoclimate evidence about past climate variability and by literature-based estimates of the effects of climate change on maintaining biosphere and cryosphere functions (see main text lines 175-177 and Table S1).

- We added to the manuscript (lines 178-180), incorporating the comment of reviewer 4 below: *“Some climate tipping points such as circulation collapse or Amazon dieback have high uncertainty or low confidence in their dynamics and potential warming thresholds, but the complementary paleoclimate and biosphere analyses independently support the safe climate ESB assessment.”*
- We make no claim nor implication of a “global ‘runaway climate’ tipping point”.
- We describe how tipping points are not the only sort of threshold we consider, including: *“To determine safe boundaries, we use assessments of tipping point risks among local and regional tipping elements, **evidence on declines in Earth system functions, analyses of historical variability and expert judgement...** We do not exclusively rely on tipping points for setting safe ESBs, however, and the **ESBs should not be interpreted as representing tipping points**”, and: **“Tipping element assessments in climate, biosphere and other Earth system domains are key, though not exclusive, evidence for our ESBs. Recent planetary boundary assessments instead emphasise risks related to departure from Holocene ranges of Earth system variability”** (emphasis added).*
- We acknowledge that several key tipping points relate to ice sheets with best estimate tipping timescales of 2 to 10 thousand years. Taking this long view is justified on the basis of an intergenerational justice approach, in which the implications for future generations of long-term changes committed to this century are accounted for: *“The irreversible impacts from cryosphere and biosphere tipping elements that are committed by anthropogenic greenhouse gas emissions in the coming decades, but which unfold over centuries or even millennia to come, also threaten intergenerational justice”* (lines 204-206).

3. Although the treatment of uncertainty improved, it still seems to emphasise the confidence on the boundary and not so much the range of values for the boundary that would receive a high confidence. In addition, it may seem that this likelihood levels result from some rigorous probability assessment when they are in reality mostly qualitative assessments of certainty from expert opinion.

Response:

- We report uncertainties for each ESB as either uncertainty ranges or as a series of likelihood levels. Checking whether the uncertainty ranges in Table 1 were also cited in the main text, we found that the uncertainty ranges for N and P were omitted, which we have now corrected (lines 352 and 371). We thank the reviewer for drawing this to our attention. We note that for climate, we are careful to qualify any boundary statement with the associated likelihood level, for example, *“Above 1.5 or 2.0°C warming, the likelihood*

of triggering multiple tipping points increases to high or very high, respectively.” (lines 187-188)

- We feel that our uncertainty analyses are actually more prominent than our confidence analysis in the manuscript. The confidence levels are relegated to an Extended Data Table, while the uncertainty ranges are listed in the main Table 1.
- We are not able to report the “range of values for the boundary that would receive a high confidence” due to limitations in the availability of high-confidence evidence. This approach is in line with assessments such as the IPCC.
- The reviewer is correct that the confidence assessment is an expert judgement, as also happens with, for example, the IPCC.
 - In response to this comment, we now clarify that expert judgement is part of our methods in the Introduction (lines 97, 104, 131 and 956) and indicate that the confidence assessment is based on expert judgement (line 956).

4. I was a bit surprised to find that the "Ethics and Inclusion Statement" is given as one of the ways to achieve scientific rigour. It's certainly important to improve analysis and to address some unfairness issues in science, but does not automatically lead to more rigour.

Response: We believe that diverse disciplinary and geographical inclusion are critical to improving scientific rigour. We agree that such inclusion is far from sufficient, and we provide other motivations for the rigour of our work (lines 854-860).

5. I still found the review of the criticisms of the ESBs to be a bit in passing and restricted to the ones addressed in this paper. It would be great if the authors could be a bit more self-critical still.

Response: We have incorporated the critique that the reviewer referred to in the previous round. We are happy to consider further specific recommendations on critiques to reference, focusing on ones relevant to this paper. We highlight that aspects of our work are highly provisional (lines 97-98 and 509-511) and suggest several avenues for future work (lines 511-523).

6. The authors continue to confuse the 15% CBD target and somehow fail to address my comments and the ones from rev. 7. First the 15% CBD target is for degraded ecosystems - see CBD documents that state "including restoration of at least 15 per cent of degraded ecosystems". So it's not in relation to the total area of the planet. In addition it just changed to be "30%", so the authors may want to update this value. Finally, I am not sure that any of those papers cite in 1.2.1 really scientifically justifies the 15% target (they were certainly published many years after the target was set, BTW the Riggio ref seems to be missing from the list of references). I continue to find this global ESB on natural ecosystem area a bit poorly defined, particularly without a more rigorous definition of what natural means (the one in the glossary is broad) and a link to a model or product that measures it.

Response: The 15% restoration basis for our ESB is in line with scientific studies indicating species extinction would be significantly reduced at this level. Strassburg et al. 2020 (see reference list in SI for all references) indicated the need for a 20% restoration of the current natural area, which is equal to approximately 10% of global land area. Jung et al. 2021 provided numbers in their figure 3 that indicate that with 60-65% of land area under intact nature, all species targets would be reached, requiring a restoration of about 15% of land area, on top of the current intact area of 50%. The adopted version of the GBF (Global Biodiversity Framework) now refers to restoring 30% of currently 'degraded lands', contrary to scientific advice provided at the time (Leadley et al. 2022). The document does not define what 'degraded lands' are. Multiple definitions of degraded land make the target difficult to implement. If we use the UNCCD GLO2(Global Land Outlook 2) as a reference, degraded land is estimated at between 20-40% of total land area (consistent with the range of Gibbs and Salmon, 2015, and see Leadley et al. cited above), directly affecting nearly half of the world's population and spanning the world's croplands, drylands, wetlands, forests, and grasslands. Restoring 30% of the upper bound of this degraded land range (40%) is about 10-15% of the global land area, which is consistent with what we propose.

- We have added text to the SI (section 1.2.1) indicating that this restoration requirement is in line with the "30% restoration of degraded lands" target of the GBF if we follow the estimate of global degraded lands by GLO2.
- We have added Riggio et al. to the reference list; thank you for noticing this omission.
- We have further clarified the definition of natural ecosystem area, including a reference to a measurement product, at the top of section 1.2.1.

7. The authors have made several improvements in explaining the "just" boundaries in response to my comments and the ones from rev. #6. But I still found Extended Data Table 1 a bit confusing or imprecise. I don't fully understand the key issues for I1 and they really seem to be the "weak link" in this concept. In addition the I2b key issues seem to be really be I2a key issues, while the key issue for I2a is mentioning I2b (?). It seems this table is still amenable to some improvements and simplification.

Response: Thank you for the feedback. We have shortened this table, focused on definitions (per reviewer 3 suggestion), converted it to a Box (Box 1) and promoted it to the main text (per editor suggestion).

8. I am not an expert, but I found the answers regarding the issues with the aerosol boundary a bit unconvincing, and wondering if would be better to remove it from the analysis or to lower the confidence in Ext Table 2 to low confidence.

Response: We appreciate this comment and agree that the effects of aerosols on the hydrological cycle are complex and involve large uncertainties. There is scientific consensus, as reported in the latest IPCC assessment (chapter 8, box 8.1), that the emissions of aerosols in the northern hemisphere (NH) have led to declining rainfall trends in the South Asian and North African monsoon regions from the 1950s to 1980s. Thus, there is high confidence in the physical mechanism by which the aerosol emissions from NH would influence the tropical

monsoons. However, we agree that the uncertainty in the quantification of the hemispheric AOD difference could be large due to aerosol-cloud interactions.

- Therefore, we have lowered the confidence in Extended Data Table 1 (previously ED Table 2) to low confidence, in line with your suggestion.

Referee #2 (Remarks to the Author):

Dear authors,

thank you for all the effort spent on the improvement of the manuscript and Supplementary Material. The overall approach applied and assessment of the individual ESBs become now clearer to the reader due to better descriptions and explanations in most sections. This has unfortunately led to a longer text, but is unavoidable.

I propose to publish the manuscript without further revisions.

Referee #3 (Remarks to the Author):

Thanks for these revisions.

I have gone through the whole revised text and the replies to my comments and to those of other Reviewers and Editor. Overall, these authors have accounted for my comments and addressed the several concerns I had raised. I think the manuscript is acceptable from my side, at least about the nutrient cycling aspects.

Just one minor comment though: although I appreciated that the authors have introduced an Extended data Table 1 about Earth system justice criteria, I found this table quite unclear. I expected from this Table some clarifications about what interspecies, intergenerational and intragenerational would mean. However, the authors have put some quite long text in this Table that makes it quite unclear (especially about interspecies justice). I would encourage the authors to make this table much shorter and strictly focussed on definitions.

Response: Agreed. We have shortened this table, focused on definitions (per your suggestion), converted it to a Box (Box 1) and promoted it to the main text (per editor suggestion).

Referee #4 (Remarks to the Author):

In this revised version of a paper I previously reviewed, the authors have made revisions that more clearly delineate the scope and logic/methodology of their assessment, i.e. the identification of safe and just boundaries for a number of 'earth system domains' affected by anthropogenic climate change. In my view, the narrative is definitely clearer than before and consequently more likely to be found useful by stakeholders and researchers.

With regard to specific points I raised in my earlier review.

1. The revised text (main manuscript page 2) makes it clearer that the proposed ESBs do not primarily correspond to hypothesised tipping points, also including evidence of gradual decline in earth system functions, beyond historical variability. However, tipping points still retain a lot of emphasis, as evident from the details provided in the supplementary methods for each ESB. For example, the climate and cryosphere ESB is entirely framed in terms of the assessed likelihood of 'passing multiple tipping points' (Table S1 and Figure S1). This tipping points thinking is just one school of thought within climate science, as noted in my previous review, largely based in theoretical analysis and predictions using models. Admittedly physical climate science relies a lot on models because of the large and long scales of the processes and feedback mechanisms involved, but it would be better to be honest and state that tipping points are just one perspective, and largely informed by model-based analysis with concomitant uncertainty.

Response: We appreciate the opportunity to clarify that the climate ESB is not solely framed in terms of the assessed likelihood of "*passing multiple tipping points*". As stated in the main text and SI, we use multiple lines of evidence to support the climate ESB. In the main text climate section, we explicitly state that "*We identify safe ESBs for warming (Table 1, Fig 1) based on minimising likelihoods of triggering climate tipping elements, maintaining biosphere and cryosphere functions, and accounting for Holocene (<0.5-1.0°C) and previous interglacial (<1.5-2°C) climate variability*" (emphasis added), followed by subsequent discussion of "*biosphere damage and the risk of global carbon sinks becoming carbon sources*". In the SI, Table S1 systematically presents these multiple lines of evidence. The part of that evidence base related to tipping points is further elaborated in Figure S1.

We thank the reviewer for prompting us through these comments to check the manuscript for how the evidence for the safe climate ESB is presented. We noticed that this point wasn't clear in Table 1.

- We have now clarified the evidence base for the safe climate ESB in Table 1.

2. Why "multiple climate tipping points" as opposed to a single tipping point? The revised text proposed in response to this query doesn't make things any clearer. Crossing a single tipping point, e.g. the breakdown of the North Atlantic thermohaline circulation to take one familiar example, could likewise "systematically threaten the stability of the Earth's climate and biosphere".

Response: Thank you for pushing us on this point. While the risk to Earth system stability rapidly escalates as more climate tipping points enter their possible or likely threshold ranges, we accept that a single climate tipping point could indeed systemically threaten climate & biosphere stability. We have now removed "*multiple*" (and our previous justification of it, lines 180-182) from the manuscript and simply discuss "*the likelihood of crossing climate tipping points*".

3. The authors provided no response to my comments questioning the state of consensus around evapotranspiration-mediated degradation/loss of the Amazon rainforest. The supplement still states "Even at present-day warming levels we are already committed to losing minor parts of some tipping elements prior to wider tipping points being reached, for example some outlet glaciers in West Antarctica (Joughin et al. 2014; Rignot et al. 2014) or in the southeastern Amazon rainforest (Gatti et al. 2021), supporting >1°C as being unsafe."

Response: A direct response wasn't included as it wasn't clear one was required, especially as the original comment "*accept[ed] an assessment like this has to capture 'best current understanding', and the wide uncertainty bounds around Amazon loss and other features in Figure S1 capture this in a seemingly reasonable way*". As discussed above in response to Reviewer 1, we agree that several climate tipping points have high uncertainties and/or low confidence in their dynamics and potential warming thresholds, including the Amazon rainforest. We reiterate that our safe climate ESB is independently supported by the palaeoclimate and biosphere functioning analyses.

We have added to the manuscript (lines 177-180): "*Some climate tipping points such as circulation collapse or Amazon dieback have high uncertainty or low confidence in their dynamics and potential warming thresholds, but the complementary palaeoclimate and biosphere analyses independently support the safe climate ESB assessment.*"

4. The added value of ESBs relative to other frameworks such as planetary boundaries and doughnut economics. I am not an expert, but despite the authors' rebuttal they have not convinced me that the ESB concept adds novelty relative to existing comparable frameworks, taken together. In my previous review, I acknowledged that expressing boundaries for safety and justice in the same units could help decision makers set targets (such as the Paris Agreement's 1.5-2 degrees), but to some extent this could be seen as a trivial advance. The revised text on page 5 seems to imply that existing frameworks do not account for harm from Earth system change. This is contrary to my understanding. For example, SDG 13 on Climate Action. The blurb on SDG 13 on the UN SDG website states "The annual average economic losses from climate-related disasters are in the hundreds of billions of dollars. This is not to mention the human impact of geo-physical disasters, which are 91 percent climate-related, and which between 1998 and 2017 killed 1.3 million people, and left 4.4 billion injured." This is specifically highlighting harm to people from Earth system change as an argument for climate action.

Response: We agree there is plenty of work that measures harm from Earth system change. Our claim at lines 144-152 is that there is little work that both (a) accounts for harm and (b) quantifies harm in comparable units to biophysical change to inform boundary- or target-setting. The quote provided by the reviewer does not meet criterion (b). Non-trivial results from our work include that accounting for justice in climate boundaries could tighten a 1.5°C boundary to 1.0°C.

Referee #5 (Remarks to the Author):

I appreciate that the authors have addressed my comments and provided more clarification and justification of the choice of safe ESB related to aerosols. I have minor comments on the edited text:

- I think this section should be prefaced by stating that the ESB values for aerosol pollution also consider potential risks from geoengineering as this is not communicated anywhere in the paper except the last sentence of the first paragraph of this section.

Response: Thank you for the suggestion. We have added the following sentence: "*We consider AOD differences and their potential impacts arising from natural emissions, anthropogenic emissions and stratospheric aerosol injection (solar geoengineering).*" (lines 382-384)

- The sentence "~0.1 simulated declined precipitation in tropical monsoon regions⁸⁷" reads odd. Is it 0.1 simulated AOD interhemispheric difference leads to declined precipitation in tropical monsoon regions in a modeling study?

Response: We apologise for the confusion that parts of the revised sentence, as indicated in the response document, did not make it into the main manuscript. The sentence is rewritten to (lines 393-395): "*The volcanic eruptions of El Chichon in the 1980s (AOD difference of 0.07) and Katmai (AOD difference of 0.08) provide empirical examples, while model-simulated AOD differences of 0.1 and 0.2 lead to declined precipitation in tropical monsoon regions.*"

- In the sentence - "...we assess that these shifts may become disruptive if the interhemispheric AOD difference, currently approximately 0.0589 on average and ~0.1 in the boreal spring and summer⁸⁷ exceeds 0.15...", I don't think citation 87 is the correct reference for the ~0.1 AOD difference in boreal and spring and summer based on observations.

Response: Yes, the correct reference is Vogel et al., 2022 [<https://doi.org/10.1029/2021JD035483>]. Thank you for pointing this out. This has been corrected (line 400).

Referee #6 (Remarks to the Author):

Thanks for sending this revised article, which I have now examined. This article is certainly improved from the previous version I saw. The language has been tightened in parts giving clearer signposting to the reader and avoiding some ambiguity or possible misunderstanding. For example, the language of 'Anthropocene' has either been removed or nuanced, which is an improvement. I also reiterate my sense that this article engages an immensely important theme and represents a potentially significant intervention, despite its flaws. I would think the piece is, from the perspective of 'justice' much closer to publishable standard now. I do have some concerns, which I lay out below. I am happy to leave a decision on whether or how to incorporate these to the editors. I am also happy for any or none of the below to be passed to the authors as you see fit.

I will not comment on the amendments or text relating to other reviewer's comments, and will focus on the three specific questions you asked, turning to your second question first (as it provides relevant context for the first question):

1. Do you think the discussion regarding justice is now sufficiently transparent and consistent to be acceptable as a first attempt?
2. Do you think the paper accepted by Nature Sustainability entitled ' Earth System Justice needed to identify and live within Earth System Boundaries ' supports the authors justice criteria and the concept of Earth system justice or do you think it has similar limitations?
3. We should be most grateful if you would look at the revised manuscript, and tell us whether you feel that the points raised in the previous round of review have been satisfactorily addressed.

A. Gupta et al (2022) article in Nature Sustainability

The Gupta et al article is interesting and very rich. For me it is a bit of a curiosity, as it sits very much in my own field, but is published in a journal from another discipline; it is too short to qualify as an article in the usual sense (at just under 5600 words not including citations); and it has the feel of a literature review or opinion piece. (Speaking as a journal editor myself, I would be intrigued to understand the peer review process leading to its acceptance.) It is largely devoted to the topic of justice and is considerably richer in its sourcing and deliberations than the present piece. It shows awareness of the many strands of 'justice' debates in contemporary global and climate matters (at lines 166-250), something lacking in the article here under review, including in its revised form.

Response: Gupta et al (2023) is a Perspective that was subject to 7 reviews in two rounds. Many new references were added but had to be removed at the end to meet journal specifications. In our view, environmental and earth system justice is not just the subject of study of philosophers, legal scholars, and social scientists but also natural scientists. This is

because discussing water justice is not only about the social issues but also about relevant hydrological and ecological issues.

From the perspective of the present article, the Gupta et al piece has the important function of supplying a much more thorough account of the term ‘earth system justice’ to which this article can refer. I do find this an exciting and innovative notion and one that can bear significantly more research and development. Although I am unsure it is, as yet, well-founded (see further below), the term’s use in the present article is enhanced by the Gupta et al piece, and is somewhat clearer.

That said, the Gupta et al article has, in my view, three significant limitations. The first is that, whereas the piece aims to sketch the main arteries of thinking about climate and global justice it makes little reference to the principal figures in the decades-long debates in these fields. Henry Shue – by any measure a giant in this field – is entirely absent. So are Stephen Gardiner and Dale Jamieson. Meyer and Gosseries, who have done immense work to flesh out the notion of intergenerational justice (alongside Shue), are again entirely absent. Simon Caney, another immense figure, gets a single oblique mention, which cannot do justice (ahem) to the meticulousness of his work. The work of Steve Vanderheiden, Edith Brown Weiss and Peter Singer is also touched upon very briefly; Martha Nussbaum slightly more. The 144 footnotes are very rich, but insofar as they source the ‘justice’ debates sketched in the text it is at second or third-hand at best in most cases.

Response: We understand this concern and respect these scholars but were required in the Gupta 2023 forthcoming article to cut our reference list from 160 to 70 and were asked to ensure we included justice scholars from diverse regions and identities. We cannot change what is in the paper at this stage but will try to include some of these scholars in papers where we can have a more extended bibliography and discussion.

This needn’t be a problem in itself but it does contribute to the second shortcoming, which is that none of the strands of ‘justice’ identified is entered into in any detail, and their inconsistencies and inadequacies are not engaged. These are big and difficult debates, and the positions staked out tend to be nuanced and detailed, striving for both consistency and realism. The Gupta piece provides competent sketches, showing a good awareness of the various positions canvassed – but they tend to appear like tinned offerings on a supermarket shelf. The contradictions and distinctions between them are not drawn out: they sit side by side synoptically. The very short length of the piece would tend to militate against any serious consideration of these positions in any case. So, for example, while it is correct to say that justice may be ‘relational’ and/or ‘non-relational’ (215-218), very much depends on where we stand on this spectrum. Are our obligations to everyone everywhere the same as those to our compatriots and families? Some may think the answer is an obvious yes (such as the authors of the present piece—though it’s unclear), but a long history of politics and practice clearly disagrees. The piece does not note the importance of this distinction, especially to climate debate.

Response: This is a very important insight. However, that paper focuses on safe and just boundaries and makes a first attempt at incorporating some justice perspectives into the safe boundaries. Identifying a number does not in itself tell us whether it could be achieved in a just way or how obligations from a justice perspective need to be defined. We were also bound by publisher requirements on word count and citations.

Finally, ‘based on the definitions above’, the piece arrives at its own preferred definition of justice at lines 254-5, as ‘an equitable sharing among all people in the world of nature’s benefits, risks and related responsibilities, within safe and just Earth system boundaries to provide universal life support.’ Here the authors also introduce the ‘3Is’ that will feature in the present article. However, since no attempt is made to express how their preferred definition is ‘based’ on the earlier definitions—which were not themselves assessed for their strengths, weaknesses, convergences and contradictions—it is impossible to know how this synthetic position has been arrived at. In fact where certain choices are made (justice is ‘non-relational’; ‘human exceptionalism’ is rejected), they are not argued. The definition comprises an intuitively attractive set of desiderata but it is difficult to know how these would survive close scrutiny in light of the reality of ethical trade-offs, the availability of political and legal institutions, and the history of economic analyses and decisions.

Response: This definition is the result of the Earth Commission’s discussion of justice in relation to the biophysical domains under consideration, and builds on the work on Access and Allocation within the Earth System Governance project; see also Gupta et al. 2023. The 3 I’s were based on the need to identify boundaries in relation to the use of resources. We built on Edith Brown Weiss’s seminal work on inter and intragenerational equity and added to that ‘interspecies and earth system stability’. Balancing between the 3I’s could be improved, but we felt it necessary to show what that would imply for the different domains because there are complicated trade-offs and challenges. Further, balancing between the 3I’s does not for example take into account the historical allocation of property rights in water. As a research paper, the authors aim to show, given the best available science, how safe boundaries may not be just – e.g. in the case of climate change and aerosols; and to show how safe boundaries may be just, but only if boundaries are drawn to take existing injustices into account. We think that if we have made these two points in the paper, we have succeeded in pushing the justice agenda further within the scope of this paper.

B. Adequacy of revisions on ‘justice’

Turning to your first question, then, I do think the piece is more transparent and consistent in this revision, and its case is plausibly helped by the existence of the Gupta et al. piece.

Importantly, the language on justice is more nuanced in many places (at eg 1124-1126). In particular, the 4 caveats at 1161-1171 acknowledge the limits of the study’s analysis in a

way that I believe is helpful. (I am surprised, though, that this qualification appears very near the end of a lengthy article – whereas the questions addressed will be at the forefront of many readers’ minds from the moment the terms are introduced 30 pages earlier (at line 171).)

Response: Thank you. The location of the caveats at the end of the paper is in accordance with the style of the journal, which allows very limited introductory material.

However, my sense is the ‘3Is’ themselves remain problematic. No attempt is made to explain these terms in the glossary or supplementary materials, nor in their initial statement (line 171). A late paragraph (1128-1139) goes into more detail, but it is both difficult to follow and, I think, superficial. I do appreciate (as I noted before) that these conceptual concerns are too complex to fully manage in the context of this paper -- and so I welcome the caveats at 1161. But unfortunately I doubt the existing qualifications are sufficient to establish terminological security. For example, to define ‘interspecies justice’ (I1) as ‘the relation between humans and nature’ (Gupta et al at 243—there is no definition in the present paper) is plainly anthropocentric, a seeming performative contradiction. I am all for ‘rejecting human exceptionalism’ but it is impossible to know if this is being put forward as a policy prescription? And if so, who is to achieve it and how? On the other hand, ‘justice between past and present generations’ (I2a)—that is, the question of past responsibility—suffers from the reverse problem. This is a well-known and understood matter of perennial and critical importance in climate policy, yet the present paper largely brushes it aside. (My sense is the paper can retain I2b and I3, because the terms are well established and their usage here does not deviate significantly from intuition—though a reader versed in these matters may find their articulation unsophisticated.)

Response: Agreed. We have now explained these terms in a Box (Box 1), following your and other reviewer suggestions. Our definition of ‘interspecies justice and Earth system stability’ is not based on a top-down definition and criteria, but on a domain specific analysis based on the scholarship. For example, our surface water boundaries aim to ensure species and ecosystem survival in line with historical records and reject human exceptionalism. But the groundwater boundaries are also based on the risks of subsidence for current generations as well as risks of low availability to humans and nature alike in the future. Thus there are different kinds of logic used to determine this. Taking I2a out does not make sense – our argument that 1.5 doesn’t protect current generations and hence 1.0 is better for climate change is trying to push the science forward on what is a just boundary for climate, even if unachievable, because it will show how unjust the climate target is.

One approach for now might be:

(i) drop the notion of ‘interspecies justice’ altogether, or at a minimum clarify how it relates to ‘biodiversity’ (a well-established term, and one which does not imply that the mere achievement of non-extinction is a kind of ‘justice’). Reading the authors’

rebuttal, p35, I fear we may be slightly at cross-purposes on this: it is the opacity of this term that I am querying, not the (laudable) objective of preserving species!];
(ii) also remove 12(a) – justice between past and present generations -- which is far too important and complicated to be introduced briefly, as it is here, and then largely ignored. Past responsibility remains a critical component of any notion of ‘equity’ and ‘justice’ in climate policy, its thorniness notwithstanding. Setting it aside because it has been historically difficult begs the question as to why other desiderata laid out here will be any easier to achieve (clearly they will not); and
(iii) state upfront that ‘intergenerational’ and ‘intragenerational’ justice are inter-related, and are likely to involve trade-offs. I would think the paper can be a little more sensitive to these trade-offs, especially given the very different local and national stakes in climate policy in both present and future.

Response:

Our paper refers to ‘interspecies justice and Earth system stability’ as a compound term. This term was defined using an inductive approach for each domain based on the nature of each system and its relation to species and ecosystems. We have taken option (iii), clarifying the tradeoffs in new text at lines 981-983. We have also added definitions of the 3I’s in Box 1 (a refined version of the former Extended Data Table 1).

If these three suggestions were taken up, they would nuance the text and also table 1, without involving significant further revisions. They can also be integrated easily with the discussion in supp materials 2.2 and table s9. Indeed the latter point (iii) is supported repeatedly in the supplementary methods doc. The authors may further benefit from looking at the work of Henry Shue and Simon Caney (and I also attach an article just now published in the European Journal of International Law entitled ‘Against Future Generations’ that may be of interest).

Response: Thanks for the paper; it is very interesting and we will take it into account in our follow-up justice papers.

As a last point on this, my sense is that ‘justice’ is intended in this paper to mean minimising harm in climate actions in a manner that is as fair as possible to all affected constituencies, present and future. From this point of view, the definition of harm matters (see below). For what it’s worth, one gets the sense that the authors often mean ‘equity’ when they say ‘justice’ but perhaps wish to avoid the former term because of its tricky baggage in climate law/policy. Indeed I see a decision was made to remove the term in this revision (whereas, I note the Gupta paper occasionally reverts to ‘equity’ to explain ‘justice’). I would be concerned that, in both philosophy and law, ‘justice’ is not a good substitute for ‘equity’. The latter term is, I think, more easily grasped – a just outcome is not necessarily an equitable (ie fair) one – and stands in any case to suffer the same fate were it to replace the latter, since the problem is not with the concept itself but with the opposition it triggers. I am thus unsure that the removal of ‘equitable’ and ‘equity’ in this revision improves clarity.

Response: We as a Commission chose to use the term justice because it is broader. Any choice would have its strengths and weaknesses.

C. Response to reviews

To your third question, some of which I have touched on above. I am very grateful for the authors' rebuttal document which is informative and transparent. It is clear my (and other reviewers') comments have been read closely and incorporated where the authors felt they could. Reading through the doc, I can see the authors and I are in agreement on much of the substance, that many suggestions made have been considered, and most have been met with relatively straightforward tweaks, which do, in many cases, successfully bracket the complexities and queries that might otherwise arise. There are many minor clarifications that I think help the text. The glossary too helps clarify a number of terms used (I wondered should it be referred to in the main text?).

In particular I found the discussion of 'significant harm' interesting and informative, notably in comparing different sources of definition. At the same time that paragraph is not always crystal clear and raises questions that I don't believe are addressed in the piece. How should one understand 'no deaths additional to background rates'? This seems to be a matter of drawing a boundary (as to what qualifies as the 'background rate'), both temporally (now? 1850-1900? 1989?) and spatially (can we really find a 'global' background rate given the vastly differing conditions around the world)? The causal chains of climate-related mortality are notoriously complex, but it is also true that rapid mitigation policies will also affect any background rate. Is the IPCC figure 10 million to 100 million or just 10 to 100 million? The latter does seem to comport with the IFRC figure, but also seems operationally inutile – whereas 10 million is a very high lower bound. (I note a discussion at bottom page 41 of the supplementary methods that seems relevant here.)

Response: IPCC uses a range from 10 million to 100 million. We agree that 10 million is a very high lower bound. We wanted to show that 1.5°C is not just because even at 1°C global warming tens of millions of people are already exposed to significant harm from climate change. We had discussed taking 1950 or 1970 as a base year and we may revert to that in follow-up papers. For this paper, we wanted to make the point that 1.5°C is not just.

Whereas these matters could potentially be clarified, I can also see the sense of avoiding entering these (thorny) discussions and choosing a broader definition, as is done here using Gupta et al ('Existential and/or irreversible negative impact on countries, communities or people, such as substantial loss of life, livelihood and income, loss of access to nature's contributions to people, loss of land, chronic disease, injury, malnutrition and displacement'). Still, I am not sure this solves the problem. It seems a somewhat impatient response that amounts to relying on intuition. I don't think this is an indefensible posture, but I wonder do we not still need a clearer grasp of the kinds of

research needed to get us past intuition on this? Ideas about harm are inherently relational and I wonder whether we can still (in 2023) speak as though hard trade-offs are avoidable in a way we could plausibly in 1991? In sum, my sense is this paragraph on significant harm tries to contain and bracket a potential minefield for the paper, but I don't believe it is altogether successful.

Response: Agreed. After three years of discussing these thorny issues, we chose a qualitative definition. It is not so much impatience but the diversity of views and difficulty on how to quantify such an issue.

Yes, the no significant harm narrative does open up a minefield. The International Law Commission tried to avoid doing so, particularly in relation to lawful activities two decades ago; but we think it does need opening up.

Hard trade-offs are not avoidable; society needs massive transformational change.

I find the glossary entry on 'global commons' confusing and unhelpful. It's an attractive concept but also an unwieldy one and I would suggest removing it if it is not to be further elaborated.

Response: Thank you for the suggestion. We elaborate the concept in several sentences in the glossary entry. We find it an important concept and prefer to retain it.

159-160: New line: 'Articulating sociopolitical notions such as justice and converting them into biophysical units enables a better understanding of the space within which humans can function.' I wonder is 'converting' the right term here? In general, I wonder can 'justice' really be expressed in 'biophysical units'? And, were it possible, would it be desirable? Once one has units, decisions may be based on thresholds; and once we move away from the (admittedly absolutist and unrealistic) thresholds presupposed in this piece, we may find ourselves in a utilitarian dystopia, with life-altering decisions based on arbitrary cut-off points. I would re-examine this language.

Response: Thank you for pointing this out. We are not converting justice into biophysical units. We are calculating e.g. what a minimum need for water implies in relation to water availability. This is what is unique and interesting about this paper; but we appreciate your point and have reformulated this passage (see line 151).

I do hope these thoughts are helpful as this piece is shepherded into publication, as I am sure it will be!

Referee #7 (Remarks to the Author):

I am satisfied with the Authors' responses to my comments.